# Mind the Gap: Mixtures of Gaussians in Approximate Differential Privacy

**Huikang Liu** [1]   **Aras Selvi** [2]   **Wolfram Wiesemann** [3]

## Abstract

We design a class of additive noise mechanisms that satisfy $(\varepsilon, \delta)$-differential privacy (DP) for scalar, real-valued query functions with known sensitivities, with a particular focus on moderate and low-privacy regimes. These mechanisms, which we call *mixture mechanisms*, are constructed by mixing multiple Gaussian distributions that share the same variance but differ in their means and mixture weights. The resulting distributions can be interpreted as convex combinations of a zero-mean Gaussian (as used in the analytic Gaussian mechanism, Balle & Wang 2018) and additional Gaussians whose means depend on the sensitivity of the query function. We derive tight conditions on the variances required for $(\varepsilon, \delta)$-DP and provide efficient algorithms to compute them. Compared to the analytic Gaussian mechanism, our mechanisms yield substantially lower expected noise amplitudes ($l_1$-loss) and variances ($l_2$-loss for zero-mean distributions). In the low-privacy regime that motivates our design, our mechanisms approach optimality, mitigating nearly all of the optimality gap of the analytic Gaussian mechanism.

## 1. Introduction

Differential privacy (DP, Dwork et al. 2006b; Dwork & Roth 2014) is a formal framework that enables the release of statistics of datasets while providing a quantifiable privacy guarantee to the individuals represented. By definition, DP guarantees hold under any privacy attack, including reconstruction (Dinur & Nissim, 2003; Dwork et al., 2017) and de-identification attacks (Sweeney, 1997; Heffetz & Ligett, 2014). As a result, differentially private (or privacy-preserving) variants of machine learning (ML) algorithms

offer protection against a broad range of threats, including membership inference attacks tailored to specific ML tasks and platforms offering ML as a service (Shokri et al., 2017; Hu et al., 2022). We refer the reader to the surveys (Ji et al., 2014; Gong et al., 2020) for DP in ML and (Demelius et al., 2025) for DP in deep learning. Much of the literature focuses on asymptotically high-privacy regimes, while many real-world deployments operate in moderate or low-privacy, often in single-shot settings.

While the definition of DP has inspired a broad range of related notions of computational privacy (Desfontaines & Pejó, 2020), we focus on the most widely adopted formulation, known as $(\varepsilon, \delta)$-DP, or approximate differential privacy (Dwork et al., 2006a). Approximate DP is a property of a randomized algorithm $\mathcal{A}$, which maps databases $D \in \mathcal{D}$ to random outputs $\omega \in \Omega$. Specifically, for any $\varepsilon, \delta \geq 0$, an algorithm $\mathcal{A}$ satisfies $(\varepsilon, \delta)$-DP if we have

$$\mathbb{P}[\mathcal{A}(D) \in A] \leq e^{\varepsilon} \cdot \mathbb{P}[\mathcal{A}(D') \in A] + \delta,$$
$$\forall (D, D') \in \mathcal{N}, \ \forall A \in \mathcal{F},$$

where $(\Omega, \mathcal{F}, \mathbb{P})$ is a probability space and $\mathcal{N} \subset \mathcal{D} \times \mathcal{D}$ is the set of neighboring databases that differ in a single individual. While the definition treats all values of $(\varepsilon, \delta)$ uniformly, the quantitative implications of DP depend strongly on the privacy regime, and mechanisms that are optimal in high-privacy regimes need not perform well when $\varepsilon$ is moderate or large. Due to the strong privacy guarantees promised to individuals (Dwork, 2011), various Bayesian (Vadhan, 2017, §1.6) and information-theoretic (Cuff & Yu, 2016) interpretations, and a wide range of properties such as composition (Dwork & Roth, 2014, Thm 3.16) and post-processing (Dwork & Roth, 2014, Prop 2.1), DP found manifold applications in statistics and ML (Chaudhuri & Monteleoni, 2008; Friedman & Schuster, 2010; Chaudhuri et al., 2011; Abadi et al., 2016; Cai & Kou, 2019), large language models (Harder et al., 2020; McKenna et al., 2025) and optimization (Mangasarian, 2011; Hsu et al., 2014; Han et al., 2016; Hsu et al., 2016). It also emerged as the *de facto* standard in industry (Desfontaines, 2021b).

In this work, we study *additive noise mechanisms* for approximate differential privacy, motivated by moderate and low-privacy regimes where reducing expected noise is critical for utility. Specifically, we perturb a scalar, real-valued query function $q : \mathcal{D} \mapsto \mathbb{R}$ by returning $\mathcal{A}(D) = q(D) + \tilde{X}$

[1] Shanghai Jiao Tong University [2] UCL School of Management [3] Imperial Business School. Correspondence to: Huikang Liu <hkl1u@sjtu.edu.cn>, Aras Selvi <a.selvi@ucl.ac.uk>, Wolfram Wiesemann <ww@imperial.ac.uk>.

*Proceedings of the 43rd International Conference on Machine Learning*, Seoul, South Korea. PMLR 306, 2026. Copyright 2026 by the author(s).

for a carefully designed random variable $\tilde{X}$, to share an approximate answer to the query $q(D)$ while preserving privacy via an additive noise $\tilde{X}$. Since $\tilde{X}$ is not a function of $D$, such noise mechanisms are data independent (see Nissim et al. 2007; McSherry & Talwar 2007 for examples of data dependent mechanisms). Typically, the noise $\tilde{X}$ must be a function of the privacy parameters $\varepsilon$ and $\delta$ as well as the sensitivity $\Delta$ of the query $q$ where $\Delta = \sup\{|q(D) - q(D')| : (D, D') \in \mathcal{N}\}$. Arguably, the most common mechanism in this context is the Gaussian mechanism (Dwork & Roth, 2014, App A), where $\tilde{X}$ follows a zero-mean Gaussian distribution with standard deviation $\sigma = \sqrt{2\log(1.25/\delta)(\Delta/\varepsilon)^2}$. While analytically convenient, this construction can introduce substantial excess noise in moderate and low-privacy regimes.

Differential privacy suffers from a *privacy-accuracy trade-off* (Alvim et al., 2011), where stronger privacy guarantees tend to deteriorate the utility gained from data (*i.e.*, $\mathcal{A}(D)$ may significantly deviate from $q(D)$). This can already be observed in the definition of the Gaussian mechanism, where the standard deviation of the noise increases as $\varepsilon$ and $\delta$ decrease. To this end, an active area of research aims to find optimal distributions that minimize a pre-specified loss, such as the standard deviation, amplitude $\mathbb{E}[|\tilde{X}|]$, and power $\mathbb{E}[\tilde{X}^2]$. The optimal noise distributions are characterized for the cases with either $\varepsilon = 0$ (Geng et al., 2019) or $\delta = 0$ (Soria-Comas & Domingo-Ferrer, 2013; Geng & Viswanath, 2014), corresponding to extreme or asymptotic privacy regimes. For privacy regimes with $\varepsilon, \delta > 0$, and in particular for moderate or low values of privacy, the design of optimal mechanisms is less understood. While there exist (near-)optimal mechanisms for specific tasks, such as returning integer-valued queries (Geng & Viswanath, 2015) or hypothesis testing for binary data (Awan & Slavkovic, 2020; Awan & Vadhan, 2023), it is shown that widely-adopted additive noise distributions can be suboptimal in the general setting where only the sensitivity of the query is known. Indeed, Balle & Wang (2018) argue that the Gaussian mechanism may be far from optimal in terms of its standard deviation $\sigma$ and derive the optimal Gaussian distribution, called the *analytic Gaussian mechanism*, by numerically tuning $\sigma$ for a given sensitivity $\Delta$ and privacy regime $(\varepsilon, \delta)$ with a bisection search method. However, Gaussian distributions are themselves suboptimal as demonstrated by Geng et al. (2020): a truncated Laplace mechanism, sampling noise from a Laplace distribution whose tails are truncated to a bounded interval, can significantly improve the noise amplitude and noise power attained by the analytic Gaussian mechanism in all privacy regimes.

Despite having stronger alternatives in terms of expected losses attained, Gaussian mechanisms may still be preferred in large-scale practice, especially in moderate-privacy regimes where composition, tail behavior, and simplicity

outweigh asymptotic optimality, with applications varying from the redistricting data analysis for the 2020 US Census (US Census Bureau, 2020) to Google's most recent large language model trained with DP (Sinha et al., 2025). The wide adoption of Gaussian mechanisms is due to several reasons, some of which we list. Unlike truncated distributions, the Gaussian distribution has an unbounded support and thus does not suffer from distinguishing events (events with $\mathbb{P}[\mathcal{A}(D) \in A] = 0$; Dwork & Rothblum 2016, Remark 1.3). Moreover, despite having unbounded supports, Gaussian distributions benefit from the empirical rule where *almost all* of the noise sampled is within $[-3\sigma, 3\sigma]$ (Desfontaines, 2021a). More importantly, they enjoy tight compositions (how $\varepsilon$ and $\delta$ accumulate over time, Bun & Steinke 2016; Dong et al. 2022) for applications with iterative releases of multiple $(\varepsilon, \delta)$-DP statistics. We show that mixtures of Gaussians may significantly improve the expected loss attained by a single Gaussian, particularly in moderate and low-privacy regimes, while retaining the desirable properties of Gaussian mechanisms. Our results suggest that widely deployed systems could benefit from lower-noise mechanisms without abandoning the Gaussian simplicity.

The paper unfolds as follows. Section 2 provides intuition for how the current state of the art in approximate DP motivates the use of Gaussian mixtures, and discusses their strengths and limitations. Section 3 introduces a mechanism that samples additive noise from a mixture of multiple Gaussian distributions with identical variances but varying means and mixture weights. It characterizes the parameters required for this distribution to satisfy $(\varepsilon, \delta)$-DP and develops an efficient algorithm to compute them. Since this algorithm has hyperparameters (*e.g.*, the number of distributions to mix), we also develop a hyperparameter-free alternative which is faster to optimize. To this end, Section 4 introduces a mechanism that samples additive noise from a mixture of a zero-mean Gaussian and a quasi-Gaussian distribution. Section 5 presents numerical experiments comparing the expected losses of our mechanisms with those of the analytic Gaussian mechanism as well as other families such as the truncated Laplace mechanism (Geng et al., 2020) and the Tulap mechanism (Awan & Slavkovic, 2020; Awan & Vadhan, 2023). We conclude the paper in Section 6. We borrow the standard notation in DP, which is elaborated on in our appendices. All proofs are relegated to appendices and source codes are made available[1].

## 2. Motivation, Promise, and Limitations

Recently, Selvi et al. (2025) proposed a numerical optimization framework to design optimal noise distributions that satisfy $(\varepsilon, \delta)$-DP for given $\varepsilon, \delta > 0$. Their method employs a cutting-plane-based optimization method. Such a numeri-

---

[1] https://github.com/selvi-aras/MindTheGap

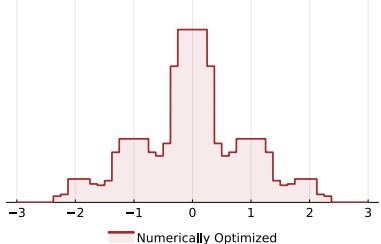 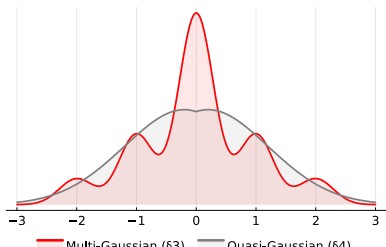

*Figure 1.* Different noise distributions that guarantee $(1, 0.1)$-DP for a query with $\Delta = 1$.

cal optimization framework is not free of drawbacks, since the optimal distributions lack closed-form representations, require iterative numerical optimization with no guarantees on the number of iterations, are significantly slower to tune than alternative mechanisms, and have bounded supports. Yet, they offer some key insights, visualized in the center of Figure 1: *(i)* the aforementioned privacy mechanisms (analytic Gaussian, truncated Laplace) can be significantly suboptimal in terms of expected losses, with suboptimalities reaching up to $700\%$ in the moderate and low-privacy regimes that motivate this work; *(ii)* optimal distributions are not monotonic and enforcing monotonicity strictly increases expected losses; *(iii)* optimal distributions appear to be multimodal with the density peaking at every $\Delta$-length interval and the density ratio within two consecutive peaks being approximately equal to $e^{\varepsilon}$. Building on these insights, we develop new privacy mechanisms based on mixtures of Gaussians, where each Gaussian is shifted by $\Delta$ from the others and scaled down by a factor of $e^{-\varepsilon}$. The right-hand side of Figure 1 shows an example of such distributions.

Our findings position themselves within a growing direction in DP research that seeks mechanisms tailored to non-asymptotic, moderate-privacy regimes, rather than relying exclusively on high-privacy limits. Recent work (Rinberg et al., 2025) shows that generalized Gaussians do not improve upon the standard Gaussian. We emphasize that this result applies only to *unimodal* generalizations. In contrast, our affirmative result demonstrates that mixtures of Gaussians do improve upon Gaussians, suggesting that new generalizations could be pursued through multimodal extensions.

A central motivation of this work is performance in moderate and low-privacy regimes, where $\varepsilon$ is not asymptotically small and Gaussian mechanisms exhibit their largest optimality gaps. Our results indicate that mixtures of Gaussians substantially improve upon the (analytic) Gaussian mechanisms across *all* privacy regimes (*cf.* Table 1), closing up to $99\%$ of the optimality gap associated with a single Gaussian. Moreover, recall that there exist non-Gaussian mechanisms that are asymptotically optimal as $\varepsilon, \delta \downarrow 0$, including the truncated Laplace mechanism. While these mechanisms attain near-optimal expected losses in high-privacy regimes,

they have two caveats: *(i)* they are less broadly applicable than the Gaussian mechanism, as they are designed to minimize noise variance and do not compose as Gaussians do (they lack Gaussian tails), and *(ii)* their optimality analyses are restricted to asymptotic privacy regimes. In response, we show that *(i)* although, similarly to these asymptotically optimal non-Gaussian mechanisms, we reduce the expected loss of the Gaussian mechanism by roughly half, our mechanism enjoys composition properties comparable to those of Gaussian mechanisms, due to its Gaussian tails; and *(ii)* despite non-Gaussian mechanisms being asymptotically optimal in high-privacy regimes, we significantly improve upon them in regimes with $\varepsilon \geq 1$ (*cf.* Figure 2), which is remarkable given that we are constrained to using mixtures of Gaussians. We note, however, that for $\varepsilon < 1$, we cannot improve upon the non-Gaussian objective, as this regime reflects fundamental limitations of Gaussian-based mechanisms. While this improvement depends on $\varepsilon$ it is notably independent of $\delta$. In practice, $\delta$ is required to be cryptographically small in differential privacy applications, whereas real-world deployments often operate at relatively large values of $\varepsilon$. For example, in industrial and governmental deployments, privacy budgets often exceed the high-privacy regimes commonly emphasized in the theoretical literature. In the consumer technology sector, Apple's local differential privacy configuration sets $\varepsilon = 4$ for emoji suggestions and allows for daily budgets of $\varepsilon = 16$ for Safari autoplay intent detection (Apple Privacy Team, 2017), while independent analyses of Apple's implementation have shown that cumulative privacy loss across features can be larger (*e.g.*, up to $\sim 14$ under some accounting assumptions) (Tang et al., 2017). Similarly, LinkedIn's labor market insights are reported to use a privacy budget on the order of $\varepsilon = 14.4$ to maintain analytical fidelity (Desfontaines, 2021b). In the public sector, the U.S. Census Bureau approved a global privacy-loss budget of $\varepsilon = 19.61$ for the seminal 2020 Redistricting Data (P.L. 94-171) release to balance accuracy and confidentiality in the released tabulations (Abowd et al., 2022). Open-source deep learning resources also report privacy budgets outside the high-privacy asymptotic regime, including $\varepsilon = 50$ in the Opacus DP-SGD tutorial (Opacus, 2026) and $\varepsilon = 2$ for Google's VaultGemma LLM (Sinha

et al., 2025). These deployments highlight a critical gap: mechanisms optimized for high-privacy regimes can incur substantial excess noise when operated at such moderate or low-privacy levels.

Finally, our approach is limited to one-dimensional queries. However, the relevant literature is similarly restricted to one dimension. For instance, the numerical-optimization-based mechanism of Selvi et al. (2025), which shows that unimodal or monotonic mechanisms need not be optimal, is one-dimensional. More generally, it is not known whether optimal mechanisms in higher dimensions can be unimodal. Similarly, the benchmark mechanisms considered in the literature, including the truncated Laplace and Tulap mechanisms, are all one-dimensional. Moreover, mechanisms that are close to optimal in higher dimensions may be suboptimal in one dimension. For example, the Flipped Huber mechanism of Muthukrishnan & Kalyani (2023), which modifies the Laplace mechanism in higher dimensions by combining Laplace-like bodies with Gaussian tails, is shown to be close to optimal in multiple dimensions, yet does not improve upon the truncated Laplace mechanism in one dimension. This suggests that progress in this setting requires one-dimensional certificates of optimality, rather than relying solely on high-dimensional or high-privacy asymptotics. In our conclusions, we briefly discuss bottlenecks and future directions for extending our results to more general distributions and higher dimensions.

## 3. Mixtures of Gaussians

We construct a distribution as a mixture of $2K + 1$ $(K \in \mathbb{N})$ Gaussian distributions with identical variances but different means and mixture weights. This distribution will be used for the additive noise $\tilde{X}$ in an $(\varepsilon, \delta)$-DP mechanism.

**Definition 3.1** (multi-Gaussian mixture). For privacy parameter $\varepsilon > 0$, query sensitivity $\Delta > 0$, scale parameter $\sigma > 0$, and modality parameter $K \in \mathbb{N}$, the *multi-Gaussian mixture distribution* is defined by the pdf

$$f_{\mathrm{m}}(x; \sigma, K) = \frac{1}{c_K} \sum_{k=-K}^{K} e^{-|k|\varepsilon} \phi(x; k\Delta, \sigma), \quad (1)$$

where $\phi(x; \mu, \sigma) = \exp(-\frac{(x-\mu)^2}{2\sigma^2})$ is the unnormalized pdf of a Gaussian with mean $\mu$ and variance $\sigma^2$, and $c_K = \sqrt{2\pi}\sigma \sum_{k=-K}^{K} e^{-|k|\varepsilon}$ is the normalization constant.

Because this mechanism is specified through a closed-form density, it is easy to work with and sample from once $\sigma$ is fixed. Indeed, Appendix B.1 reviews that Definition 3.1 specifies a well-defined distribution and plots it, derives its cdf, and discusses how to sample from it. Appendix B.2 derives the expectation of $|\tilde{X}|$ (noise amplitude) and $\tilde{X}^2$ (noise power) in closed form where $\tilde{X} \sim f_{\mathrm{m}}(\cdot; \sigma, K)$. Next,

we show that this distribution can be used to obtain a feasible additive noise mechanism. That is, for a query function $q : \mathcal{D} \mapsto \mathbb{R}$ with sensitivity $\Delta$, and for privacy parameters $\varepsilon, \delta$, there exists $\sigma > 0$ so that the randomized algorithm $\mathcal{A}(D) := q(D) + \tilde{X}$, $D \in \mathcal{D}$, satisfies $(\varepsilon, \delta)$-DP if $\tilde{X} \sim f_{\mathrm{m}}(\cdot; \sigma)$. We remark that for $K = 0$, Definition 3.1 coincides with the Gaussian distribution, and we know that there are feasible $\sigma > 0$ values providing $(\varepsilon, \delta)$-DP guarantees. On the other hand, for $K \geq 1$, the multi-Gaussian mixture mechanism exhibits a specific mixture structure with mixture weights chosen to be proportional to $e^{-|k|\varepsilon}$ and the Gaussian components centered at $k\Delta$. Appendix B.3 provides intuition for such construction in (1) for the case $K = 1$. We next establish the formal privacy guarantee of the multi-Gaussian mixture mechanism for any $K \in \mathbb{N}$.

**Theorem 3.2** (multi-Gaussian mechanism). *For privacy parameters $\varepsilon > 0$ and $\delta \in (0, 1)$, query sensitivity $\Delta > 0$, and the modality parameter $K \in \mathbb{N}$, consider an additive noise mechanism whose noise follows a multi-Gaussian mixture distribution with density $f_{\mathrm{m}}(x; \sigma, K)$. For a given $\eta \in (0, 1)$ and any $\beta \leq \sqrt{2\pi}\eta\sigma\delta$, this mechanism satisfies $(\varepsilon, \delta)$-DP if we have:*

$$\int_{-\infty}^{\infty} \min\{e^{\varepsilon} f_{\mathrm{m}}(x; \sigma, K) - f_{\mathrm{m}}(x + \varphi; \sigma, K), 0\}\mathrm{d}x$$
$$+ (1 - \eta)\delta \geq 0 \qquad \forall \varphi \in \{0, \beta, 2\beta, \ldots, \Delta\}. \quad (2)$$

Theorem 3.2 provides a sufficient condition for $(\varepsilon, \delta)$-DP by ensuring a stronger notion that replaces $\delta$ with $(1-\eta)\delta$, and in return avoids satisfying the definition over an uncountable set of neighbors by restricting it only on a discretized grid for $\varphi$. As $\eta$ approaches zero, this condition converges to the standard definition of $(\varepsilon, \delta)$-DP.

In the remainder of this section, we develop an algorithm to compute a tight value of $\sigma$ satisfying the DP condition in Theorem 3.2. In the numerical experiments, we then demonstrate that the multi-Gaussian mixture mechanism attains significantly smaller expected losses than the analytic Gaussian mechanism. However, since the sufficient DP condition (2) depends on the parameter $\eta \in (0, 1)$, these numerical gains do not themselves constitute a theoretical guarantee. We thus first show analytically that the multi-Gaussian mechanism is guaranteed to strictly improve upon the analytic Gaussian mechanism for sufficiently large $\varepsilon$.

**Proposition 3.3.** *For any $\delta \in (0, 1/2)$, there exists $\varepsilon_0 > 0$ such that we have $L_2^{\mathrm{MG}}(\varepsilon, \delta) < L_2^{\mathrm{AG}}(\varepsilon, \delta)$ for all $\varepsilon \geq \varepsilon_0$, where $L_2^{\mathrm{MG}}$ and $L_2^{\mathrm{AG}}$ are the expected $l_2$-losses (noise powers) of the multi-Gaussian mixture and analytic Gaussian mechanisms, respectively.*

The next result shows that condition (2) is monotonic in $\sigma$, hence we can search for minimum $\sigma$ satisfying this condition via bisection search.

**Lemma 3.4.** *For a given $\eta \in (0, 1)$, if $\sigma > 0$ satisfies (2), then any $\sigma' \geq \sigma$ must also satisfy (2). Furthermore, by construction, $\sigma = \sigma_{\mathrm{g}}$ satisfies (2) where $\sigma_{\mathrm{g}}$ denotes the standard deviation required by the (standard or analytic) Gaussian mechanism preserving $(\varepsilon, (1 - \eta)\delta)$-DP.*

The proof of Lemma 3.4 establishes a more general result that if a multi-Gaussian mixture mechanism with parameter $\sigma$ satisfies $(\varepsilon, \delta)$-DP, so does one with $\sigma' \geq \sigma$. This proves an analogous result to that of the Gaussian mechanisms: larger values of $\sigma$ correspond to stronger DP guarantees.

We now present Algorithm 1 that computes a value for $\sigma > 0$ to ensure that the multi-Gaussian mixture distribution with a given modality parameter $K \in \mathbb{N}$ and discretization parameter $\eta \in (0, 1)$ satisfies $(\varepsilon, \delta)$-DP. Note that this does not guarantee that we find the minimum possible such $\sigma$, since the condition (2) for $\sigma$ satisfying $(\varepsilon, \delta)$-DP (*cf.* Theorem 3.2) is itself a sufficient condition. However, upon fixing the hyperparameters $(K, \eta)$, we find the *smallest $\sigma$* satisfying (2), hence we find the optimal $\sigma$ within the scope of our analysis. In the presentation of Algorithm 1 we use ANALYTIC_GAUSSIAN($\varepsilon, \delta, \Delta$) to denote the computation of the standard deviation $\sigma_{\mathrm{g}}$ required by the analytic Gaussian mechanism and BISECTION($l, r, \psi(\sigma; \eta) = 0$) to denote the bisection search algorithm to find the root of a monotonic function $\psi(\cdot; \eta)$ within the region $(l, r)$. Algorithm 2, which is called within Algorithm 1, numerically computes the integral on the left-hand side of (2) multiple times. In our numerical experiments, we use the QuadGK package of the Julia programming language and propose some algorithmic improvements for numerical efficiency.

**Theorem 3.5.** *For any $K \in \mathbb{N}$ and $\eta \in (0, 1)$, Algorithm 1 returns the minimum $\sigma$ satisfying the DP condition in Theorem 3.2 in time $\mathcal{O}\left(\frac{K^2}{\eta \delta}\left(\log(1 + \frac{1}{\varepsilon}) + \log(1 + \log \frac{1}{\delta})\right)\right)$.*

Algorithm 1 should be viewed as a one-time offline calibration tool for the multi-Gaussian mixture mechanism, as opposed to a general-purpose alternative to black-box privacy verification methods (*e.g.*, Wang et al. 2023). Section 4 provides a hyperparameter-free mechanism with a faster algorithm which only logarithmically depends on $\delta$.

One of the key reasons behind the suitability of Gaussian mechanisms in ML, particularly for DP optimization algorithms including DP stochastic gradient descent (Abadi et al., 2016) and DP proximal coordinate descent (Mangold et al., 2022), is its favorable composition property where one can obtain tighter compositions than distribution-agnostic compositions. This is because the Gaussian mechanism fits well with the framework of zero-concentrated DP (zCDP, Mironov 2017; Bun & Steinke 2016; Dong et al. 2022), under which composition is lossless. Specifically, if mechanisms $\mathcal{A}_1$ and $\mathcal{A}_2$ are $\rho_1$-zCDP and $\rho_2$-zCDP, respectively, then their composition is $(\rho_1 + \rho_2)$-zCDP (Bun & Steinke,

---

**Algorithm 1** Tuning $\sigma > 0$ for density (1).

---

**Input:** privacy parameters $\varepsilon, \delta > 0$, sensitivity $\Delta > 0$, modality parameter $K \in \mathbb{N}$ and discretization $\eta \in (0, 1)$
**Output:** smallest $\sigma > 0$ (w.r.t. Thm 3.2) of multi-Gaussian mixture distribution (1) that satisfies $(\varepsilon, \delta)$-DP as an additive noise mechanism
Set $l = 0, r = \text{ANALYTIC\_GAUSSIAN}(\varepsilon, (1 - \eta)\delta, \Delta)$.
Set $\sigma = \text{BISECTION}(l, r, \psi(\sigma; \eta) = 0)$ where $\psi(\sigma; \eta)$ is the monotonically increasing privacy shortfall function whose root is sought. The computation of $\psi(\sigma; \eta)$ is from Algorithm 2.
**Return:** $\sigma$

---

**Algorithm 2** Computing the worst-case left-hand side of (2) for a given $\sigma > 0$.

---

**Input:** privacy parameters $\varepsilon, \delta > 0$, sensitivity $\Delta > 0$, modality parameter $K \in \mathbb{N}$, discretization $\eta \in (0, 1)$ and scale parameter $\sigma > 0$
**Output:** privacy shortfall $\psi(\sigma; \eta)$
**Initialize:** $\mathcal{S} = \emptyset$ and $\beta = \Delta / \lceil \Delta / (\sqrt{2\pi}\eta\sigma\delta) \rceil$
**for** $\varphi \in \{0, \beta, 2\beta, \dots, \Delta\}$ **do**
  Numerically compute the 1D integral

$$s = \int_{-\infty}^{\infty} \min\{e^{\varepsilon} f_{\mathrm{m}}(x; \sigma, K) - f_{\mathrm{m}}(x + \varphi; \sigma, K), 0\} \mathrm{d}x,$$

  and append $\mathcal{S} = \mathcal{S} \cup \{s\}$.
**end for**
Set $s_{\min} = \min\{\mathcal{S}\}$.
**Return:** $\psi(\sigma; \eta) = s_{\min} + (1 - \eta)\delta$

---

2016, Lemma 7). While we postpone the definitions to the supplementary materials, we review the crucial part below.

**Theorem 3.6** (Bun & Steinke 2016, Proposition 6)**.** *A Gaussian mechanism with parameter $\sigma$ satisfies $\rho$-zCDP with $\rho = \Delta^2 / 2\sigma^2$.*

Our mechanism enjoys an *identical* composition theory.

**Corollary 3.7.** *A multi-Gaussian mixture mechanism with parameter $\sigma$ satisfies $\rho$-zCDP with $\rho = \Delta^2 / 2\sigma^2$.*

Since the multi-Gaussian mixture mechanism can be viewed as a convex combination of Gaussians, the proof follows directly by exploiting the quasi-convexity of the $\alpha$-Rényi divergence (Bun & Steinke, 2016, Lemma 15), on which zCDP is defined. A proof is provided in Appendix B.8.

A direct composition in terms of $(\varepsilon, \delta)$-DP is as follows.

**Corollary 3.8.** *Consider $T$ multi-Gaussian mixture mechanisms applied to queries with sensitivities $\Delta_t$, $t \in [T]$, each using scale parameter $\sigma_t > 0$. Then, for any $\delta_{\mathrm{tot}} \in (0, 1)$, their composition satisfies $(\varepsilon_{\mathrm{tot}}, \delta_{\mathrm{tot}})$-DP with $\varepsilon_{\mathrm{tot}} = \rho_{\mathrm{tot}} + 2\sqrt{\rho_{\mathrm{tot}} \log(1/\delta_{\mathrm{tot}})}$ where $\rho_{\mathrm{tot}} = \sum_{t=1}^{T} \Delta_t^2 / (2\sigma_t^2)$.*

# 4. Quasi-Gaussian Mixtures

The multi-Gaussian mixture mechanism has a modality hyperparameter $K \in [N]$. Furthermore, for a fixed $K$, the quality of Algorithm 1 depends on a discretization parameter $\eta \in (0, 1)$. While in numerical experiments we show choosing $K \in [20]$ and $\eta = 0.01$ is a good rule-of-thumb, here we present a hyperparameter-free mixture mechanism. To this end, we define a distribution whose density function is the mixture of a zero-mean Gaussian and a quasi-Gaussian[2].

**Definition 4.1** (quasi-Gaussian mixture). For privacy parameter $\varepsilon > 0$, query sensitivity $\Delta > 0$, and scale parameter $\sigma > 0$, the *quasi-Gaussian mixture distribution* is defined by the pdf

$$f_{\mathrm{q}}(x; \sigma) = \frac{e^\varepsilon}{c} \exp\left(-\frac{x^2}{2\sigma^2}\right) + \frac{1}{c} \exp\left(-\frac{(|x| - \Delta)^2}{2\sigma^2}\right),$$

(3)

with normalization constant $c = \sqrt{2\pi}\sigma(e^\varepsilon + 2\Phi(\Delta/\sigma))$.

Appendices C.1-C.3 show that Definition 4.1 specifies a well-defined distribution and plot it, derive its cdf, and show how to sample from it. In Appendix C.4, we derive the expectation of $|\tilde{X}|$ (noise amplitude) and $\tilde{X}^2$ (noise power) in closed form where $\tilde{X} \sim f_{\mathrm{q}}(\cdot \, ; \sigma)$. Next, we show that this distribution can be used to obtain a feasible additive noise mechanism. That is, for a query function $q : \mathcal{D} \mapsto \mathbb{R}$ with sensitivity $\Delta$, and for privacy parameters $\varepsilon, \delta$, there exists $\sigma > 0$ so that the randomized algorithm $\mathcal{A}(D) := q(D) + \tilde{X}$, $D \in \mathcal{D}$, satisfies $(\varepsilon, \delta)$-DP if $\tilde{X} \sim f_{\mathrm{q}}(\cdot \, ; \sigma)$.

**Theorem 4.2** (quasi-Gaussian mechanism). *For privacy parameters $\varepsilon > 0$ and $\delta \in (0, 1)$, and query sensitivity $\Delta > 0$, an additive noise mechanism whose noise follows a quasi-Gaussian mixture distribution with density $f_{\mathrm{q}}(x; \sigma)$ satisfies $(\varepsilon, \delta)$-DP if $\sigma \geq \max\{\sigma_1, \sigma_2\}$ where*

$$\sigma_1 = \min\{\sigma : h_1(\sigma) + h_2(\sigma) \geq 0\} \qquad (4a)$$

*for*

$$h_1(\sigma) = e^{2\varepsilon}\Phi\left(-\frac{\varepsilon\sigma}{\Delta} - \frac{\Delta}{\sigma}\right) - \Phi\left(-\frac{\varepsilon\sigma}{\Delta} + \frac{\Delta}{\sigma}\right),$$

$$h_2(\sigma) = \left(e^\varepsilon + 2\Phi\left(\frac{\Delta}{\sigma}\right)\right)\delta,$$

(4b)

*as well as*

$$\sigma_2 = \min\left\{\sigma : \frac{f_{\mathrm{q,max}}(\sigma)}{f_{\mathrm{q,min}}(\sigma)} \leq e^\varepsilon\right\} \qquad (5a)$$

*for*

$$f_{\mathrm{q,max}}(\sigma) = \max_{x \in [0, \Delta]} f_{\mathrm{q}}(x; \sigma), \; f_{\mathrm{q,min}}(\sigma) = \min_{x \in [0, \Delta]} f_{\mathrm{q}}(x; \sigma).$$

(5b)

----

[2]The name 'quasi' follows since the term $\propto \exp(-(|x| - \Delta)^2/(2\sigma^2))$ replaces $x$ in a Gaussian density with $|x|$.

In Theorem 4.2, the definition of $\sigma_2$ is independent of $\delta$, hence, for small values of $\delta$, we have $\sigma_1 > \sigma_2$, while for large values of $\delta$ we have $\sigma_1 < \sigma_2$. Since the variance of the quasi-Gaussian mixture distribution scales with $\sigma$, we will be interested in the smallest $\sigma$ satisfying the above result, that is, $\sigma = \max\{\sigma_1, \sigma_2\}$. To this end, the rest of this section is devoted to developing efficient algorithms to find the exact values of $\sigma_1$ and $\sigma_2$. From here on, we fix the privacy parameters $\varepsilon > 0$ and $\delta \in (0, 1)$ as well as the query sensitivity $\Delta > 0$ and do not repeat their definitions.

**Lemma 4.3.** *The function $\sigma \mapsto h_1(\sigma) + h_2(\sigma)$ is monotonically increasing on the region $\sigma \in (0, \sqrt{2(\varepsilon - \log\delta)}\Delta/\varepsilon)$, and nonnegative for all $\sigma \geq \sqrt{2(\varepsilon - \log\delta)}\Delta/\varepsilon$. If additionally $e^\varepsilon + 2 \geq \delta^{-1}$, then this function is nonnegative everywhere.*

Lemma 4.3 shows that $\sigma_1$ as defined in (4a) satisfies $\sigma_1 \in (0, \sqrt{2(\varepsilon - \log\delta)}\Delta/\varepsilon)$ and the constraint in its definition satisfies monotonicity on this region, thus allowing us to search for $\sigma_1$ via bisection search. We next provide a similar result for $\sigma_2$. As a step towards showing this, we will first study the max and min problems of (5b) in the following abridged result (unabridged version is in Appendix C.7).

**Lemma 4.4** (abridged). *For any $\sigma > 0$, we have:*

*(i) $f_{\mathrm{q}}(\cdot; \sigma)$ is unimodal on the interval*

$$\mathcal{R}_{\mathrm{max}}(\sigma) = \left(0, \frac{\Delta - \sqrt{(\Delta^2 - 4\sigma^2)^+}}{2}\right)$$

*with a unique maximum. The maximizer also solves $\max_{x \in [0, \Delta]} f_{\mathrm{q}}(x; \sigma)$.*

*(ii) $f_{\mathrm{q}}(\cdot; \sigma)$ is unimodal on the interval*

$$\mathcal{R}_{\mathrm{min}}(\sigma) = \left(\frac{\Delta}{2}, \frac{\Delta + \sqrt{(\Delta^2 - 4\sigma^2)^+}}{2}\right)$$

*with a unique minimum. If $\mathcal{R}_{\mathrm{min}}(\sigma) = \emptyset$, then $x = \Delta$ solves $\min_{x \in [0, \Delta]} f_{\mathrm{q}}(x; \sigma)$. Otherwise, either $x = \Delta$, or the minimizer of the unimodal region solves $\min_{x \in [0, \Delta]} f_{\mathrm{q}}(x; \sigma)$.*

Lemma 4.4 allows us to find the values $f_{\mathrm{q,max}}(\sigma)$ and $f_{\mathrm{q,min}}(\sigma)$ as defined in (5b) for a given $\sigma > 0$ within unimodal regions, allowing the use of a golden-section search method. This lemma also leads us to the following result that will be helpful in computing $\sigma_2$.

**Lemma 4.5.** *The function*

$$\sigma \mapsto \max_{x \in [0, \Delta]} f_{\mathrm{q}}(x; \sigma)/\min_{x \in [0, \Delta]} f_{\mathrm{q}}(x; \sigma)$$

*is monotonically nonincreasing in $\sigma$, and is upper bounded by $e^\varepsilon$ at the point $\sigma = \sqrt{\Delta^2/(2\varepsilon)}$.*

Lemma 4.5 allows us to find $\sigma_2$ as defined in (5a) via a bisection search in the region $(0, \sqrt{\Delta^2/(2\varepsilon)})$.

We now present Algorithm 3 that computes a value for $\sigma > 0$ to ensure that the quasi-Gaussian mixture distribution satisfies $(\varepsilon, \delta)$-DP. Note that this does not guarantee that we find the minimum possible such $\sigma$, since the condition for $\sigma \geq \max\{\sigma_1, \sigma_2\}$ satisfying $(\varepsilon, \delta)$-DP (*cf.* Theorem 4.2) is itself a sufficient condition. However, we find the *exact* values of $\sigma_1$ and $\sigma_2$ in Theorem 4.2 so that we return the tightest possible $\sigma \geq \max\{\sigma_1, \sigma_2\}$ within the scope of our analysis. In the presentation of Algorithm 3 we use BISECTION$(l, r, \psi(\sigma) = 0)$ to denote the bisection search algorithm to find the root of a monotonic function $\psi$ within the region $(l, r)$. Moreover, Algorithm 4, which is called within Algorithm 3, uses GOLDEN$(l, r, f(x))$ to denote the golden-section search algorithm to find the maximum value of a unimodal function $f$ within the range $(l, r)$. In Appendix C.9, we share plots that visualize various steps of the algorithms presented in this section, for further intuition.

**Theorem 4.6.** *Algorithm 3 returns the minimum $\sigma$ satisfying the DP condition in Theorem 4.2 in time $\mathcal{O}\left(\log(1 + \frac{1}{\varepsilon}) + \log(1 + \log \frac{1}{\delta})\right)$.*

Similarly to the multi-Gaussian mixture mechanism in Section 3, here we developed an algorithm to compute a tight value of $\sigma$ satisfying the DP condition in Theorem 4.2. We will then demonstrate through numerical experiments in Section 5 that the quasi-Gaussian mechanism attains smaller losses than the analytic Gaussian mechanism in low-privacy regimes, while requiring substantially lighter computations than the multi-Gaussian mechanism. Since these numerical gains do not themselves constitute a theoretical guarantee, we show analytically that the quasi-Gaussian mechanism is guaranteed to strictly improve upon the analytic Gaussian mechanism for sufficiently large $\varepsilon$. This is an analogous result to Proposition 3.3 for the multi-Gaussian mixture mechanism.

**Proposition 4.7.** *For any $\delta \in (0, 1/2)$, there exists $\varepsilon_0 > 0$ such that we have $L_2^{\mathrm{QG}}(\varepsilon, \delta) < L_2^{\mathrm{AG}}(\varepsilon, \delta)$ for all $\varepsilon \geq \varepsilon_0$, where $L_2^{\mathrm{QG}}$ and $L_2^{\mathrm{AG}}$ are the expected $l_2$-losses (noise powers) of the quasi-Gaussian mixture and analytic Gaussian mechanisms, respectively.*

## 5. Numerical Experiments

In this section, we compare expected losses attained by multimodal Gaussian mechanisms with unimodal Gaussian mechanisms. Furthermore, we compare our multimodal Gaussian mechanisms with the non-Gaussian mechanisms that are asymptotically optimal in high-privacy regimes.

For each pair of $(\varepsilon, \delta)$, we tune the parameters of the quasi-Gaussian and multi-Gaussian mixture distributions via Algo-

---

**Algorithm 3** Tuning $\sigma > 0$ for density (3).

**Input:** privacy parameters $\varepsilon, \delta > 0$, sensitivity $\Delta > 0$
**Output:** smallest $\sigma > 0$ (w.r.t. Thm 4.2) of quasi-Gaussian mixture distribution (3) that satisfies $(\varepsilon, \delta)$-DP as an additive noise mechanism
**if** $e^\varepsilon + 2 < \delta^{-1}$ **then**
    Set $\sigma_1 = $ BISECTION$(l_1, r_1, \psi_1(\sigma_1) = 0)$ where $l_1 = 0$, $r_1 = \sqrt{2(\varepsilon - \log \delta)}\Delta/\varepsilon$ are the left and right limits of the bisection search region, and

$$\psi_1(\sigma) = e^{2\varepsilon}\Phi\left(-\frac{\varepsilon\sigma}{\Delta} - \frac{\Delta}{\sigma}\right) - \Phi\left(-\frac{\varepsilon\sigma}{\Delta} + \frac{\Delta}{\sigma}\right) + \left(e^\varepsilon + 2\Phi\left(\frac{\Delta}{\sigma}\right)\right)\delta$$

    is the monotonically increasing function whose root is sought.
**else**
    Set $\sigma_1 = 0$.
**end if**
Set $\sigma_2 = $ BISECTION$(l_2, r_2, \psi_2(\sigma_2) = 0)$ where $l_2 = 0$, $r_2 = \sqrt{\Delta^2/(2\varepsilon)}$ are the left and right limits of the bisection search region, and

$$\psi_2(\sigma) = \max_{x \in [0, \Delta]} f_{\mathrm{q}}(x; \sigma) \Big/ \min_{x \in [0, \Delta]} f_{\mathrm{q}}(x; \sigma) - e^\varepsilon$$

is the monotonically decreasing function whose root is sought. The computation of max- and min-terms are from Algorithm 4.
**Return:** $\sigma = \max\{\sigma_1, \sigma_2\}$

---

rithms 1 and 3. We then evaluate the resulting distributions in terms of their expected noise amplitudes, $\mathbb{E}[|\tilde{X}|]$ ($l_1$-loss; results in the main paper), and noise powers, $\mathbb{E}[\tilde{X}^2]$ ($l_2$-loss; results in the appendices), where $\tilde{X}$ is a random variable drawn from each of these noise distributions. All experiments fix the query sensitivity to $\Delta = 1$.

Our Julia implementation is provided in the supplements, where we rely on the `Roots` package for the bisection search method in Algorithms 1 and 3, the `Optim` package for the golden-section search method in Algorithm 4, and `QuadGK` for the one-dimensional integral evaluations in Algorithm 2. Any errors introduced by finite discretizations are handled conservatively, so that the resulting certificate still ensures DP. All experiments were run on Intel Xeon 2.66GHz cluster nodes with 16GB of memory in single-core and single-thread mode with a wall-clock time limit of 1-hour for our algorithms and 24-hours for the numerical-optimization lower bound benchmark (Selvi et al., 2025).

**Algorithm 4** Computing $\max$ and $\min$ problems (5b) for a given $\sigma > 0$.

---

**Input:** privacy parameters $\varepsilon, \delta > 0$, sensitivity $\Delta > 0$, and $\sigma > 0$ of $f_{\mathrm{q}}(\cdot; \sigma)$
**Output:** $f_{\mathrm{q,max}}(\sigma) = \max_{x \in [0,\Delta]} f_{\mathrm{q}}(x; \sigma)$ and $f_{\mathrm{q,min}}(\sigma) = \min_{x \in [0,\Delta]} f_{\mathrm{q}}(x; \sigma)$
**Initialize:**

$$x_1 = \frac{\Delta - \sqrt{(\Delta^2 - 4\sigma^2)^+}}{2}, \; x_2 = \frac{\Delta + \sqrt{(\Delta^2 - 4\sigma^2)^+}}{2},$$

$$t = -e^{\varepsilon} + \frac{\Delta - x_2}{x_2} \exp\left(\frac{2x_2\Delta - \Delta^2}{2\sigma^2}\right).$$

**if** $\Delta^2 \leq 4\sigma^2$ **then**
    Set $x_{\min} = \Delta$ and $x_{\max} = \mathrm{GOLDEN}(0, \frac{\Delta}{2}, f_{\mathrm{q}}(x; \sigma))$
**else if** $\Delta^2 > 4\sigma^2$ and $t \leq 0$ **then**
    Set $x_{\min} = \Delta$ and $x_{\max} = \mathrm{GOLDEN}(0, x_1, f_{\mathrm{q}}(x; \sigma))$
**else**
    Set $x_{\min} = \Delta$ and $x_{\max} = \mathrm{GOLDEN}(0, x_1, f_{\mathrm{q}}(x; \sigma))$
    Set $x_{\mathrm{tmp}} = \mathrm{GOLDEN}(\frac{\Delta}{2}, x_2, -f_{\mathrm{q}}(x; \sigma))$
    **if** $f_{\mathrm{q}}(x_{\mathrm{tmp}}; \sigma) < f_{\mathrm{q}}(x_{\min}; \sigma)$ **then**
        Update $x_{\min} = x_{\mathrm{tmp}}$
    **end if**
**end if**
**Return:** $f_{\mathrm{q,max}}(\sigma) = f_{\mathrm{q}}(x_{\max}; \sigma)$ and $f_{\mathrm{q,min}}(\sigma) = f_{\mathrm{q}}(x_{\min}; \sigma)$.

---

## 5.1. COMPARISON WITHIN GAUSSIANS

Recall that the analytic Gaussian mechanism gives the optimal *unimodal* Gaussian. The multi-Gaussian mechanism (1), on the other hand, specifies a class of *multimodal* Gaussians (reduces to the unimodal Gaussian when $K = 0$). Table 1 shows that multimodal Gaussians can significantly outperform unimodal Gaussians. The entries of the table represent $100\% \cdot (a - m)/\max(a, m)$ where $m$ and $a$ are the expected losses attained by the best multimodal Gaussian ($K \in \{1, \dots, 20\}$; fixes $\eta = 0.01$ in Algorithm 1) and the unimodal Gaussian mechanism, respectively. We observe that in $142/150$ of the cases, multimodality provides strict improvements (expected $l_1$-loss) over unimodality. Across all instances, the mean improvement is $53.73\%$ (sd $34.86$) and the median improvement is $61.86\%$.

We share more numerical experiments in appendices where we *(i)* report the best values of $K$ to obtain the entries of Table 1, *(ii)* revise Table 1 for the $l_2$-loss, *(iii)* share the percentage of the optimality gap multimodal Gaussians close (over the unimodal Gaussian) by using optimal lower bounds of Selvi et al. (2025), *(iv)* share a similar table to Table 1 under our lighter and hyperparameter-free quasi-Gaussian mechanism, *(v)* discuss the runtimes of our algorithms.

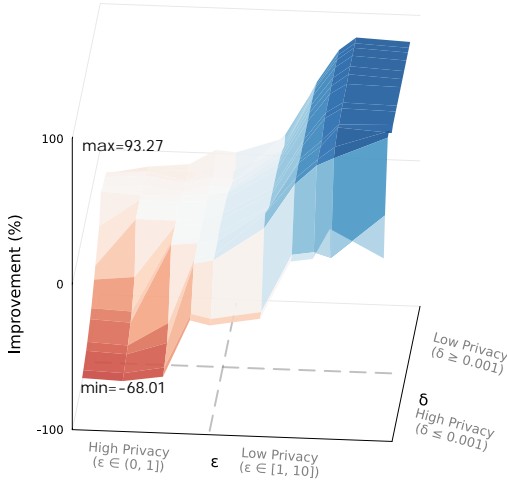

*Figure 2.* Improvement (% of $l_1$-loss) of the best mixture mechanism (blue) over the best benchmark mechanism (red).

## 5.2. COMPARISON BEYOND GAUSSIANS

As discussed in Section 2, mechanisms that are optimal in high-dimensional, high-composition, or high-privacy asymptotics may be strictly suboptimal in one-dimensional, one-shot, and low-privacy regimes. We address this by comparing our multimodal Gaussian mechanisms with the best mechanism out of the truncated Laplace mechanism (that is asymptotically optimal as $\varepsilon, \delta \downarrow 0$, Geng et al. 2020), Tulap mechanism (that is canonical for approximate DP, Awan & Slavkovic 2020; Awan & Vadhan 2023), staircase mechanism (that is optimal for $\delta = 0$, Soria-Comas & Domingo-Ferrer 2013; Geng et al. 2015), cactus mechanism (that is optimal under asymptotically large number of compositions, Alghamdi et al. 2022), and flipped Huber mechanism (that is designed under the notion of optimality in high dimensions, Muthukrishnan & Kalyani 2023). The results are reported in Table 2 (computed and reported similarly to Table 1) and further visualized in Figure 2. Since our $\varepsilon$ grid spans the interval $(0, 10]$, a direct plot of the improvements would misleadingly suggest that our method dominates across the entire range. To address this in Figure 2, we apply a monotone logarithmic transformation $\mathcal{T}$ to $\varepsilon$ such that $\mathcal{T}(0) = 0$, $\mathcal{T}(1) = 0.5$, and $\mathcal{T}(10) = 1$. This scaling ensures that the high-privacy regime ($\varepsilon \leq 1$) occupies half of the figure, while the low-privacy regime ($\varepsilon \geq 1$) occupies the other half. An analogous transformation is applied for $\delta$, with $\delta \leq 0.001$. We observe that our mechanisms offer a strict improvement over the best known approximate DP mechanisms when $\varepsilon \geq 1$. We share the data entries for the $l_2$-loss in our appendices.

*Table 1.* Improvement (% of $l_1$-loss) of multimodal Gaussians over the best unimodal Gaussian.

| $\delta \downarrow \mid \varepsilon \rightarrow$ | 0.1 | 0.25 | 0.5 | 0.75 | 1 | 2 | 3 | 4 | 5 | 10 |
|---|---|---|---|---|---|---|---|---|---|---|
| $5 \cdot 10^{-7}$ | NA | 2.12% | 16.74% | 61.20% | 60.39% | 65.19% | 78.55% | 79.75% | 85.53% | 82.54% |
| $10^{-6}$ | NA | 2.88% | 19.02% | 60.13% | 59.36% | 64.43% | 78.10% | 79.37% | 79.02% | 88.08% |
| $5 \cdot 10^{-6}$ | NA | 3.40% | 25.86% | 67.34% | 69.02% | 77.93% | 87.84% | 92.05% | 93.78% | 98.22% |
| $10^{-5}$ | NA | 3.86% | 35.99% | 66.04% | 67.80% | 79.16% | 87.58% | 91.87% | 94.68% | 98.31% |
| $5 \cdot 10^{-5}$ | NA | 6.01% | 60.00% | 62.29% | 64.20% | 74.90% | 86.78% | 91.11% | 95.48% | 95.37% |
| $10^{-4}$ | 0.08% | 7.32% | 57.62% | 59.79% | 63.14% | 72.86% | 85.76% | 90.75% | 94.71% | 95.63% |
| $5 \cdot 10^{-4}$ | NA | 15.04% | 50.47% | 53.45% | 57.57% | 69.36% | 84.10% | 91.87% | 94.78% | 95.28% |
| $10^{-3}$ | NA | 40.90% | 46.36% | 49.97% | 54.42% | 67.43% | 83.21% | 91.45% | 94.53% | 95.11% |
| $5 \cdot 10^{-3}$ | 0.25% | 27.93% | 34.86% | 39.39% | 44.23% | 61.43% | 80.50% | 90.20% | 93.78% | 94.60% |
| 0.01 | 13.91% | 19.84% | 27.68% | 33.74% | 38.03% | 57.84% | 79.12% | 89.54% | 93.89% | 94.32% |
| 0.02 | 8.35% | 15.33% | 18.67% | 27.76% | 29.75% | 53.27% | 77.18% | 88.74% | 93.42% | 94.00% |
| 0.05 | 8.05% | 9.69% | 9.57% | 17.49% | 18.73% | 44.93% | 73.74% | 87.24% | 92.63% | 93.47% |
| 0.1 | 8.05% | 6.73% | 0.37% | 13.71% | 13.13% | 35.70% | 70.07% | 85.67% | 91.81% | 92.95% |
| 0.15 | 3.17% | 3.75% | 3.47% | NA | 0.85% | 35.72% | 67.22% | 84.48% | 91.19% | 92.57% |
| 0.25 | 2.19% | 3.21% | 2.06% | 10.91% | 9.64% | 24.29% | 62.46% | 82.50% | 90.19% | 91.97% |

*Table 2.* Improvement (% of $l_1$-loss) of multimodal Gaussians (green) over the best approximate DP benchmark (red).

| $\delta \downarrow \mid \varepsilon \rightarrow$ | 0.1 | 0.25 | 0.5 | 0.75 | 1 | 2 | 3 | 4 | 5 | 10 |
|---|---|---|---|---|---|---|---|---|---|---|
| $5 \cdot 10^{-7}$ | -66.95% | -68.01% | -63.94% | -24.58% | -27.51% | -21.66% | 18.47% | 20.79% | 41.95% | 22.98% |
| $10^{-6}$ | -65.52% | -66.51% | -61.58% | -24.07% | -27.01% | -21.02% | 19.05% | 21.41% | 17.99% | 48.56% |
| $5 \cdot 10^{-6}$ | -61.46% | -62.80% | -54.02% | 1.13% | 3.99% | 27.18% | 58.14% | 71.70% | 77.19% | 92.74% |
| $10^{-5}$ | -59.32% | -60.78% | -44.33% | 1.39% | 4.13% | 33.70% | 58.67% | 71.95% | 81.07% | 93.27% |
| $5 \cdot 10^{-5}$ | -52.97% | -54.38% | 0.13% | 1.95% | 4.09% | 27.27% | 59.64% | 71.71% | 85.10% | 82.67% |
| $10^{-4}$ | -49.43% | -50.70% | 0.11% | 0.95% | 6.22% | 24.87% | 58.31% | 71.70% | 83.19% | 84.14% |
| $5 \cdot 10^{-4}$ | -39.02% | -36.27% | -0.08% | 0.76% | 5.93% | 24.79% | 58.30% | 77.54% | 84.93% | 84.14% |
| $10^{-3}$ | -33.44% | -0.52% | -0.43% | 0.73% | 5.62% | 24.70% | 58.28% | 77.54% | 84.93% | 84.14% |
| $5 \cdot 10^{-3}$ | -19.21% | -0.11% | -0.21% | 0.45% | 3.76% | 24.13% | 58.15% | 77.51% | 84.92% | 84.14% |
| 0.01 | -0.64% | -2.49% | -1.78% | 0.43% | 2.05% | 23.51% | 58.42% | 77.60% | 86.09% | 84.14% |
| 0.02 | -3.48% | -1.09% | -4.91% | 0.77% | -1.33% | 22.43% | 58.18% | 77.67% | 86.08% | 84.14% |
| 0.05 | -3.82% | -1.88% | -5.98% | -1.37% | -3.83% | 19.65% | 57.55% | 77.53% | 86.04% | 84.14% |
| 0.1 | -5.95% | -3.94% | -10.38% | 1.21% | -2.10% | 15.67% | 56.64% | 77.33% | 85.98% | 84.14% |
| 0.15 | -11.91% | -7.20% | -5.77% | -9.78% | -10.26% | 21.13% | 55.83% | 77.14% | 85.93% | 84.14% |
| 0.25 | -13.55% | -8.10% | -5.48% | 5.09% | 3.82% | 15.22% | 54.39% | 76.79% | 85.85% | 84.16% |

## 5.3. PRICE OF PRIVACY IN ML

While this paper focuses on designing privacy mechanisms with smaller one-shot expected losses, we also study the price of privacy in a ML setting and how unimodal and multimodal Gaussian mechanisms compare under composition. To this end, we consider building regularized logistic classifiers via DP proximal coordinate descent, which is a natural fit for our setting. This method constructs multi-dimensional classifiers through a sequence of one-dimensional coordinate updates, allowing privacy to be enforced via scalar noise injection via our mechanisms. Due to space constraints, we defer all ML experiments to the appendices. There, we study the generalization performance of different privacy mechanisms under composition. The results show that the multi-Gaussian mechanism consistently performs best, while the second-best mechanism alternates between the quasi-Gaussian and analytic Gaussian baselines.

## 6. Conclusions

We introduced new additive noise mechanisms for approximate DP based on Gaussian mixtures and developed efficient algorithms to tune their parameters to minimize expected losses under given $(\varepsilon, \delta)$-DP constraints. Our experiments show that, relative to the minimum-variance Gaussian mechanism, Gaussian mixtures yield substantially lower expected losses across all privacy regimes. In comparison with near-optimal non-Gaussian mechanisms, our approach continues to provide improvements when $\varepsilon \geq 1$ while additionally admitting a zero-concentrated DP guarantee identical to that of the Gaussian mechanism.

There are several avenues for future research. We did not optimize mixture weights in our distributions, and tuning them could offer further improvements in our mechanisms. Moreover, constituent distributions in our mixtures share the same scale parameter $\sigma$, and one could allow them to differ. We considered scalar queries similar to the non-Gaussian benchmarks, and extensions to multi-dimensional queries would increase the applicability of our mechanisms.

## Acknowledgements

The authors are grateful to the four anonymous reviewers for their comments and suggestions, which have helped improve the manuscript.

## Impact Statement

This paper presents work whose goal is to advance the field of Machine Learning. There are many potential societal consequences of our work, none which we feel must be specifically highlighted here.

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

## A. Notation

Throughout the proofs, we denote by $\log(\cdot)$ the natural logarithm. The cumulative distribution function of a standard Gaussian distribution is denoted by $\Phi$, and $\Phi'$ is its derivative (probability density function; pdf). For univariate $f$, we use $\mathrm{d}f(x)/\mathrm{d}x$ to denote its derivative, while for a multivariate function $\partial f(x,y)/\partial x$ denotes the partial derivative. The notation $\tilde{X} \sim f(\cdot)$ denotes a random variable whose distribution is specified by the pdf $f$. Bold lowercase symbols are vectors. The nonnegative part of a real number $x$ is $(x)^+ = \max\{0, x\}$. For a vector $\boldsymbol{x}$, the $l_p$ norm is denoted by $\|\boldsymbol{x}\|_p$, $p \geq 1$. We use $\mathcal{O}(\cdot)$ for the big O notation.

## B. Proofs for Section 3

### B.1. The multi-Gaussian mixture distribution is well-defined

To show that the density function (1) of the multi-Gaussian mixture distribution is well-defined, fix arbitrary $\sigma > 0$ and $k \in \mathbb{N}$, and note that $f_{\mathrm{m}}(x; \sigma, K) \geq 0$ for all $x \in \mathbb{R}$ since all terms in its definition are nonnegative. Moreover, observe that the density function integrates to 1 since each $\phi$-term is the unnormalized Gaussian density function where only the mean varies, that is,

$$\int_{-\infty}^{\infty} \phi(x; k\Delta, \sigma)\mathrm{d}x = \sqrt{2\pi}\sigma,$$

hence by the linearity of integrals, the density (1) integrates to 1. We can also simply represent the cumulative distribution function $F_{\mathrm{m}}(\bar{x}; \sigma)$ via Gaussian cumulative distribution functions:

$$
\begin{aligned}
F_{\mathrm{m}}(\bar{x}; \sigma) &= \frac{1}{c_K} \sum_{k=-K}^{K} e^{-|k|\varepsilon} \int_{-\infty}^{\bar{x}} \phi(x; k\Delta, \sigma)\mathrm{d}x \\
&= \frac{1}{c_K} \sum_{k=-K}^{K} e^{-|k|\varepsilon} \sqrt{2\pi}\sigma \Phi\left(\frac{\bar{x} - k\Delta}{\sigma}\right) \\
&= \frac{1}{\sqrt{2\pi}\sigma \sum_{k=-K}^{K} e^{-|k|\varepsilon}} \sum_{k=-K}^{K} e^{-|k|\varepsilon} \sqrt{2\pi}\sigma \Phi\left(\frac{\bar{x} - k\Delta}{\sigma}\right) \\
&= \frac{1}{\sum_{k=-K}^{K} e^{-|k|\varepsilon}} \sum_{k=-K}^{K} e^{-|k|\varepsilon} \Phi\left(\frac{\bar{x} - k\Delta}{\sigma}\right),
\end{aligned}
$$

where the first equality exploits the linearity of integrals, the second equality replaces the integral via a closed-form term using the Gaussian cumulative distribution, the third equality substitutes the definition of $c_K$, and the final equality cancels the $\sqrt{2\pi}\sigma$ terms.

Figure 3 visualizes the probability density function (1) of the multi-Gaussian mixture distribution for $\varepsilon = 1$ and $\sigma = 0.25$ for $K = 1$ and $K = 3$. For a better intuition, one can rewrite (1) as:

$$f_{\mathrm{m}}(x; \sigma, K) = \frac{1}{c_K}\left[\phi(x; 0, \sigma) + \sum_{k=1}^{K} e^{-k\varepsilon}\left(\phi(x; k\Delta, \sigma) + \phi(x; -k\Delta, \sigma)\right)\right].$$

Note that sampling from this distribution is straightforward since for $k \in \{-K, \ldots, K\}$, we sample from the $k\Delta$-mean Gaussian distribution with standard deviation $\sigma$ with probability $e^{-|k|\varepsilon}/\sum_{k=-K}^{K} e^{-|k|\varepsilon}$.

### B.2. Noise amplitude and power of the multi-Gaussian mixture distributions

For the *noise amplitude* of the random variable $\tilde{X}$ with density $f_{\mathrm{m}}(\cdot; \sigma, K)$, we have:

$$\mathbb{E}_{\tilde{X}}[|x|] = \frac{1}{c_K} \sum_{k=-K}^{K} e^{-|k|\varepsilon} \int_{-\infty}^{\infty} |x|\phi(x; k\Delta, \sigma)\mathrm{d}x.$$

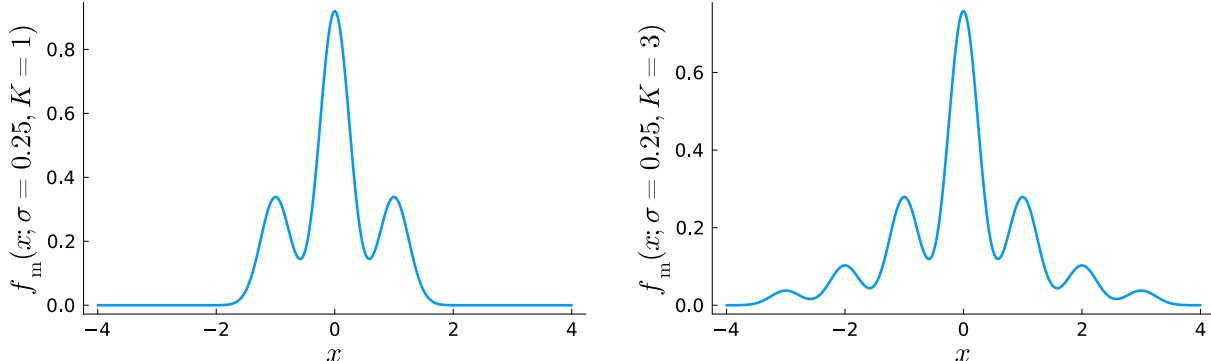

*Figure 3.* The probability density function (1) of the multi-Gaussian mixture distribution for $\varepsilon = 1$ and $\sigma = 0.25$ where $K$ is set to 1 (left) and 3 (right).

The integral can be written as

$$
\begin{aligned}
&\int_{-\infty}^{\infty} |x| \exp\left(-\frac{(x - k\Delta)^2}{2\sigma^2}\right) \mathrm{d}x \\
=&\int_{-\infty}^{\infty} |y + k\Delta| \exp\left(-\frac{y^2}{2\sigma^2}\right) \mathrm{d}y \\
=&-\int_{-\infty}^{-k\Delta} (y + k\Delta) \exp\left(-\frac{y^2}{2\sigma^2}\right) \mathrm{d}y + \int_{-k\Delta}^{\infty} (y + k\Delta) \exp\left(-\frac{y^2}{2\sigma^2}\right) \mathrm{d}y \\
=&-\int_{-\infty}^{-k\Delta} y \exp\left(-\frac{y^2}{2\sigma^2}\right) \mathrm{d}y - k\Delta \int_{-\infty}^{-k\Delta} \exp\left(-\frac{y^2}{2\sigma^2}\right) \mathrm{d}y \\
&+ \int_{-k\Delta}^{\infty} y \exp\left(-\frac{y^2}{2\sigma^2}\right) \mathrm{d}y + k\Delta \int_{-k\Delta}^{\infty} \exp\left(-\frac{y^2}{2\sigma^2}\right) \mathrm{d}y \\
=&-\int_{-\infty}^{-k\Delta} y \exp\left(-\frac{y^2}{2\sigma^2}\right) \mathrm{d}y + \int_{-k\Delta}^{\infty} y \exp\left(-\frac{y^2}{2\sigma^2}\right) \mathrm{d}y + k\Delta\sqrt{2\pi}\sigma\left(\Phi\left(\frac{k\Delta}{\sigma}\right) - \Phi\left(-\frac{k\Delta}{\sigma}\right)\right) \\
=&\int_{-\infty}^{\infty} y \exp\left(-\frac{y^2}{2\sigma^2}\right) \mathrm{d}y - 2\int_{-\infty}^{-k\Delta} y \exp\left(-\frac{y^2}{2\sigma^2}\right) \mathrm{d}y + k\Delta\sqrt{2\pi}\sigma\left(\Phi\left(\frac{k\Delta}{\sigma}\right) - \Phi\left(-\frac{k\Delta}{\sigma}\right)\right) \\
=&-2\int_{-\infty}^{-k\Delta} y \exp\left(-\frac{y^2}{2\sigma^2}\right) \mathrm{d}y + k\Delta\sqrt{2\pi}\sigma\left(\Phi\left(\frac{k\Delta}{\sigma}\right) - \Phi\left(-\frac{k\Delta}{\sigma}\right)\right) \\
=&-2\left[-\sigma^2 \exp\left(-\frac{y^2}{2\sigma^2}\right)\right]_{y=-\infty}^{-k\Delta} + k\Delta\sqrt{2\pi}\sigma\left(\Phi\left(\frac{k\Delta}{\sigma}\right) - \Phi\left(-\frac{k\Delta}{\sigma}\right)\right) \\
=&2\sigma^2 \exp\left(-\frac{k^2\Delta^2}{2\sigma^2}\right) + k\Delta\sqrt{2\pi}\sigma\left(1 - 2\Phi\left(-\frac{k\Delta}{\sigma}\right)\right).
\end{aligned}
$$

Here, the first equality is due to variable change, the second equality partitions the integral region into two regions, the third equality exploits the linearity of integrals, the fourth equality replaces the Gaussian density integrals with their closed form representation, the fifth equality breaks down its second integral (from $-k\Delta$ to $\infty$) as the subtraction of two integrals (from $-\infty$ to $\infty$ and from $-\infty$ to $-k\Delta$), the sixth equality notes that the first integral is the expected value of a 0-mean Gaussian, and the following two equalities compute the definite integral. Hence, we have

$$
\mathbb{E}_{\tilde{X}}[|x|] = \frac{1}{c_K} \sum_{k=-K}^{K} e^{-|k|\varepsilon} \left[2\sigma^2 \exp\left(-\frac{k^2\Delta^2}{2\sigma^2}\right) + k\Delta\sqrt{2\pi}\sigma\left(1 - 2\Phi\left(-\frac{k\Delta}{\sigma}\right)\right)\right].
$$

For the *noise power* of the random variable $\tilde{X}$ with density $f_{\mathrm{m}}(\cdot; \sigma, K)$, we similarly have:

$$\mathbb{E}_{\tilde{X}}[x^2] = \frac{1}{c_K} \sum_{k=-K}^{K} e^{-|k|\varepsilon} \int_{-\infty}^{\infty} x^2 \phi(x; k\Delta, \sigma) \mathrm{d}x.$$

The integral can be written as

$$\int_{-\infty}^{\infty} x^2 \exp\left(-\frac{(x-k\Delta)^2}{2\sigma^2}\right) \mathrm{d}x$$

$$= \int_{-\infty}^{\infty} (y+k\Delta)^2 \exp\left(-\frac{y^2}{2\sigma^2}\right) \mathrm{d}y$$

$$= \underbrace{\int_{-\infty}^{\infty} y^2 \exp\left(-\frac{y^2}{2\sigma^2}\right) \mathrm{d}y}_{=\sqrt{2\pi}\sigma^3} + k^2\Delta^2 \underbrace{\int_{-\infty}^{\infty} \exp\left(-\frac{y^2}{2\sigma^2}\right) \mathrm{d}y}_{=\sqrt{2\pi}\sigma} + 2k\Delta \underbrace{\int_{-\infty}^{\infty} y \exp\left(-\frac{y^2}{2\sigma^2}\right) \mathrm{d}y}_{=0}$$

$$= \sqrt{2\pi}\sigma^3 + k^2\Delta^2\sqrt{2\pi}\sigma$$

where the first equality is due to variable change, the second equality expands the squared term, and the final equality uses the fact that $\exp(-y^2/2\sigma^2)$ is the (unnormalized) density function of a Gaussian distribution with $0$ mean and $\sigma^2$ variance. This concludes that:

$$\mathbb{E}_{\tilde{X}}[x^2] = \frac{1}{c_K} \sum_{k=-K}^{K} e^{-|k|\varepsilon}(\sqrt{2\pi}\sigma^3 + k^2\Delta^2\sqrt{2\pi}\sigma)$$

$$= \frac{1}{\sum_{k=-K}^{K} e^{-|k|\varepsilon}} \sum_{k=-K}^{K} e^{-|k|\varepsilon}(\sigma^2 + k^2\Delta^2).$$

### B.3. Intuition for the Multi-Gaussian Mixture Mechanism

We give an intuition behind the construction in (1), particularly regarding why the mixture weights are chosen to be proportional to $e^{-|k|\varepsilon}$ and why the Gaussian components are centered at $k\Delta$, can be given for the case $K = 1$. To this end, consider a mixture of three Gaussians with common variance $\sigma^2$, means $-a, 0, a$ for some $a \in \mathbb{R}_+$, and symmetric nonnegative mixture weights $p_\varepsilon, q_\varepsilon, p_\varepsilon$, respectively. Equivalently, the noise can be decomposed as

$$\tilde{X} = \tilde{Z} + \tilde{G},$$

where $\tilde{G} \sim \mathcal{N}(0, \sigma^2)$ is a Gaussian random variable and $\tilde{Z}$ is an independent discrete random variable supported on $\{-a, 0, a\}$ with

$$\mathbb{P}[\tilde{Z} = 0] = q_\varepsilon, \qquad \mathbb{P}[\tilde{Z} = \pm a] = p_\varepsilon.$$

Thus, the mixture can be viewed as a Gaussian smoothing of a three-level discrete mechanism.

To see why the spacing $a = \Delta$ and the weight ratio $p_\varepsilon/q_\varepsilon = e^{-\varepsilon}$ are natural (as suggested by our construction (1)), first ignore the Gaussian smoothing and consider the discrete mechanism $q^{\mathrm{disc}}(D) + \tilde{Z}$, where $q^{\mathrm{disc}}$ is a discrete query $q^{\mathrm{disc}} : \mathcal{D} \mapsto \{-\Delta, 0, \Delta\}$ with sensitivity $\Delta$. To examine the DP guarantees for the discrete mechanism $q^{\mathrm{disc}}(D) + \tilde{Z}$, consider neighboring databases $(D, D') \in \mathcal{N}$ with $q^{\mathrm{disc}}(D) = 0$ and $q^{\mathrm{disc}}(D') = \Delta$. The supports of $q^{\mathrm{disc}}(D) + \tilde{Z}$ and $q^{\mathrm{disc}}(D') + \tilde{Z}$ are $\{-a, 0, a\}$ and $\{\Delta - a, \Delta, \Delta + a\}$, respectively. If $a \notin \{\Delta/2, \Delta\}$, these supports have no overlap, and the DP constraints are infeasible unless $\delta \geq 1$, which coincides with the no-privacy case. If $a = \Delta/2$, the event $\{-\Delta/2, 0\}$ has probability $1 - p_\varepsilon$ under $D$ and probability zero under $D'$, forcing $\delta \geq 1 - p_\varepsilon$. By contrast, when $a = \Delta$, the two shifted supports overlap at the two points $0$ and $\Delta$, and only one endpoint is left unmatched in each direction. For the comparison from $D$ to $D'$, this unmatched output is $-\Delta$, which has probability $p_\varepsilon$; hence, setting $\delta = p_\varepsilon$ accounts for this mass and leaves only the multiplicative DP constraints on the overlapping support points. This shows that $a = \Delta$ gives the tight spacing in this three-level construction.

Now that we fixed $a = \Delta$ as the tight spacing, we investigate how to set the weights $p_\varepsilon$ and $q_\varepsilon$. The event $A_1 = \{-\Delta\}$ forces $\delta \geq p_\varepsilon$ in the DP constraint. Letting the smallest possible value $\delta = p_\varepsilon$, the event $A_2 = \{-\Delta, 0\}$ implies

$$p_\varepsilon + q_\varepsilon \;\leq\; e^\varepsilon p_\varepsilon + \delta \;=\; e^\varepsilon p_\varepsilon + p_\varepsilon,$$

which implies $q_\varepsilon \leq e^\varepsilon p_\varepsilon$. Choosing the weights so that this inequality is tight yields

$$p_\varepsilon = q_\varepsilon e^{-\varepsilon}.$$

Therefore, each Gaussian component centered at $\pm\Delta$ receives an $e^{-\varepsilon}$ fraction of the mass assigned to the Gaussian component centered at 0. This is exactly the $K = 1$ instance of the mixture weights in (1), which are proportional to $e^{-|k|\varepsilon}$. In this sense, the three-level argument gives intuition for both the $k\Delta$ spacing and the exponential decay of the mixture weights.

Finally, adding the Gaussian term $\tilde{G}$ turns this discrete construction into a continuous additive noise mechanism. This also explains why multimodality can help: a single broad Gaussian centered at zero smooths out the useful peak structure and allocates mass to intermediate regions, whereas the mixture places separate Gaussian components around the preferred points $k\Delta$, paying only the local smoothing cost around each point.

### B.4. Proof of Theorem 3.2

We first state and prove two intermediary results that will be used in the proof of Theorem 3.2.

**Lemma B.1.** *An additive noise with density $f(x)$ satisfies $(\varepsilon, \delta)$-DP if and only if it satisfies*

$$\int_{x \in A} (e^\varepsilon f(x + \varphi) - f(x)) \mathrm{d}x \geq -\delta \qquad \forall \varphi \in [0, \Delta], \ \forall A \in \mathcal{F},$$

*or, alternatively, if and only if it satisfies*

$$\int_{x \in A} (e^\varepsilon f(x) - f(x + \varphi)) \mathrm{d}x \geq -\delta \qquad \forall \varphi \in [0, \Delta], \ \forall A \in \mathcal{F}.$$

*Proof.* An additive noise with density $f(x)$ achieves $(\varepsilon, \delta)$-DP if and only if it satisfies

$$\int_{x \in A} (e^\varepsilon f(x + \varphi) - f(x)) \mathrm{d}x \geq -\delta \qquad \forall \varphi \in [-\Delta, \Delta], \ \forall A \in \mathcal{F}$$

as per the definition. We can replace $\varphi \in [-\Delta, \Delta]$ with $\varphi \in [0, \Delta]$ without loss of generality since:

$$
\begin{aligned}
&\int_{x \in A} (e^\varepsilon f(x + \varphi) - f(x)) \mathrm{d}x \geq -\delta && \forall \varphi \in [0, \Delta], \ \forall A \in \mathcal{F} \\
\iff &\int_{x \in A} (e^\varepsilon f(-x + \varphi) - f(-x)) \mathrm{d}x \geq -\delta && \forall \varphi \in [0, \Delta], \ \forall A \in \mathcal{F} \\
\iff &\int_{x \in A} (e^\varepsilon f(x - \varphi) - f(x)) \mathrm{d}x \geq -\delta && \forall \varphi \in [0, \Delta], \ \forall A \in \mathcal{F} \\
\iff &\int_{x \in A} (e^\varepsilon f(x + \varphi) - f(x)) \mathrm{d}x \geq -\delta && \forall \varphi \in [-\Delta, 0], \ \forall A \in \mathcal{F},
\end{aligned}
$$

where the first equivalence follows from substituting $x = -x$ as well as from the fact that $\{A \mid A \in \mathcal{F}\} = \{-A \mid A \in \mathcal{F}\}$, the second equivalence uses the symmetry of $f$, and the final equivalence substitutes $\varphi = -\varphi$. We thus obtained the first representation of the DP constraints presented in the statement of this Lemma. Applying analogous steps to the DP definition

$$\int_{x \in A} (e^\varepsilon f(x) - f(x + \varphi)) \mathrm{d}x \geq -\delta \qquad \forall \varphi \in [0, \Delta], \ \forall A \in \mathcal{F}$$

concludes the proof. $\qquad\square$

**Lemma B.2.** *For a multi-Gaussian mixture distribution with parameters $\varphi, \Delta, \sigma > 0$ and $K \in \mathbb{N}$, and for any $0 \leq \varphi_1 \leq \varphi_2 \leq \Delta$, we have:*

$$\int_{-\infty}^{\infty} |f_{\mathrm{m}}(x + \varphi_1; \sigma, K) - f_{\mathrm{m}}(x + \varphi_2; \sigma, K)| \mathrm{d}x \leq \frac{\sqrt{2/\pi}}{\sigma}(\varphi_2 - \varphi_1).$$

*Proof.* Since $\sigma > 0$ and $K \in \mathbb{N}$ are fixed, we denote by $f'_{\mathrm{m}}(x; \sigma, K)$ the derivative of $f_{\mathrm{m}}$ in $x$. By using this notation, the left-hand side of the inequality in the statement can be upper bounded by

$$
\begin{aligned}
\int_{-\infty}^{\infty} |f_{\mathrm{m}}(x + \varphi_1; \sigma, K) - f_{\mathrm{m}}(x + \varphi_2; \sigma, K)| \mathrm{d}x &= \int_{-\infty}^{\infty} \left| \int_{\varphi_1}^{\varphi_2} f'_{\mathrm{m}}(x + \varphi; \sigma, K) \mathrm{d}\varphi \right| \mathrm{d}x \\
&\leq \int_{-\infty}^{\infty} \int_{\varphi_1}^{\varphi_2} |f'_{\mathrm{m}}(x + \varphi; \sigma, K)| \, \mathrm{d}\varphi \, \mathrm{d}x \\
&= \int_{\varphi_1}^{\varphi_2} \underbrace{\int_{-\infty}^{\infty} |f'_{\mathrm{m}}(x + \varphi; \sigma, K)| \, \mathrm{d}x}_{(i)} \, \mathrm{d}\varphi,
\end{aligned}
$$

where the first equality follows from the fundamental theorem of calculus, the inequality follows from the triangle inequality of the absolute value, and the final equality changes the order of integrals via Fubini's theorem. Finally, term $(i)$ satisfies

$$
\begin{aligned}
(i) &= \int_{-\infty}^{\infty} |f'_{\mathrm{m}}(x; \sigma, K)| \, \mathrm{d}x \\
&\leq \frac{1}{c_K} \sum_{k=-K}^{K} e^{-|k|\varepsilon} \int_{-\infty}^{\infty} \left| \frac{\partial}{\partial x} \exp\left( -\frac{(x - k\Delta)^2}{2\sigma^2} \right) \right| \mathrm{d}x \\
&= \frac{1}{c_K} \sum_{k=-K}^{K} e^{-|k|\varepsilon} \int_{-\infty}^{\infty} \frac{|x - k\Delta|}{\sigma^2} \exp\left( -\frac{(x - k\Delta)^2}{2\sigma^2} \right) \mathrm{d}x \\
&= \frac{1}{c_K} \sum_{k=-K}^{K} e^{-|k|\varepsilon} \int_{-\infty}^{\infty} \frac{|y|}{\sigma^2} \exp\left( -\frac{y^2}{2\sigma^2} \right) \mathrm{d}y \\
&= \frac{2}{\sqrt{2\pi}\sigma \sum_{k=-K}^{K} e^{-|k|\varepsilon}} \sum_{k=-K}^{K} e^{-|k|\varepsilon} = \frac{\sqrt{2/\pi}}{\sigma},
\end{aligned}
$$

where the first equality follows from variable change $x = x + \varphi$, the inequality uses the definition of $f_{\mathrm{m}}(x; \sigma, K)$ and exploits the triangle inequality of the absolute value, the second equality explicitly writes the derivative term, the third equality follows from variable change $y = x - k\Delta$, the fourth equality explicitly writes the expression for $c_K$ as well as notes that the integral computes the expected absolute value of a random variable that follows a Gaussian distribution with $0$ mean and $\sigma^2$ variance (*cf.* term $(i)$ of Appendix C.4 for full derivation), and the final equality cancels common terms. Using this upper bound for $(i)$ back allows us to conclude

$$\int_{\varphi_1}^{\varphi_2} \int_{-\infty}^{\infty} |f'_{\mathrm{m}}(x + \varphi; \sigma, K)| \, \mathrm{d}x \, \mathrm{d}\varphi \leq \int_{\varphi_1}^{\varphi_2} \frac{\sqrt{2/\pi}}{\sigma} \mathrm{d}\varphi = \frac{\sqrt{2/\pi}}{\sigma}(\varphi_2 - \varphi_1)$$

which coincides with the right-hand side presented in the lemma. $\square$

We can now prove Theorem 3.2.
**Proof of Theorem 3.2.** Given $\eta \in (0, 1)$ fix any $\beta \leq \sqrt{2\pi}\eta\sigma\delta$, and assume the condition in the statement of this theorem holds:

$$\int_{-\infty}^{\infty} \min\{e^\varepsilon f_{\mathrm{m}}(x; \sigma, K) - f_{\mathrm{m}}(x + \varphi; \sigma, K), 0\} \mathrm{d}x + (1 - \eta)\delta \geq 0 \quad \forall \varphi \in \{0, \beta, 2\beta, \ldots, \Delta\}. \tag{6}$$

To show the desired result, we should show that this condition guarantees the following definition of $(\varepsilon, \delta)$-DP (*cf.* Lemma B.1):

$$\int_{x \in A} (e^\varepsilon f_\mathrm{m}(x; \sigma, K) - f_\mathrm{m}(x + \varphi; \sigma, K)) \mathrm{d}x \geq -\delta \qquad \forall \varphi \in [0, \Delta], \ \forall A \in \mathcal{F}$$

$$\iff \inf_{A \in \mathcal{F}} \left\{ \int_{x \in A} (e^\varepsilon f_\mathrm{m}(x; \sigma, K) - f_\mathrm{m}(x + \varphi; \sigma, K)) \mathrm{d}x \right\} \geq -\delta \qquad \forall \varphi \in [0, \Delta]$$

$$\iff \int_{-\infty}^\infty \min\{e^\varepsilon f_\mathrm{m}(x; \sigma, K) - f_\mathrm{m}(x + \varphi; \sigma, K), 0\} \mathrm{d}x \geq -\delta \qquad \forall \varphi \in [0, \Delta]. \qquad (7)$$

Here, the first equivalence follows from taking the infimum of the left hand side over $A \in \mathcal{F}$, and the second equivalence follows since the worst-case event $A$ does not include points $x$ with $e^\varepsilon f_\mathrm{m}(x; \sigma, K) - f_\mathrm{m}(x + \varphi; \sigma, K)) > 0$. We will show (6) $\implies$ (7).

Fix an arbitrary $\varphi \in [0, \Delta]$, let $k' \in \mathbb{N}$ be the index that satisfies $|\varphi - k'\beta| \leq \beta/2$, and note that the left-hand side of (7) satisfies:

$$\int_{-\infty}^\infty \min\{(e^\varepsilon f_\mathrm{m}(x; \sigma, K) - f_\mathrm{m}(x + \varphi; \sigma, K)), 0\} \mathrm{d}x$$

$$\geq \int_{-\infty}^\infty \min\{f_\mathrm{m}(x; \sigma, K) - f_\mathrm{m}(x + \varphi; \sigma, K), 0\} \mathrm{d}x$$

$$= \int_{-\infty}^\infty \min\{(f_\mathrm{m}(x; \sigma, K) - f_\mathrm{m}(x + k'\beta; \sigma, K)) - (f_\mathrm{m}(x + \varphi; \sigma, K) - f_\mathrm{m}(x + k'\beta; \sigma, K)), 0\} \mathrm{d}x$$

$$\geq \int_{-\infty}^\infty \min\{f_\mathrm{m}(x; \sigma, K) - f_\mathrm{m}(x + k'\beta; \sigma, K), 0\} - |f_\mathrm{m}(x + \varphi; \sigma, K) - f_\mathrm{m}(x + k'\beta; \sigma, K)| \mathrm{d}x$$

$$= \underbrace{\int_{-\infty}^\infty \min\{f_\mathrm{m}(x; \sigma, K) - f_\mathrm{m}(x + k'\beta; \sigma, K), 0\} \mathrm{d}x}_{(i)}$$

$$\underbrace{- \int_{-\infty}^\infty |f_\mathrm{m}(x + \varphi; \sigma, K) - f_\mathrm{m}(x + k'\beta; \sigma, K)| \mathrm{d}x}_{(ii)} \, .$$

Here, the first inequality follows from $e^\varepsilon \geq 1$, the equality that follows adds and subtracts the common term $f_\mathrm{m}(x + k'\beta; \sigma, K)$ in the $\min$-term, the second inequality is due to the fact that for any $a, b \in \mathbb{R}$ we have $\min\{a - b, 0\} \geq \min\{a, 0\} - |b|$, and the final equality exploits the linearity of integrals. To conclude this proof, we will now show that $(i) - (ii) \geq -\delta$.

The fact that $\varphi \in [0, \Delta]$ holds implies that $k'\beta \in \{0, \beta, 2\beta, \ldots, \Delta\}$, hence from (6) we have:

$$(i) \geq -(1 - \eta)\delta.$$

Moreover, term $(ii)$ can be upper bounded by

$$(ii) \leq \frac{\sqrt{2/\pi}}{\sigma} |\varphi - k'\beta| \leq \frac{1}{\sqrt{2\pi}\sigma} \beta \leq \eta\delta,$$

where the first inequality follows from Lemma B.2, the second inequality is due to the definition of $k' \in \mathbb{N}$, and the final inequality is due to the definition of $\beta$.

Putting these terms together finally concludes the proof since $(i) - (ii) \geq -(1 - \eta)\delta - \eta\delta = -\delta$. $\qquad \square$

### B.5. Proof of Proposition 3.3

Throughout this proof, we fix an arbitrary query with sensitivity $\Delta > 0$. The proof of Proposition 3.3 relies on three auxiliary lemmas which we prove first.

**Lemma B.3.** *For any $\delta \in (0, 1/2)$ and as $\varepsilon \to \infty$, the standard deviation $\sigma^{\mathrm{AG}}(\varepsilon, \delta)$ of the analytic Gaussian mechanism satisfies*

$$\sigma^{\mathrm{AG}}(\varepsilon, \delta) = \frac{\Delta}{\sqrt{2\varepsilon}} + \frac{\Delta \cdot |\Phi^{-1}(\delta)|}{2\varepsilon} + \mathcal{O}(\varepsilon^{-3/2}),$$

*which implies that the expected $l_2$-loss of the analytic Gaussian mechanism satisfies*

$$L_2^{\mathrm{AG}}(\varepsilon, \delta) = [\sigma^{\mathrm{AG}}(\varepsilon, \delta)]^2 = \frac{\Delta^2}{2\varepsilon} + \frac{|\Phi^{-1}(\delta)| \cdot \Delta^2}{\sqrt{2}\varepsilon^{3/2}} + \mathcal{O}(\varepsilon^{-2}).$$

*Proof.* According to Balle & Wang (2018), $\sigma^{\mathrm{AG}}(\varepsilon, \delta)$ is the unique positive solution $\sigma$ to

$$\delta = \Phi\left(\frac{\Delta}{2\sigma} - \frac{\varepsilon\sigma}{\Delta}\right) - e^\varepsilon \Phi\left(-\frac{\Delta}{2\sigma} - \frac{\varepsilon\sigma}{\Delta}\right), \tag{8}$$

where $\Phi$ is the standard normal cdf. For simplicity of notation, we let

$$r := \frac{\sigma}{\Delta}, \qquad a := \frac{1}{2r} - \varepsilon r, \qquad b := -\frac{1}{2r} - \varepsilon r. \tag{9}$$

Then (8) can be written as

$$\delta = \Phi(a) - e^\varepsilon \Phi(b) = \Phi(a) - e^\varepsilon \Phi\left(-\sqrt{a^2 + 2\varepsilon}\right), \tag{10}$$

where the second equality exploits the identity $b = -\sqrt{a^2 + 2\varepsilon}$ which holds since we have

$$(-b)^2 - a^2 = \left(\frac{1}{2r} + \varepsilon r\right)^2 - \left(\frac{1}{2r} - \varepsilon r\right)^2 = 2\varepsilon.$$

Our proof unfolds as follows: since $r = \sigma/\Delta$ for $\sigma$ solving the desired equation (8), we aim to derive an asymptotic expression for $r$, and subsequently multiply it by $\Delta$ to recover $\sigma$. However, since $r$ also defines $a$ in (9), we will start by deriving an asymptotic expression for $a$. To this end, consider (10) and observe that the final term on the right-hand side can be bounded as

$$e^\varepsilon \Phi\left(-\sqrt{a^2 + 2\varepsilon}\right) \leq \frac{e^\varepsilon}{\sqrt{2\pi}} \frac{\exp\left(-\dfrac{a^2 + 2\varepsilon}{2}\right)}{\sqrt{a^2 + 2\varepsilon}} = \frac{1}{\sqrt{2\pi}} \frac{\exp\left(-\dfrac{a^2}{2}\right)}{\sqrt{a^2 + 2\varepsilon}} \leq \frac{1}{\sqrt{2\pi}} \frac{1}{\sqrt{2\varepsilon}} = \frac{1}{2\sqrt{\pi\varepsilon}},$$

where the first inequality is due to Mills' inequality (*i.e.*, bounding Gaussian tail probability via its pdf), the equality that follows cancels the $\pm\varepsilon$ terms in the exponentials, the second inequality uses the fact that $\exp(-a^2/2) \leq 1$ and $\sqrt{a^2 + 2\varepsilon} \geq \sqrt{2\varepsilon}$, and the final equality simplifies the expression. Plugging this in (10) yields

$$\delta \geq \Phi(a) - \frac{1}{2\sqrt{\pi\varepsilon}} \implies \Phi(a) = \delta + \mathcal{O}(\varepsilon^{-1/2}) \iff a = \Phi^{-1}(\delta + \mathcal{O}(\varepsilon^{-1/2})).$$

Now that we have representation $a = \Phi^{-1}(\delta + \mathcal{O}(\varepsilon^{-1/2}))$, we can apply a first-order Taylor expansion for $a$ around $\delta$ and obtain

$$a = \Phi^{-1}(\delta) + (\Phi^{-1})'(\delta) \cdot \mathcal{O}(\varepsilon^{-1/2}) + \mathcal{O}([\varepsilon^{-1/2}]^2) = \Phi^{-1}(\delta) + \mathcal{O}(\varepsilon^{-1/2}),$$

where the final equation follows from the fact that $(\Phi^{-1})'(\delta)$ is constant[3] for fixed $\delta$, and also from the fact that $\varepsilon^{-1/2}$ dominates $[\varepsilon^{-1/2}]^2 = \varepsilon^{-1}$. If we then let $q := \Phi^{-1}(\delta)$, then the previous derivation can be represented as

$$a = q + \mathcal{O}(\varepsilon^{-1/2}). \tag{11}$$

---

[3]Note also that $(\Phi^{-1})'$ exists by the inverse function theorem, since $\Phi$ is continuously differentiable and strictly increasing with $\Phi'(\Phi^{-1}(\delta)) > 0$ for any fixed $\delta \in (0, 1)$.

We next derive a closed-form expression for $r$ as

$$
\begin{aligned}
r &= \frac{-a + \sqrt{a^2 + 2\varepsilon}}{2\varepsilon} = \frac{-a + \sqrt{2\varepsilon} + \mathcal{O}(\varepsilon^{-1/2})}{2\varepsilon} = \frac{-q - \mathcal{O}(\varepsilon^{-1/2}) + \sqrt{2\varepsilon} + \mathcal{O}(\varepsilon^{-1/2})}{2\varepsilon} \\
&= \frac{1}{\sqrt{2\varepsilon}} - \frac{q}{2\varepsilon} + \mathcal{O}(\varepsilon^{-3/2}) \\
&= \frac{1}{\sqrt{2\varepsilon}} + \frac{|\Phi^{-1}(\delta)|}{2\varepsilon} + \mathcal{O}(\varepsilon^{-3/2})
\end{aligned}
$$

where the first equality follows from the definition of $a$ in (9) (as a function of $r$) and the fact that $r > 0$, the second equality holds since (11) implies $a = q + \mathcal{O}(\varepsilon^{-1/2}) = \mathcal{O}(1)$, the third equality exploits (11) to substitute $a = q + \mathcal{O}(\varepsilon^{-1/2})$, the fourth equality uses the fact that $\mathcal{O}(\varepsilon^{-1/2}) - \mathcal{O}(\varepsilon^{-1/2}) = \mathcal{O}(\varepsilon^{-1/2})$, and the final equality uses the fact that $\delta \in (0, 1/2)$ implies $q = \Phi^{-1}(\delta) < 0$. We can now use the definition $r = \sigma/\Delta$ to conclude that

$$
\sigma = \frac{\Delta}{\sqrt{2\varepsilon}} + \frac{\Delta \cdot |\Phi^{-1}(\delta)|}{2\varepsilon} + \mathcal{O}(\varepsilon^{-3/2}),
$$

which coincides with the identity in the statement of this lemma. Taking the square of $\sigma$ gives the $l_2$-loss of the analytic Gaussian mechanism as this coincides with the variance as the Gaussian mechanism has zero mean. $\square$

**Lemma B.4.** *Let $g_\mu(x)$ denote the density function of $\mathcal{N}(\mu, \sigma^2)$ for $\sigma = \Delta/\sqrt{2\varepsilon}$. For any $\Delta > 0$ and $\varphi \in [0, \Delta]$, we have*

$$
g_0(x) \le \theta g_{\varphi - \Delta}(x) + (1 - \theta) e^\varepsilon g_\varphi(x) \tag{12}
$$

*as well as*

$$
g_\Delta(x) \le \theta e^{\varepsilon(1-\theta)} g_\varphi(x) + (1 - \theta) g_{\varphi + \Delta}(x), \tag{13}
$$

*where $\theta \coloneqq \varphi/\Delta \in [0, 1]$.*

*Proof.* We first prove inequality (12). We can show that

$$
\begin{aligned}
g_{\varphi - \Delta}(x)^\theta g_\varphi(x)^{1-\theta} &= \left( \frac{1}{\sqrt{2\pi\sigma^2}} \right)^\theta \exp\left( -\theta \cdot \frac{(x - (\varphi - \Delta))^2}{2\sigma^2} \right) \left( \frac{1}{\sqrt{2\pi\sigma^2}} \right)^{1-\theta} \exp\left( -(1 - \theta) \cdot \frac{(x - \varphi)^2}{2\sigma^2} \right) \\
&= \frac{1}{\sqrt{2\pi\sigma^2}} \exp\left( -\frac{x^2}{2\sigma^2} \right) \exp\left( \frac{2x\left(\theta(\varphi - \Delta) + (1 - \theta)\varphi\right) - \theta(\varphi - \Delta)^2 - (1 - \theta)\varphi^2}{2\sigma^2} \right) \\
&= g_0(x) \exp\left( \frac{2x\left(\varphi - \theta\Delta\right) - \theta(\varphi - \Delta)^2 - (1 - \theta)\varphi^2}{2\sigma^2} \right) \\
&= g_0(x) \exp\left( -\frac{\theta(\varphi - \Delta)^2 + (1 - \theta)\varphi^2}{2\sigma^2} \right) \\
&= g_0(x) \exp\left( -\frac{\theta(1 - \theta)^2\Delta^2 + (1 - \theta)\theta^2\Delta^2}{2\sigma^2} \right) \\
&= g_0(x) \exp\left( -\frac{\theta(1 - \theta)\Delta^2}{2\sigma^2} \right) \\
&= g_0(x) \exp\left( -\varepsilon\theta(1 - \theta) \right). \tag{14}
\end{aligned}
$$

Here, the first equality follows from the definition of the Gaussian density. The second equality follows by collecting the normalizing constants and expanding the two quadratic terms in the exponent. The third equality follows from the definition of $g_0(x)$ and the identity $\theta(\varphi - \Delta) + (1 - \theta)\varphi = \varphi - \theta\Delta$. The fourth equality uses the definition $\theta = \varphi/\Delta$, which implies $\varphi - \theta\Delta = 0$. The fifth equality uses $\varphi = \theta\Delta$, while the sixth equality follows by simplifying

$$
\theta(1 - \theta)^2\Delta^2 + (1 - \theta)\theta^2\Delta^2 = \theta(1 - \theta)\Delta^2.
$$

The final equality follows from $\sigma^2 = \Delta^2/(2\varepsilon)$. The equality (14) can be written as

$$
\begin{aligned}
g_0(x) \;=\; g_{\varphi-\Delta}(x)^\theta g_\varphi(x)^{1-\theta} e^{\varepsilon\theta(1-\theta)} \;&=\; e^{-\varepsilon(1-\theta)^2} g_{\varphi-\Delta}(x)^\theta (e^\varepsilon g_\varphi(x))^{1-\theta} \\
&\leq\; g_{\varphi-\Delta}(x)^\theta (e^\varepsilon g_\varphi(x))^{1-\theta} \\
&\leq\; \theta g_{\varphi-\Delta}(x) + (1-\theta) e^\varepsilon g_\varphi(x).
\end{aligned}
$$

where the second equality uses the identity $\varepsilon\theta(1-\theta) = \varepsilon(1-\theta) - \varepsilon(1-\theta)^2$, first inequality holds since $-\varepsilon(1-\theta)^2 \leq 0$ and thus the first $\exp$-term lies in $[0,1]$, and the final inequality applies the weighted arithmetic mean-geometric mean inequality. This concludes the desired inequality (12).

Inequality (13) can be proved similarly. An analogous application of equality (14) yields

$$
g_\Delta(x) \;=\; e^{\varepsilon\theta(1-\theta)} g_\varphi(x)^\theta g_{\varphi+\Delta}(x)^{1-\theta} \;=\; \left(e^{\varepsilon(1-\theta)} g_\varphi(x)\right)^\theta g_{\varphi+\Delta}(x)^{1-\theta},
$$

and an application of the weighted arithmetic mean-geometric mean inequality concludes the desired inequality. $\qquad\square$

**Lemma B.5.** *For any $\delta \in (0, 1/2)$ and $\varepsilon > 0$ satisfying $\varepsilon \geq \log(\delta^{-1} - 2)$, setting $\sigma = \Delta/\sqrt{2\varepsilon}$ for the $K = 1$ multi-Gaussian mixture distribution gives a feasible $(\varepsilon, \delta)$-DP mechanism. Consequently, the $l_2$-loss of the multi-Gaussian mixture mechanism satisfies*

$$
L_2^{\mathrm{MG}}(\varepsilon, \delta) \;\leq\; \frac{\Delta^2}{2\varepsilon} + 2\Delta^2 e^{-\varepsilon}.
$$

*Proof.* If $\sigma = \Delta/\sqrt{2\varepsilon}$ is feasible for $(\varepsilon, \delta)$-DP, then the $l_2$-loss bound follows immediately from the closed-form expression for the noise power derived in Appendix B.2. That is, the $K = 1$ multi-Gaussian mixture mechanism satisfies

$$
L_2^{\mathrm{MG}}(\varepsilon, \delta) = \mathbb{E}[\tilde{X}^2] = \frac{e^{-\varepsilon}(\sigma^2 + \Delta^2) + \sigma^2 + e^{-\varepsilon}(\sigma^2 + \Delta^2)}{1 + 2e^{-\varepsilon}} \;=\; \sigma^2 + \frac{2\Delta^2 e^{-\varepsilon}}{1 + 2e^{-\varepsilon}} \;\leq\; \frac{\Delta^2}{2\varepsilon} + 2\Delta^2 e^{-\varepsilon},
$$

where the last inequality uses $\sigma^2 = \Delta^2/(2\varepsilon)$ since we set $\sigma = \Delta/\sqrt{2\varepsilon}$. Therefore, the remainder of this proof will prove the feasibility of $\sigma = \Delta/\sqrt{2\varepsilon}$ for the $K = 1$ multi-Gaussian mixture mechanism. For simplicity of notation, set

$$
\sigma := \frac{\Delta}{\sqrt{2\varepsilon}}, \qquad p_\varepsilon := \frac{1}{e^\varepsilon + 2}, \qquad q_\varepsilon := \frac{e^\varepsilon}{e^\varepsilon + 2} = e^\varepsilon p_\varepsilon. \tag{15}
$$

We will study the DP condition by directly comparing the output densities associated with any two neighboring databases. To this end, consider any neighboring databases $(D, D') \in \mathcal{N}$ with a query difference $\varphi \in [0, \Delta]$. After translation, we may assume that the two query values are 0 and $\varphi$. Let $g_\mu$ denote the density of $\mathcal{N}(\mu, \sigma^2)$. Using (1) for $K = 1$, the two output densities (the likelihoods of observing a realization $x$ under these two neighboring databases) are given by

$$
\begin{aligned}
f_\varphi(x) &= p_\varepsilon g_{-\Delta}(x) + q_\varepsilon g_0(x) + p_\varepsilon g_\Delta(x), \\
h_\varphi(x) &= p_\varepsilon g_{\varphi-\Delta}(x) + q_\varepsilon g_\varphi(x) + p_\varepsilon g_{\varphi+\Delta}(x).
\end{aligned}
$$

We will prove feasibility of the DP constraint by establishing the following stronger pointwise condition: the density $f_\varphi$ is dominated by $e^\varepsilon h_\varphi$ up to a residual Gaussian component. In particular, we will show that

$$
f_\varphi(x) \;\leq\; e^\varepsilon h_\varphi(x) + p_\varepsilon g_{-\Delta}(x) \qquad \forall x \in \mathbb{R}. \tag{16}
$$

To see why this is sufficient, let $A \subseteq \mathbb{R}$ be any measurable set. Integrating (16) over $A$ gives

$$
\mathbb{P}[q(D) + \tilde{X} \in A] \;=\; \int_A f_\varphi(x)\mathrm{d}x \;\leq\; e^\varepsilon \int_A h_\varphi(x)\mathrm{d}x + p_\varepsilon \int_A g_{-\Delta}(x)\mathrm{d}x \;\leq\; e^\varepsilon \mathbb{P}[q(D') + \tilde{X} \in A] + p_\varepsilon.
$$

Hence, once $p_\varepsilon \leq \delta$ (which translates as $\varepsilon \geq \log(\delta^{-1} - 2)$, coinciding with the condition in the statement of the lemma), the desired DP inequality follows for the comparison from $D$ to $D'$. The reverse comparison follows by interchanging the roles of $D$ and $D'$. It therefore remains to prove the pointwise bound (16).

To prove (16), we multiply inequalities (12) and (13) of Lemma B.4 by $q_\varepsilon$ and $p_\varepsilon$, respectively, and sum them to obtain:

$$q_\varepsilon g_0(x) + p_\varepsilon g_\Delta(x) \leq \theta q_\varepsilon g_{\varphi-\Delta}(x) + \underbrace{\left[(1-\theta)e^\varepsilon q_\varepsilon + \theta e^{\varepsilon(1-\theta)}p_\varepsilon\right]}_{(*)} g_\varphi(x) + (1-\theta)p_\varepsilon g_{\varphi+\Delta}(x)$$

$$\leq \theta q_\varepsilon g_{\varphi-\Delta}(x) + e^\varepsilon q_\varepsilon g_\varphi(x) + (1-\theta)p_\varepsilon g_{\varphi+\Delta}(x)$$
$$\leq q_\varepsilon g_{\varphi-\Delta}(x) + e^\varepsilon q_\varepsilon g_\varphi(x) + q_\varepsilon g_{\varphi+\Delta}(x)$$
$$= e^\varepsilon p_\varepsilon g_{\varphi-\Delta}(x) + e^\varepsilon q_\varepsilon g_\varphi(x) + e^\varepsilon p_\varepsilon g_{\varphi+\Delta}(x)$$
$$= e^\varepsilon h_\varphi(x). \tag{17}$$

Here, the second inequality follows since the term $(*)$ is upper bounded by $(1-\theta)e^\varepsilon q_\varepsilon + \theta e^\varepsilon q_\varepsilon = e^\varepsilon q_\varepsilon$, as a result of $p_\varepsilon \leq q_\varepsilon$ and $e^{\varepsilon(1-\theta)} \leq e^\varepsilon$. The third inequality follows since $\theta q_\varepsilon \leq q_\varepsilon$ and $(1-\theta)p_\varepsilon \leq p_\varepsilon \leq q_\varepsilon$. The first equality follows from $q_\varepsilon = e^\varepsilon p_\varepsilon$, and the final equality recognizes the definition of $h_\varphi(x)$.

The desired inequality now can be concluded as

$$f_\varphi(x) = p_\varepsilon g_{-\Delta}(x) + q_\varepsilon g_0(x) + p_\varepsilon g_\Delta(x) \leq p_\varepsilon g_{-\Delta}(x) + e^\varepsilon h_\varphi(x),$$

where the first equality is due to the definition of $f_\varphi(x)$ and the inequality is due to (17). □

We are now ready to prove Proposition 3.3.
**Proof of Proposition 3.3.** Auxiliary Lemma B.3 yields

$$L_2^{\mathrm{AG}}(\varepsilon, \delta) = \frac{\Delta^2}{2\varepsilon} + c_\delta \frac{\Delta^2}{\varepsilon^{3/2}} + \mathcal{O}(\varepsilon^{-2}),$$

where for simplicity of presentation we set

$$c_\delta := \frac{|\Phi^{-1}(\delta)|}{\sqrt{2}} > 0.$$

The remainder term $\mathcal{O}(\varepsilon^{-2})$ is asymptotically smaller than the positive correction term $\varepsilon^{-3/2}$. In particular, since $\varepsilon^{-2} = o(\varepsilon^{-3/2})$, there exists $\varepsilon_1(\delta) > 0$ such that, for all $\varepsilon \geq \varepsilon_1(\delta)$,

$$\mathcal{O}(\varepsilon^{-2}) \geq -\frac{c_\delta}{2}\frac{\Delta^2}{\varepsilon^{3/2}}.$$

Therefore, for all $\varepsilon \geq \varepsilon_1(\delta)$, we obtain

$$L_2^{\mathrm{AG}}(\varepsilon, \delta) \geq \frac{\Delta^2}{2\varepsilon} + \frac{c_\delta}{2}\frac{\Delta^2}{\varepsilon^{3/2}}.$$

Moreover, the third auxiliary Lemma B.5 shows that, for all $\varepsilon > 0$ satisfying $\varepsilon \geq \log(\delta^{-1} - 2)$, we have

$$L_2^{\mathrm{MG}}(\varepsilon, \delta) \leq \frac{\Delta^2}{2\varepsilon} + 2\Delta^2 e^{-\varepsilon}.$$

Since $e^{-\varepsilon} = o(\varepsilon^{-3/2})$, there exists $\varepsilon_2(\delta) > 0$ such that, for all $\varepsilon \geq \varepsilon_2(\delta)$,

$$2e^{-\varepsilon} < \frac{c_\delta}{2}\varepsilon^{-3/2}.$$

Now if we define

$$\varepsilon_0 := \max\left\{\varepsilon_1(\delta), \ \varepsilon_2(\delta), \ \log(\delta^{-1} - 2)\right\},$$

then the above derivations conclude that for all $\varepsilon \geq \varepsilon_0$, we obtain

$$L_2^{\mathrm{MG}}(\varepsilon, \delta) \leq \frac{\Delta^2}{2\varepsilon} + 2\Delta^2 e^{-\varepsilon} < \frac{\Delta^2}{2\varepsilon} + \frac{c_\delta}{2}\frac{\Delta^2}{\varepsilon^{3/2}} \leq L_2^{\mathrm{AG}}(\varepsilon, \delta).$$

This proves the desired strict inequality.

## B.6. Proof of Lemma 3.4

Throughout this proof, we will use the representation

$$f_{\mathrm{m}}(x; \sigma, K) = \frac{1}{\sum_{k=-K}^{K} e^{-|k|\varepsilon}} \sum_{k=-K}^{K} \mathcal{N}(x; k\Delta, \sigma),$$

where

$$\mathcal{N}(x; \mu, \sigma) := \frac{1}{\sqrt{2\pi}\sigma} \exp\left(-\frac{(x-\mu)^2}{2\sigma^2}\right)$$

denotes the normalized Gaussian density function. Using such notation, we first provide an intermediary result.

**Lemma B.6.** *For fixed $\varepsilon > 0$, $\delta \in (0, 1)$, $\Delta > 0$, $K \in \mathbb{N}$, and $\varphi \in [0, \Delta]$, define*

$$\ell(x; \sigma) := e^{\varepsilon} f_{\mathrm{m}}(x; \sigma, K) - f_{\mathrm{m}}(x + \varphi; \sigma, K)$$

*as the integrand of the left-hand side of the constraint (2) for any $\sigma > 0$. For all $\sigma' \geq \sigma > 0$, we have*

$$\ell(x; \sigma') := (\ell(\cdot; \sigma) \otimes \mathcal{N}(\cdot; 0, \sqrt{\sigma'^2 - \sigma^2}))(x),$$

*where $\otimes$ denotes the convolution operator defined as $(f \otimes g)(x) := \int_{-\infty}^{\infty} f(y)g(x-y)\mathrm{d}y$.*

*Proof.* For $\sigma' \geq \sigma$, if we denote by $\tau = \sqrt{\sigma'^2 - \sigma^2}$, then from (Bromiley, 2003) the following identity holds:

$$\mathcal{N}(x; k\Delta, \sigma') = \mathcal{N}(x; k\Delta, \sigma) \otimes \mathcal{N}(x; 0, \tau).$$

Note that we have a slight breach of notation since with $\mathcal{N}(x; k\Delta, \sigma) \otimes \mathcal{N}(x; 0, \tau)$ we mean $(\mathcal{N}(\cdot; k\Delta, \sigma) \otimes \mathcal{N}(\cdot; 0, \tau))(x)$. We can now show

$$
\begin{aligned}
f_{\mathrm{m}}(x; \sigma', K) &= \frac{1}{\sum_{k=-K}^{K} e^{-|k|\varepsilon}} \sum_{k=-K}^{K} \mathcal{N}(x; k\Delta, \sigma') \\
&= \frac{1}{\sum_{k=-K}^{K} e^{-|k|\varepsilon}} \sum_{k=-K}^{K} \mathcal{N}(x; k\Delta, \sigma) \otimes \mathcal{N}(x; 0, \tau) \\
&= \frac{1}{\sum_{k=-K}^{K} e^{-|k|\varepsilon}} \left[ \left( \sum_{k=-K}^{K} \mathcal{N}(x; k\Delta, \sigma) \right) \otimes \mathcal{N}(x; 0, \tau) \right] \\
&= \left( \frac{1}{\sum_{k=-K}^{K} e^{-|k|\varepsilon}} \sum_{k=-K}^{K} \mathcal{N}(x; k\Delta, \sigma) \right) \otimes \mathcal{N}(x; 0, \tau) \\
&= f_{\mathrm{m}}(x; \sigma, K) \otimes \mathcal{N}(x; 0, \tau),
\end{aligned}
$$

where the first equality is due to the definition of $f_{\mathrm{m}}$, the second equality is due to the convolution property, the third equality is due to the distributivity property of the convolution operator, the fourth equality is due to the associativity property of the convolution operator with scalar multiplication, and the final equality substitutes the definition of $f_{\mathrm{m}}(x; \sigma, K)$. Thanks to this identity, we can derive

$$
\begin{aligned}
\ell(x; \sigma') &= e^{\varepsilon} f_{\mathrm{m}}(x; \sigma', K) - f_{\mathrm{m}}(x + \varphi; \sigma', K) \\
&= e^{\varepsilon} f_{\mathrm{m}}(x; \sigma, K) \otimes \mathcal{N}(x; 0, \tau) - f_{\mathrm{m}}(x + \varphi; \sigma, K) \otimes \mathcal{N}(x; 0, \tau) \\
&= (e^{\varepsilon} f_{\mathrm{m}}(x; \sigma, K) - f_{\mathrm{m}}(x + \varphi; \sigma, K)) \otimes \mathcal{N}(x; 0, \tau) \\
&= \ell(x; \sigma) \otimes \mathcal{N}(x; 0, \tau),
\end{aligned}
$$

which coincides with the statement of this intermediary lemma. $\square$

We now prove Lemma 3.4.

**Proof of Lemma 3.4.** For the given $\eta \in (0,1)$, select any $\beta \leq \sqrt{2\pi}\eta\sigma\delta$ and assume a given $\sigma > 0$ satisfies (2). Such selection of $\beta$ is also valid for $\sigma' \geq \sigma$. Hence, for any fixed $\varphi \in \{0, \beta, 2\beta, \ldots, \Delta\}$, if we denote by $\ell(x;\sigma) := e^\varepsilon f_{\mathrm{m}}(x;\sigma, K) - f_{\mathrm{m}}(x + \varphi; \sigma, K)$, then we should show that

$$\int_{-\infty}^{\infty} \min\{\ell(x;\sigma'), 0\}\mathrm{d}x \geq \int_{-\infty}^{\infty} \min\{\ell(x;\sigma), 0\}\mathrm{d}x \tag{18}$$

holds to conclude the proof. The min-term on the left-hand side of inequality (18) satisfies

$$\min\{\ell(x;\sigma'), 0\} = \min\left\{\int_{-\infty}^{\infty} \ell(x - t;\sigma, K)\mathcal{N}(t;0,\tau)\mathrm{d}t, 0\right\}$$

$$\geq \int_{-\infty}^{\infty} \min\{\ell(x - t;\sigma, K), 0\}\mathcal{N}(t;0,\tau)\mathrm{d}t, \tag{19}$$

where the equality follows from Lemma B.6 as well as the definition of the convolution operator, and the inequality follows from Jensen's inequality applied to the concave function $t \mapsto \min\{t, 0\}$ which is applicable since $\mathcal{N}(t;0,\tau)$ is a density function. Hence, the desired inequality is shown as:

$$\int_{-\infty}^{\infty} \min\{\ell(x;\sigma'), 0\}\mathrm{d}x \geq \int_{-\infty}^{\infty}\int_{-\infty}^{\infty} \min\{\ell(x - t;\sigma, K), 0\}\mathcal{N}(t;0,\tau)\mathrm{d}t\mathrm{d}x$$

$$= \int_{-\infty}^{\infty} \left[\int_{-\infty}^{\infty} \min\{\ell(x - t;\sigma, K), 0\}\mathcal{N}(t;0,\tau)\mathrm{d}x\right]\mathrm{d}t$$

$$= \int_{-\infty}^{\infty} \left[\int_{-\infty}^{\infty} \min\{\ell(x;\sigma), 0\}\mathcal{N}(t;0,\tau)\mathrm{d}x\right]\mathrm{d}t$$

$$= \int_{-\infty}^{\infty} \mathcal{N}(t;0,\tau) \left[\int_{-\infty}^{\infty} \min\{\ell(x;\sigma), 0\}\mathrm{d}x\right]\mathrm{d}t$$

$$= \underbrace{\left[\int_{-\infty}^{\infty} \mathcal{N}(t;0,\tau)\mathrm{d}t\right]}_{=1} \left[\int_{-\infty}^{\infty} \min\{\ell(x;\sigma), 0\}\mathrm{d}x\right]$$

$$= \int_{-\infty}^{\infty} \min\{\ell(x;\sigma), 0\}\mathrm{d}x.$$

Here, the inequality follows from (19), the equalities that follow change the order of the integral, apply variable change $x = x - t$, rearrange constants within integrals, note that $\mathcal{N}(t;0,\tau)$ is a density function integrating to 1. The monotonicity proof is complete since we proved the desired inequality (18).

Now, to see that $\sigma = \sigma_{\mathrm{g}}$ satisfies (2), consider the stronger condition

$$\int_{-\infty}^{\infty} \min\{e^\varepsilon f_{\mathrm{m}}(x;\sigma, K) - f_{\mathrm{m}}(x + \varphi;\sigma, K), 0\}\mathrm{d}x + (1 - \eta)\delta \geq 0 \quad \forall \varphi \in [0, \Delta],$$

which coincides with the definition of $(\varepsilon, (1 - \eta)\delta)$-DP (*cf.* proof of Theorem 3.2). A sufficient condition for this is when each mixture density $\frac{1}{\sqrt{2\pi}\sigma} \exp(-(x - \mu)^2/(2\sigma^2))$ of (1) satisfies $(\varepsilon, (1 - \eta)\delta)$-DP since approximate DP is closed under mixtures (Selvi et al., 2025, §2.1). Moreover, since approximate DP is also closed under shifting distributions (*cf.* proof of Lemma B.1), it is sufficient to show the zero-mean Gaussian density $\frac{1}{\sqrt{2\pi}\sigma} \exp(-x^2/(2\sigma^2))$ satisfies $(\varepsilon, (1 - \eta)\delta)$-DP. We can therefore borrow the standard deviation $\sigma_{\mathrm{g}}$ needed for the Gaussian mechanism (Dwork & Roth, 2014, Thm 3.22) or the analytic Gaussian mechanism (Balle & Wang, 2018), and conclude that a sufficient condition for (2) is $\sigma = \sigma_{\mathrm{g}}$. □

## B.7. Proof of Theorem 3.5

We first note that, for the runtime analysis, we may set $\Delta = 1$ without loss of generality. To see this, let $f_{\mathrm{m}}^\Delta(\cdot;\sigma, K)$ denote the multi-Gaussian density with sensitivity $\Delta$ (which we did not parameterize before as $\Delta > 0$ was fixed). Under the change of variables $y = x/\Delta$ and $\lambda = \sigma/\Delta$, we have

$$f_{\mathrm{m}}^\Delta(\Delta y; \Delta\lambda, K) = \frac{1}{\Delta} f_{\mathrm{m}}^1(y; \lambda, K).$$

Thus, the privacy integral in (2) with shift $\varphi \in [0, \Delta]$ is equivalent to the corresponding normalized integral with shift $\varphi/\Delta \in [0, 1]$. Moreover, the grid size also normalizes, since

$$\frac{\beta}{\Delta} = \frac{1}{\lceil 1/(\sqrt{2\pi}\eta\lambda\delta)\rceil}.$$

Hence the algorithmic complexity depends on the dimensionless scale $\lambda = \sigma/\Delta$, and the scale for the original sensitivity-$\Delta$ problem is recovered as $\sigma = \Delta\lambda$.

The correctness of Algorithm 1 follows directly from Lemma 3.4. The first step of this algorithm is to compute the analytic Gaussian mechanism, which sets $\sigma_g$ in Lemma 3.4. In light of the monotonicity presented in this lemma, the bisection method finds the smallest $\sigma$ value so that any $\sigma' \geq \sigma$ satisfies condition (2). The computation for checking whether a given $\sigma$ satisfies this condition is given in Algorithm 2, which simply evaluates the left-hand side of the constraint (2) for every $\varphi$ in the grid. Note that this grid in $\varphi$ is a function of $\beta$. To make sure that *(i)* $\beta \leq \sqrt{2\pi}\eta\sigma\delta$ and also *(ii)* $\beta$ divides $\Delta$, Algorithm 1 sets $\beta$ as $\Delta/\lceil\Delta/(\sqrt{2\pi}\eta\sigma\delta)\rceil$.

For the runtime complexity, first note that we treat the tolerance in the bisection search method as a constant. One needs to be careful in implementation since numerical tolerances should be reflected in the presentation of $\delta$ to ensure feasibility.

Note that Algorithm 1 first finds the standard deviation $r = \sigma_g$ needed by the analytic Gaussian mechanism. The implementation of the analytic Gaussian (Balle & Wang, 2018) is via a bisection search method where the root is sought in a region with an upper bound for $\sigma_g$ as $\mathcal{O}(\varepsilon^{-1}\sqrt{\log(\delta^{-1})})$, where we use the normalization $\Delta = 1$. The function, however, can be evaluated in constant time. Given that Algorithm 1 applies a bisection search within $(0, r)$, the runtime of the rest of this algorithm dominates the computation of the analytic Gaussian. We borrow the upper bound $r = \mathcal{O}(\varepsilon^{-1}\sqrt{\log(\delta^{-1})})$ for the analysis of our bisection search method next.

Our bisection method is restricted to an interval of radius $\mathcal{O}(\varepsilon^{-1}\sqrt{\log(\delta^{-1})})$. Since our paper also considers regimes with $\varepsilon > 1$, we write the logarithmic dependence on $\varepsilon$ as $\log(1 + \frac{1}{\varepsilon})$ rather than $\log(1/\varepsilon)$; this keeps the bound nonnegative and treats the contribution of $\varepsilon$ as constant once $\varepsilon$ is large. Similarly, we write the $\delta$-dependent logarithmic term as $\log(1 + \log\frac{1}{\delta})$ to avoid a regime-specific discussion of when $\log\log(1/\delta)$ is nonnegative. The number of bisection iterations is therefore

$$\mathcal{O}\left(\log\left(1 + \frac{1}{\varepsilon}\right) + \log\left(1 + \log\frac{1}{\delta}\right)\right).$$

The iterations of this bisection search, as shown in Algorithm 2, compute an integral, in total

$$\mathcal{O}\left(\frac{1}{\eta\delta}\right)$$

times. The computation of this integral has complexity

$$\mathcal{O}\left(K^2\right)$$

because, after the normalization $\Delta = 1$, the integral can be approximated on a fixed support of radius $\mathcal{O}(K)$ since the Gaussian tails vanish exponentially fast, and the evaluation of $f_m(x; \sigma, K)$ takes time $\mathcal{O}(K)$. Multiplying the three displayed bounds concludes the runtime represented in the statement of this theorem.

### B.8. Proof of Corollary 3.7

For any two distribution $P, Q \in \mathcal{F}$, the $\alpha$-Rényi divergence, denoted by $D_\alpha(P\|Q)$, is given by

$$D_\alpha(P\|Q) = \frac{1}{\alpha - 1}\log\left(\int_{x \in \Omega} P(x)^\alpha Q(x)^{1-\alpha}\mathrm{d}x\right). \tag{20}$$

**Definition B.7** (Zero-Concentrated Differential Privacy (zCDP; Bun & Steinke 2016))**.** A mechanism $\mathcal{A} : \mathcal{N} \mapsto \mathcal{F}$ is called $\rho$-zCDP if for all neighbors $(D, D') \in \mathcal{N}$, it satisfies

$$D_\alpha(\mathcal{A}(D)\|\mathcal{A}(D')) \leq \rho\alpha, \quad \forall\alpha \in (1, \infty).$$

An important property of $\alpha$-Renyi divergence, according to Bun & Steinke (2016, Lemma 15), is the quasi-convexity: the convex combinations $\sum_i w_i P_i$ and $\sum_i w_i Q_i$ with $\sum w_i = 1$ and $w_i \geq 0$ satisfy

$$D_\alpha \left( \sum_i w_i P_i \middle\| \sum_i w_i Q_i \right) \leq \max_i \left\{ D_\alpha (P_i \| Q_i) \right\}. \tag{21}$$

According to Theorem 3.6, the Gaussian mechanism with noise $\mathcal{N}(\mu, \sigma^2)$ satisfies $\left( \frac{\Delta^2}{2\sigma^2} \right)$-zCDP. That is, for any $\mu$ and any neighbors $(D, D') \in \mathcal{N}$ satisfying $|q(D) - q(D')| \leq \Delta$, we have

$$D_\alpha \left( \mathcal{N}(\mu + q(D), \sigma) \| \mathcal{N}(\mu + q(D'), \sigma) \right) \leq \alpha \left( \frac{\Delta^2}{2\sigma^2} \right), \quad \forall \alpha \in (1, \infty). \tag{22}$$

Thus, our multi-Gaussian mixture mechanism $f_{\mathrm{m}}(x; \sigma, K) = \frac{1}{c_K} \sum_{k=-K}^K e^{-|k|\varepsilon} \phi(x; k\Delta, \sigma)$ is also $\left( \frac{\Delta^2}{2\sigma^2} \right)$-zCDP, which follows from the fact that for any $\alpha \in (1, \infty)$ we have

$$
\begin{aligned}
& D_\alpha \left( \frac{1}{c_K} \sum_{k=-K}^K e^{-|k|\varepsilon} \phi(\cdot; q(D) + k\Delta, \sigma) \middle\| \frac{1}{c_K} \sum_{k=-K}^K e^{-|k|\varepsilon} \phi(\cdot; q(D') + k\Delta, \sigma) \right) \\
=\; & D_\alpha \left( \sum_{k=-K}^K \frac{e^{-|k|\varepsilon}}{\sum_{k'=-K}^K e^{-|k'|\varepsilon}} \mathcal{N}(q(D) + k\Delta, \sigma) \middle\| \sum_{k=-K}^K \frac{e^{-|k|\varepsilon}}{\sum_{k'=-K}^K e^{-|k'|\varepsilon}} \mathcal{N}(q(D') + k\Delta, \sigma) \right) \\
\leq\; & \max_{k \in \{-K, -K+1, \cdots, K\}} \left\{ D_\alpha \left( \mathcal{N}(q(D) + k\Delta, \sigma) \middle\| \mathcal{N}(q(D') + k\Delta, \sigma) \right) \right\} \\
\leq\; & \alpha \left( \frac{\Delta^2}{2\sigma^2} \right)
\end{aligned}
$$

where the equality writes the distributions as mixtures of normalized Gaussians, the first inequality holds because of the quasi-convexity of $\alpha$-Renyi divergence and the second inequality holds from (22).

### B.9. Proof of Corollary 3.8

By Corollary 3.7, the $t$th multi-Gaussian mixture mechanism is $\rho_t$-zCDP with $\rho_t = \Delta_t^2 / (2\sigma_t^2)$. Since zCDP composes additively, the composition is $\rho_{\mathrm{tot}}$-zCDP with

$$\rho_{\mathrm{tot}} = \sum_{t=1}^T \rho_t = \sum_{t=1}^T \frac{\Delta_t^2}{2\sigma_t^2}.$$

The standard conversion from $\rho$-zCDP to approximate DP (Bun & Steinke, 2016, Lemma 3.5) then implies that, for any $\delta_{\mathrm{tot}} \in (0, 1)$, the composed mechanism is

$$\left( \rho_{\mathrm{tot}} + 2\sqrt{\rho_{\mathrm{tot}} \log(1/\delta_{\mathrm{tot}})}, \delta_{\mathrm{tot}} \right) \text{-DP}.$$

This proves the claim.

## C. Proofs for Section 4

### C.1. The quasi-Gaussian mixture distribution is well-defined

To show that the density function (3) of the quasi-Gaussian mixture distribution is well-defined, fix arbitrary $\sigma > 0$, and note that $f_{\mathrm{q}}(x; \sigma) \geq 0$ for all $x \in \mathbb{R}$ since all terms in its definition are nonnegative. Moreover, the density function integrates to

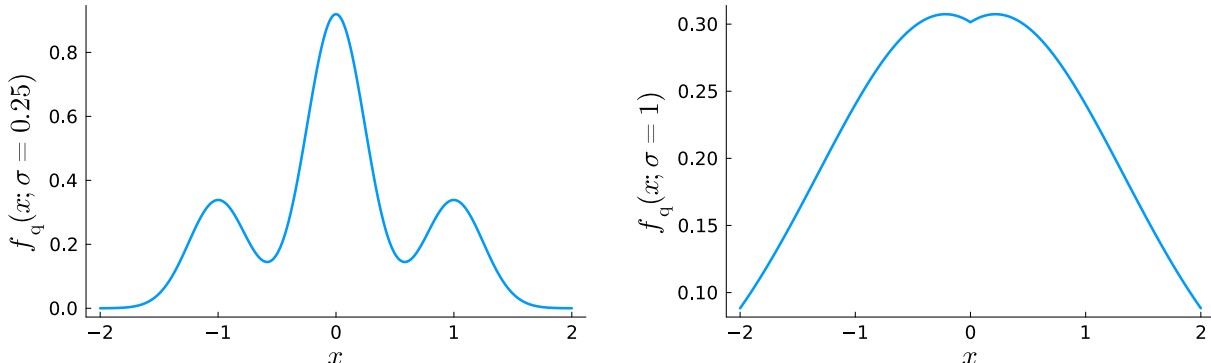

*Figure 4.* The probability density function (3) of the quasi-Gaussian mixture distribution for $\varepsilon = 1$ where $\sigma$ is set to $0.25$ (left) and $1$ (right).

1 since

$$
\begin{aligned}
\int_{-\infty}^{\infty} f_q(x;\sigma)\mathrm{d}x &= \frac{1}{c}\int_{-\infty}^{\infty} e^{\varepsilon}\exp\left(-\frac{x^2}{2\sigma^2}\right) + \exp\left(-\frac{(|x|-\Delta)^2}{2\sigma^2}\right)\mathrm{d}x \\
&= \frac{e^{\varepsilon}}{c}\int_{-\infty}^{\infty}\exp\left(-\frac{x^2}{2\sigma^2}\right)\mathrm{d}x + \frac{2}{c}\int_{0}^{\infty}\exp\left(-\frac{1}{2}\left(\frac{x-\Delta}{\sigma}\right)^2\right)\mathrm{d}x \\
&= \frac{e^{\varepsilon}}{c}\sqrt{2\pi}\sigma + \frac{2}{c}\int_{0}^{\infty}\exp\left(-\frac{1}{2}\left(\frac{x-\Delta}{\sigma}\right)^2\right)\mathrm{d}x \\
&= \frac{e^{\varepsilon}}{c}\sqrt{2\pi}\sigma + \frac{2}{c}\sigma\int_{-\Delta/\sigma}^{\infty}\exp\left(-\frac{1}{2}y^2\right)\mathrm{d}y \\
&= \frac{e^{\varepsilon}}{c}\sqrt{2\pi}\sigma + \frac{2}{c}\sqrt{2\pi}\sigma\Phi\left(\frac{\Delta}{\sigma}\right) \\
&= \frac{1}{c}\sqrt{2\pi}\sigma\left(e^{\varepsilon} + 2\Phi\left(\frac{\Delta}{\sigma}\right)\right) = 1.
\end{aligned}
$$

The first equality follows from Definition 4.1, the second equality applies algebraic manipulations, the third equality recognized the first integrand as a Gaussian density function, the fourth equality applies variable change $y = (x - \Delta)/\sigma$, the fifth equality recognizes the integrand as a Gaussian density function, and the final equations rearrange terms and substitute the definition of $c$.

In Figure 4, we visualize the probability density function (3) of the quasi-Gaussian mixture distribution for $\varepsilon = 1.0$ for smaller and larger values of $\sigma$.

### C.2. The cumulative distribution function of the quasi-Gaussian mixture distribution

For fixed $\varepsilon, \Delta, \sigma > 0$, the cumulative distribution function $F_q(\bar{x}; \sigma)$ of the quasi-Gaussian mixture distribution is given by:

$$
\begin{aligned}
F_q(\bar{x};\sigma) &= \int_{-\infty}^{\bar{x}} f_q(x;\sigma)\mathrm{d}x \\
&= \underbrace{\frac{1}{c}\int_{-\infty}^{\bar{x}} e^{\varepsilon}\exp\left(-\frac{x^2}{2\sigma^2}\right)\mathrm{d}x}_{(i)} + \underbrace{\frac{1}{c}\int_{-\infty}^{\bar{x}}\exp\left(-\frac{(|x|-\Delta)^2}{2\sigma^2}\right)\mathrm{d}x}_{(ii)}
\end{aligned}
$$

Here, term $(i)$ can be written in closed form:

$$\frac{1}{c}\int_{-\infty}^{\bar{x}} e^{\varepsilon}\exp\left(-\frac{x^2}{2\sigma^2}\right)\mathrm{d}x = \frac{e^{\varepsilon}}{c}\sqrt{2\pi}\sigma\Phi\left(\frac{\bar{x}}{\sigma}\right) = \frac{e^{\varepsilon}\Phi\left(\frac{\bar{x}}{\sigma}\right)}{e^{\varepsilon}+2\Phi\left(\frac{\Delta}{\sigma}\right)}.$$

Term $(ii)$ can be studied in two cases. For the first case, suppose $\bar{x}<0$. We have

$$\frac{1}{c}\int_{-\infty}^{\bar{x}}\exp\left(-\frac{(|x|-\Delta)^2}{2\sigma^2}\right)\mathrm{d}x = \frac{1}{c}\int_{-\infty}^{\bar{x}}\exp\left(-\frac{(x+\Delta)^2}{2\sigma^2}\right)\mathrm{d}x$$

$$= \frac{\sqrt{2\pi}\sigma}{c}\Phi\left(\frac{\bar{x}+\Delta}{\sigma}\right) = \frac{\Phi\left(\frac{\bar{x}+\Delta}{\sigma}\right)}{e^{\varepsilon}+2\Phi\left(\frac{\Delta}{\sigma}\right)}.$$

For the case $\bar{x}\geq 0$, we can write term $(ii)$ as

$$\frac{1}{c}\int_{-\infty}^{\bar{x}}\exp\left(-\frac{(|x|-\Delta)^2}{2\sigma^2}\right)\mathrm{d}x = \frac{1}{c}\int_{-\infty}^{0}\exp\left(-\frac{(x+\Delta)^2}{2\sigma^2}\right)\mathrm{d}x + \frac{1}{c}\int_{0}^{\bar{x}}\exp\left(-\frac{(x-\Delta)^2}{2\sigma^2}\right)\mathrm{d}x$$

$$= \frac{\sqrt{2\pi}\sigma}{c}\Phi\left(\frac{\Delta}{\sigma}\right) + \frac{1}{c}\int_{0}^{\bar{x}}\exp\left(-\frac{(x-\Delta)^2}{2\sigma^2}\right)\mathrm{d}x$$

$$= \frac{\sqrt{2\pi}\sigma}{c}\Phi\left(\frac{\Delta}{\sigma}\right) + \frac{\sqrt{2\pi}\sigma}{c}\left(\Phi\left(\frac{\bar{x}-\Delta}{\sigma}\right) - \Phi\left(-\frac{\Delta}{\sigma}\right)\right)$$

$$= \frac{\Phi\left(\frac{\Delta}{\sigma}\right) + \Phi\left(\frac{\bar{x}-\Delta}{\sigma}\right) - \Phi\left(-\frac{\Delta}{\sigma}\right)}{e^{\varepsilon}+2\Phi\left(\frac{\Delta}{\sigma}\right)}.$$

We thus have

$$F_{\mathrm{q}}(\bar{x};\sigma) = \begin{cases} \dfrac{e^{\varepsilon}\Phi\left(\frac{\bar{x}}{\sigma}\right) + \Phi\left(\frac{\bar{x}+\Delta}{\sigma}\right)}{e^{\varepsilon}+2\Phi\left(\frac{\Delta}{\sigma}\right)} & \text{if } \bar{x}<0 \\[3ex] \dfrac{e^{\varepsilon}\Phi\left(\frac{\bar{x}}{\sigma}\right) + \Phi\left(\frac{\bar{x}-\Delta}{\sigma}\right) + \Phi\left(\frac{\Delta}{\sigma}\right) - \Phi\left(-\frac{\Delta}{\sigma}\right)}{e^{\varepsilon}+2\Phi\left(\frac{\Delta}{\sigma}\right)} & \text{if } \bar{x}\geq 0. \end{cases}$$

### C.3. Sampling from the quasi-Gaussian mixture distribution

Since the quasi-Gaussian is a mixture distribution, we decide which of the two distributions to sample noise from via $\tilde{\theta}\sim\text{Bernoulli}(e^{\varepsilon}/(e^{\varepsilon}+2\Phi(\Delta/\sigma)))$. If $\tilde{\theta}=1$, then we can simply sample noise from the zero-mean Gaussian distribution with standard deviation $\sigma$. If $\tilde{\theta}=0$, then we sample noise from the distribution with density $\propto \exp(-(|x|-\Delta)^2)/(2\sigma^2))$. This could be obtained by sampling from $\propto \exp(-(z-\Delta)^2/(2\sigma^2))\cdot\mathbb{I}[z\geq 0]$, where $\mathbb{I}$ denotes the indicator function, and then flipping the sign of $z$ with $1/2$ probability. The density $\propto \exp(-(z-\Delta)^2/(2\sigma^2))\cdot\mathbb{I}[z\geq 0]$ coincides with the truncated Gaussian distribution with mean $\Delta$ and standard deviation $\sigma$, whose inverse cumulative density function is given by $\Delta+\sigma\Phi^{-1}(\Phi(-\Delta/\sigma)+p\cdot\Phi(\Delta/\sigma))$, $p\sim\text{Uniform}(0,1)$. In summary, we can sample noise from the quasi-Gaussian mechanism via Algorithm 5, where $\text{Uniform}(0,1)$ denotes the uniform distribution supported on $(0,1)$, $\text{Bernoulli}(1/2)$ denotes the Bernoulli distribution with success probability $1/2$, and $\mathcal{N}(0,\sigma)$ denotes the zero-mean Gaussian with standard deviation $\sigma$.

---

**Algorithm 5** Sampling noise from the quasi-Gaussian distribution.

---

**Input:** privacy parameter $\varepsilon > 0$, sensitivity $\Delta > 0$, variance parameter $\sigma > 0$
**Output:** a sample from the quasi-Gaussian distribution with density (3)
Sample $\theta \sim \text{Bernoulli}(e^\varepsilon/(e^\varepsilon + 2\Phi(\Delta/\sigma)))$.
**if** $\theta = 1$ **then**
    Sample $x \sim \mathcal{N}(0, \sigma)$.
**else**
    Sample $p \sim \text{Uniform}(0, 1)$.
    Set $x = \Delta + \sigma\Phi^{-1}(\Phi(-\Delta/\sigma) + p \cdot \Phi(\Delta/\sigma))$.
    Sample $s \sim \text{Bernoulli}(1/2)$.
    **if** $s = 1$ **then**
        Update $x = -x$.
    **end if**
**end if**
**Return:** $x$.

---

### C.4. Noise amplitude and power of the quasi-Gaussian mixture distributions

For the *noise amplitude* of the random variable $\tilde{X}$ with density $f_{\text{q}}(\cdot; \sigma)$, we have:

$$\mathbb{E}_{\tilde{X}}[|x|] = \frac{e^\varepsilon}{c} \underbrace{\int_{-\infty}^\infty |x| \exp\left(-\frac{x^2}{2\sigma^2}\right) \mathrm{d}x}_{(i)} + \frac{1}{c} \underbrace{\int_{-\infty}^\infty |x| \exp\left(-\frac{(|x| - \Delta)^2}{2\sigma^2}\right) \mathrm{d}x}_{(ii)}.$$

Term $(i)$ satisfies:

$$\int_{-\infty}^\infty |x| \exp\left(-\frac{x^2}{2\sigma^2}\right) \mathrm{d}x = 2 \int_0^\infty x \exp\left(-\frac{x^2}{2\sigma^2}\right) \mathrm{d}x = 2\sigma^2.$$

Term $(ii)$ satisfies:

$$\int_{-\infty}^\infty |x| \exp\left(-\frac{(|x| - \Delta)^2}{2\sigma^2}\right) \mathrm{d}x$$
$$= 2 \int_0^\infty x \exp\left(-\frac{(x - \Delta)^2}{2\sigma^2}\right) \mathrm{d}x$$
$$= 2 \int_{-\Delta/\sigma}^\infty (\sigma u + \Delta) \exp\left(-\frac{u^2}{2}\right) \sigma \mathrm{d}u$$
$$= 2\sigma^2 \int_{-\Delta/\sigma}^\infty u \exp\left(-\frac{u^2}{2}\right) \mathrm{d}u + 2\sigma\Delta \int_{-\Delta/\sigma}^\infty \exp\left(-\frac{u^2}{2}\right) \mathrm{d}u$$
$$= 2\sigma^2 \exp\left(-\frac{\Delta^2}{2\sigma^2}\right) + 2\sigma\Delta\sqrt{2\pi}\Phi\left(\frac{\Delta}{\sigma}\right)$$

where the first equality exploits the symmetry of the absolute value, the second equality applies variable change $u = (x - \Delta)/\sigma$, the third equality exploits the linearity of integrals and the final equality recognizes $\exp(-u^2/2)$ as the (unnormalized) standard Gaussian density. We conclude

$$\mathbb{E}_{\tilde{X}}[|x|] = \frac{e^\varepsilon}{c} 2\sigma^2 + \frac{1}{c}\left(2\sigma^2 \exp\left(-\frac{\Delta^2}{2\sigma^2}\right) + 2\sigma\Delta\sqrt{2\pi}\Phi\left(\frac{\Delta}{\sigma}\right)\right)$$
$$= \frac{2\sigma^2\left(e^\varepsilon + \exp\left(-\frac{\Delta^2}{2\sigma^2}\right)\right) + 2\sigma\Delta\sqrt{2\pi}\Phi\left(\frac{\Delta}{\sigma}\right)}{\sqrt{2\pi}\sigma\left(e^\varepsilon + 2\Phi\left(\frac{\Delta}{\sigma}\right)\right)}$$

$$= \frac{\sqrt{2/\pi}\,\sigma\left(e^\varepsilon + \exp\left(-\frac{\Delta^2}{2\sigma^2}\right)\right) + 2\Delta\Phi\left(\frac{\Delta}{\sigma}\right)}{e^\varepsilon + 2\Phi\left(\frac{\Delta}{\sigma}\right)},$$

where the first equality is derived above, the second equality rearranges terms, and the final equality cancels the common terms from the numerator and the denominator.

For the *noise power* of the random variable $\tilde{X}$ with density $f_q(\cdot\,;\sigma)$, we have:

$$\mathbb{E}_{\tilde{X}}[x^2] = \frac{e^\varepsilon}{c} \underbrace{\int_{-\infty}^{\infty} x^2 \exp\left(-\frac{x^2}{2\sigma^2}\right) dx}_{(i)} + \frac{1}{c} \underbrace{\int_{-\infty}^{\infty} x^2 \exp\left(-\frac{(|x|-\Delta)^2}{2\sigma^2}\right) dx}_{(ii)}.$$

Term $(i)$ satisfies

$$\int_{-\infty}^{\infty} x^2 \exp\left(-\frac{x^2}{2\sigma^2}\right) dx = \sigma^3\sqrt{2\pi}$$

since we recognize $\exp(-x^2/(2\sigma^2))$ as the (unnormalized) Gaussian density with variance $\sigma^2$.

Term $(ii)$ satisfies

$$\int_{-\infty}^{\infty} x^2 \exp\left(-\frac{(|x|-\Delta)^2}{2\sigma^2}\right) dx$$

$$=2\int_{0}^{\infty} x^2 \exp\left(-\frac{(x-\Delta)^2}{2\sigma^2}\right) dx$$

$$=2\sigma\int_{-\Delta/\sigma}^{\infty} (\sigma u + \Delta)^2 \exp\left(-\frac{u^2}{2}\right) du$$

$$=2\sigma^3\int_{-\Delta/\sigma}^{\infty} u^2 \exp\left(-\frac{u^2}{2}\right) du + 2\sigma\Delta^2 \underbrace{\int_{-\Delta/\sigma}^{\infty} \exp\left(-\frac{u^2}{2}\right) du}_{=\sqrt{2\pi}\Phi(\Delta/\sigma)}$$

$$+ 4\sigma^2\Delta \underbrace{\int_{-\Delta/\sigma}^{\infty} u \exp\left(-\frac{u^2}{2}\right) du}_{=\exp(-(\Delta^2/(2\sigma^2)))}$$

$$=2\sigma^3\int_{-\Delta/\sigma}^{\infty} u^2 \exp\left(-\frac{u^2}{2}\right) du + 2\sigma\Delta^2\sqrt{2\pi}\Phi\left(\frac{\Delta}{\sigma}\right) + 4\sigma^2\Delta\exp\left(-\frac{\Delta^2}{2\sigma^2}\right)$$

$$=2\sigma^3\left[-u\exp\left(-\frac{u^2}{2}\right)\right]_{-\Delta/\sigma}^{\infty} + 2\sigma^3 \underbrace{\int_{-\Delta/\sigma}^{\infty} \exp\left(-\frac{u^2}{2}\right) du}_{=\sqrt{2\pi}\Phi(\Delta/\sigma)}$$

$$+ 2\sigma\Delta^2\sqrt{2\pi}\Phi\left(\frac{\Delta}{\sigma}\right) + 4\sigma^2\Delta\exp\left(-\frac{\Delta^2}{2\sigma^2}\right)$$

$$=-2\sigma^2\Delta\exp\left(-\frac{\Delta^2}{2\sigma^2}\right) + 2\sigma^3\sqrt{2\pi}\Phi\left(\frac{\Delta}{\sigma}\right) + 2\sigma\Delta^2\sqrt{2\pi}\Phi\left(\frac{\Delta}{\sigma}\right) + 4\sigma^2\Delta\exp\left(-\frac{\Delta^2}{2\sigma^2}\right)$$

$$=2\sigma^2\Delta\exp\left(-\frac{\Delta^2}{2\sigma^2}\right) + 2\sigma\sqrt{2\pi}\Phi\left(\frac{\Delta}{\sigma}\right)(\sigma^2 + \Delta^2)$$

$$=2\sigma\sqrt{2\pi}\left(\Phi\left(\frac{\Delta}{\sigma}\right)(\sigma^2 + \Delta^2) + \frac{\sigma\Delta}{\sqrt{2\pi}}\exp\left(-\frac{\Delta^2}{2\sigma^2}\right)\right).$$

where the first equality exploits the symmetry of the square and absolute value functions, the second equality uses the variable change $u = (x - \Delta)/\sigma$, the third equality breaks down the square term into three components, exploits the linearity

of integral, and recognizes $\exp(-u^2/(2\sigma^2))$ as the (unnormalized) Gaussian density with variance $\sigma^2$, the fourth equality substitutes integrals with their closed-form expressions except for the first integral, hence the fifth equality uses integration by parts, the sixth equality writes the expression without an integral, and the final equalities rearrange terms. We conclude:

$$
\mathbb{E}_{\tilde{X}}[x^2] = \frac{e^\varepsilon}{c}\sigma^3\sqrt{2\pi} + \frac{1}{c}\left(2\sigma\sqrt{2\pi}\left(\Phi\left(\frac{\Delta}{\sigma}\right)(\sigma^2 + \Delta^2) + \frac{\sigma\Delta}{\sqrt{2\pi}}\exp\left(-\frac{\Delta^2}{2\sigma^2}\right)\right)\right)
$$

$$
= \frac{e^\varepsilon\sigma^3\sqrt{2\pi} + 2\sigma\sqrt{2\pi}\left(\Phi\left(\frac{\Delta}{\sigma}\right)(\sigma^2 + \Delta^2) + \frac{\sigma\Delta}{\sqrt{2\pi}}\exp\left(-\frac{\Delta^2}{2\sigma^2}\right)\right)}{\sqrt{2\pi}\sigma\left(e^\varepsilon + 2\Phi\left(\frac{\Delta}{\sigma}\right)\right)}
$$

$$
= \frac{e^\varepsilon\sigma^2 + 2\left(\Phi\left(\frac{\Delta}{\sigma}\right)(\sigma^2 + \Delta^2) + \frac{\sigma\Delta}{\sqrt{2\pi}}\exp\left(-\frac{\Delta^2}{2\sigma^2}\right)\right)}{e^\varepsilon + 2\Phi\left(\frac{\Delta}{\sigma}\right)}.
$$

### C.5. Proof of Theorem 4.2

From Lemma B.1 it suffices to show that

$$
\int_{x\in A}(e^\varepsilon f_{\mathrm{q}}(x + \varphi; \sigma) - f_{\mathrm{q}}(x; \sigma))\mathrm{d}x \geq -\delta \qquad \forall\varphi \in [0, \Delta], \ \forall A \in \mathcal{F}
$$

holds. To this end, in the rest of this proof, we will show that for an arbitrary $\varphi \in [0, \Delta]$ and $A \in \mathcal{F}$, the desired inequality holds. We first break down the left-hand side of the inequality into three cases by exploiting the linearity of integrals:

$$
\int_{x\in A}(e^\varepsilon f_{\mathrm{q}}(x + \varphi; \sigma) - f_{\mathrm{q}}(x; \sigma))\mathrm{d}x
$$

$$
= \int_{x\in A:x<-\Delta-\varphi}(e^\varepsilon f_{\mathrm{q}}(x + \varphi; \sigma) - f_{\mathrm{q}}(x; \sigma))\mathrm{d}x + \tag{23a}
$$

$$
\int_{x\in A\cap[-\Delta-\varphi,\Delta-\varphi]}(e^\varepsilon f_{\mathrm{q}}(x + \varphi; \sigma) - f_{\mathrm{q}}(x; \sigma))\mathrm{d}x + \tag{23b}
$$

$$
\int_{x\in A:x>\Delta-\varphi}(e^\varepsilon f_{\mathrm{q}}(x + \varphi; \sigma) - f_{\mathrm{q}}(x; \sigma))\mathrm{d}x. \tag{23c}
$$

To complete the proof, we show that the sum of (23a), (23b) and (23c) is greater than or equal to $-\delta$. The first terms are studied next:

(23a): For any $x < 0$, we have

$$
\frac{\partial}{\partial x}f_{\mathrm{q}}(x; \sigma) = \frac{e^\varepsilon}{c}\left(\frac{-2x}{2\sigma^2}\right)\exp\left(-\frac{x^2}{2\sigma^2}\right) + \frac{1}{c}\left(-\frac{2(x + \Delta)}{2\sigma^2}\right)\exp\left(-\frac{(x + \Delta)^2}{2\sigma^2}\right)
$$

where all terms are nonnegative as we have $x < -\Delta - \varphi \leq 0$. This concludes that $f_{\mathrm{q}}(x; \sigma)$ is increasing in the region $(-\infty, -\Delta)$ and we thus have:

$$
e^\varepsilon f_{\mathrm{q}}(x + \varphi; \sigma) - f_{\mathrm{q}}(x; \sigma) \geq f_{\mathrm{q}}(x + \varphi; \sigma) - f_{\mathrm{q}}(x; \sigma) \geq 0.
$$

(23b): We have

$$
e^\varepsilon f_{\mathrm{q}}(x + \varphi; \sigma) - f_{\mathrm{q}}(x; \sigma) \geq e^\varepsilon f_{\mathrm{q,min}}(\sigma) - f_{\mathrm{q}}(x; \sigma) \geq f_{\mathrm{q,max}}(\sigma) - f_{\mathrm{q}}(x; \sigma) \geq 0,
$$

where the first inequality holds since $x + \varphi \in [-\Delta, \Delta]$ and $f_{\mathrm{q}}(\cdot; \sigma)$ is symmetric, the second inequality holds since $\sigma \geq \sigma_2$ as specified in the statement of this theorem, and the final inequality is due to $x \leq \Delta - \varphi \leq \Delta$.

This concludes that (23a) + (23b) $\geq 0$, and it will thus be sufficient to show (23c) $\geq -\delta$.

(23c): We have

$$\int_{x \in A : x > \Delta - \varphi} (e^{\varepsilon} f_{\mathrm{q}}(x + \varphi; \sigma) - f_{\mathrm{q}}(x; \sigma)) \mathrm{d}x = (23\mathrm{c})$$

$$\geq \int_{x \in A : x > \Delta - \varphi} (e^{\varepsilon} f_{\mathrm{q}}(x + \Delta; \sigma) - f_{\mathrm{q}}(x; \sigma)) \mathrm{d}x$$

$$\geq \int_{x \in A : x > \Delta - \varphi} (e^{\varepsilon} f_{\mathrm{q}}(x + \Delta; \sigma) - f_{\mathrm{q}}(x; \sigma))^{-} \mathrm{d}x$$

$$\geq \int_{x \geq 0} (e^{\varepsilon} f_{\mathrm{q}}(x + \Delta; \sigma) - f_{\mathrm{q}}(x; \sigma))^{-} \mathrm{d}x \tag{24}$$

where $z^{-}$ denotes the negative part of $z$, (*i.e.*, $z^{-} = \min\{0, z\}$). Here, the first inequality follows since, analogously to the case of (23a), $\frac{\mathrm{d}}{\mathrm{d}x} f_{\mathrm{q}}(x; \sigma)$ is decreasing in region $x \in (\Delta, \infty)$ and as a result $f_{\mathrm{q}}(x + \varphi) \geq f_{\mathrm{q}}(x + \Delta)$ holds. The second and third inequalities follow as any $z \in \mathbb{R}$ satisfies $z \geq z^{-}$ and $0 \geq z^{-}$, respectively. The domain $\{x : x \in \mathbb{R}\}$ of integration in (24) can be replaced with $\{x : x > \varepsilon \sigma^2 / \Delta\}$ without loss of generality since the term whose negative part is taken satisfies:

$$e^{\varepsilon} f_{\mathrm{q}}(x + \Delta) - f_{\mathrm{q}}(x; \sigma)$$

$$= \frac{e^{\varepsilon}}{c} \left[ \exp\left( \varepsilon - \frac{(x + \Delta)^2}{2\sigma^2} \right) + \exp\left( -\frac{x^2}{2\sigma^2} \right) - \exp\left( -\frac{x^2}{2\sigma^2} \right) - \exp\left( -\varepsilon - \frac{(x - \Delta)^2}{2\sigma^2} \right) \right]$$

$$= \frac{e^{\varepsilon}}{c} \left[ \exp\left( \varepsilon - \frac{(x + \Delta)^2}{2\sigma^2} \right) - \exp\left( -\varepsilon - \frac{(x - \Delta)^2}{2\sigma^2} \right) \right],$$

which is nonnegative if and only if

$$\varepsilon - \frac{(x + \Delta)^2}{2\sigma^2} \geq -\varepsilon - \frac{(x - \Delta)^2}{2\sigma^2} \iff 2\varepsilon \geq \frac{(x + \Delta)^2 - (x - \Delta)^2}{2\sigma^2} \iff x \leq \frac{\varepsilon \sigma^2}{\Delta}.$$

We can thus conclude:

$$(23\mathrm{c}) \geq (24) \geq \int_{x > \varepsilon \sigma^2 / \Delta} (e^{\varepsilon} f_{\mathrm{q}}(x + \Delta; \sigma) - f_{\mathrm{q}}(x; \sigma)) \mathrm{d}x$$

$$= \frac{1}{c} \int_{\varepsilon \sigma^2 / \Delta}^{\infty} e^{2\varepsilon} \exp\left( -\frac{(x + \Delta)^2}{2\sigma^2} \right) - \exp\left( -\frac{(x - \Delta)^2}{2\sigma^2} \right) \mathrm{d}x$$

$$= \frac{e^{2\varepsilon} \sqrt{2\pi} \sigma}{c} \Phi\left( -\frac{\varepsilon \sigma}{\Delta} - \frac{\Delta}{\sigma} \right) - \frac{1}{c} \int_{\varepsilon \sigma^2 / \Delta}^{\infty} \exp\left( -\frac{(x - \Delta)^2}{2\sigma^2} \right) \mathrm{d}x$$

$$= \frac{e^{2\varepsilon} \sqrt{2\pi} \sigma}{c} \Phi\left( -\frac{\varepsilon \sigma}{\Delta} - \frac{\Delta}{\sigma} \right) - \frac{\sqrt{2\pi} \sigma}{c} \Phi\left( -\frac{\varepsilon \sigma}{\Delta} + \frac{\Delta}{\sigma} \right)$$

$$= \frac{1}{e^{\varepsilon} + 2\Phi\left( \frac{\Delta}{\sigma} \right)} \left( e^{2\varepsilon} \Phi\left( -\frac{\varepsilon \sigma}{\Delta} - \frac{\Delta}{\sigma} \right) - \Phi\left( -\frac{\varepsilon \sigma}{\Delta} + \frac{\Delta}{\sigma} \right) \right)$$

$$= \frac{h_1(\sigma)}{h_2(\sigma) / \delta} \geq -\delta.$$

Here, the first two inequalities were proved earlier, the four equalities that follow derive the integral in closed form by recognizing the integrands as Gaussian density functions, the final equality substitutes $h_1(\sigma)$ and $h_2(\sigma)$ as defined in (4b), and the final inequality follows from $\sigma \geq \sigma_1$ where $\sigma_1$ is defined in (4a). Note that while the final inequality is straightforward for $\sigma = \sigma_1$ by the definition of $\sigma_1$, to have the same conclusion for any $\sigma \geq \sigma_1$, we rely on Lemma 4.3.

We conclude the proof since the left-hand side of the DP constraints, that needs to be greater than or equal to $-\delta$, can be written as (23a) + (23b) + (23c), where (23a), (23b) $\geq 0$ and (23c) $\geq -\delta$.

### C.6. Proof of Lemma 4.3

To prove that the desired monotonicity, we investigate the derivative of the function of interest. To this end, note that

$$
\frac{\mathrm{d}}{\mathrm{d}\sigma}h_1(\sigma)
$$
$$
=e^{2\varepsilon}\left(-\frac{\varepsilon}{\Delta}+\frac{\Delta}{\sigma^2}\right)\Phi'\left(-\frac{\varepsilon\sigma}{\Delta}-\frac{\Delta}{\sigma}\right)-\left(-\frac{\varepsilon}{\Delta}-\frac{\Delta}{\sigma^2}\right)\Phi'\left(-\frac{\varepsilon\sigma}{\Delta}+\frac{\Delta}{\sigma}\right)
$$
$$
=\frac{e^{2\varepsilon}}{\sqrt{2\pi}}\left(-\frac{\varepsilon}{\Delta}+\frac{\Delta}{\sigma^2}\right)\exp\left(-\frac{1}{2}\left(\frac{\varepsilon\sigma}{\Delta}+\frac{\Delta}{\sigma}\right)^2\right)+\frac{1}{\sqrt{2\pi}}\left(\frac{\varepsilon}{\Delta}+\frac{\Delta}{\sigma^2}\right)\exp\left(-\frac{1}{2}\left(-\frac{\varepsilon\sigma}{\Delta}+\frac{\Delta}{\sigma}\right)^2\right)
$$
$$
=\frac{2\Delta}{\sqrt{2\pi}\sigma^2}\exp\left(-\frac{1}{2}\left(-\frac{\varepsilon\sigma}{\Delta}+\frac{\Delta}{\sigma}\right)^2\right),
$$

where the first equality follows from the chain rule, the second equality uses the definition of the Gaussian probability density function, and the final equality is by using the algebraic property $(a+b)^2 = (a-b)^2 + 4ab$. Moreover, we have

$$
\frac{\mathrm{d}}{\mathrm{d}\sigma}h_2(\sigma) = -2\delta\frac{\Delta}{\sigma^2}\Phi'\left(\frac{\Delta}{\sigma}\right) = -\frac{2\Delta}{\sqrt{2\pi}\sigma^2}\delta\exp\left(-\frac{1}{2}\left(\frac{\Delta}{\sigma}\right)^2\right)
$$

which concludes that

$$
\frac{\mathrm{d}}{\mathrm{d}\sigma}\left[h_1(\sigma)+h_2(\sigma)\right]
$$
$$
=\frac{2\Delta}{\sqrt{2\pi}\sigma^2}\exp\left(-\frac{1}{2}\left(-\frac{\varepsilon\sigma}{\Delta}+\frac{\Delta}{\sigma}\right)^2\right)-\frac{2\Delta}{\sqrt{2\pi}\sigma^2}\delta\exp\left(-\frac{1}{2}\left(\frac{\Delta}{\sigma}\right)^2\right)
$$
$$
=\frac{2\Delta}{\sqrt{2\pi}\sigma^2}\exp\left(-\frac{1}{2}\left(\frac{\Delta}{\sigma}\right)^2\right)\left(\exp\left(\varepsilon-\frac{1}{2}\left(\frac{\varepsilon\sigma}{\Delta}\right)^2\right)-\delta\right),
$$

which is positive whenever $\varepsilon - \varepsilon^2\sigma^2/(2\Delta^2) > \log\delta$ which is equivalent to $\sigma < \sqrt{2(\varepsilon - \log\delta)}\Delta/\varepsilon$.

To conclude the function is nonnegative for $\sigma \geq \sqrt{2(\varepsilon - \log\delta)}\Delta/\varepsilon$, note that

$$
h_1(\sigma)+h_2(\sigma) \geq h_1(\sigma)+2\delta
$$
$$
\geq -\Phi\left(-\frac{\varepsilon\sigma}{\Delta}+\frac{\Delta}{\sigma}\right)+2\delta, \tag{25}
$$

where the first inequality holds since $e^\varepsilon \geq 1$ and $\Phi(\Delta/\sigma) \geq \Phi(0) = 0.5$ and the second inequality follows from removing the positive part from the definition of $h_1(\sigma)$. The expression (25) is trivially nonnegative for $\delta \geq 0.5$. To conclude the proof, we next show that (25) is nonnegative for $\delta < 0.5$. To this end, suppose $\delta < 0.5$. Note that $\varepsilon\sigma/\Delta - \Delta/\sigma$ is monotonically increasing with respect to $\sigma$, so the fact that $\sigma \geq \sqrt{2(\varepsilon - \log\delta)}\Delta/\varepsilon$ implies

$$
\frac{\varepsilon\sigma}{\Delta}-\frac{\Delta}{\sigma} \geq \sqrt{2(\varepsilon - \log\delta)} - \frac{\varepsilon}{\sqrt{2(\varepsilon - \log\delta)}} = \frac{\varepsilon - 2\log\delta}{\sqrt{2(\varepsilon - \log\delta)}} > 0. \tag{26}
$$

By using the substitution $t = (\varepsilon - 2\log\delta)/\sqrt{2(\varepsilon - \log\delta)}$, we can further bound (25) via

$$
\Phi\left(-\frac{\varepsilon\sigma}{\Delta}+\frac{\Delta}{\sigma}\right) \leq \Phi(-t) \leq \frac{1}{\sqrt{2\pi}t}\exp\left(-\frac{\varepsilon^2 - 4(\varepsilon - \log\delta)\log\delta}{4(\varepsilon - \log\delta)}\right) \tag{27}
$$
$$
= \frac{\delta}{\sqrt{2\pi}t}\exp\left(-\frac{\varepsilon^2}{4(\varepsilon - \log\delta)}\right), \tag{28}
$$

where the first inequality is due to (26) and the second inequality is due to Mill's inequality, asserting that for $t > 0$ we have $\Phi(-t) \leq (1/\sqrt{2\pi}t) \exp\left(-t^2/2\right)$. This allows us to conclude the proof as:

$$(25) \geq 2\delta - \frac{\delta}{\sqrt{2\pi}t} \exp\left(-\frac{\varepsilon^2}{4(\varepsilon - \log \delta)}\right)$$

$$\geq 2\delta - \frac{1}{\sqrt{\pi \log 2}} \exp\left(-\frac{\varepsilon^2}{4(\varepsilon - \log \delta)}\right)\delta$$

$$\geq \delta > 0.$$

Here the first inequality follows from (27), the second inequality follows from

$$t = \frac{\varepsilon - 2\log \delta}{\sqrt{2(\varepsilon - \log \delta)}} \geq \frac{\varepsilon - \log \delta}{\sqrt{2(\varepsilon - \log \delta)}} = \sqrt{\frac{\varepsilon - \log \delta}{2}} \geq \sqrt{\frac{\log 2}{2}},$$

and the third inequality is due to the fact that $1/\sqrt{\pi \log 2} \leq 1$.

Finally, to conclude that $e^\varepsilon + 2 \geq \delta^{-1}$ implies the function is nonnegative everywhere, we investigate the limiting case where

$$\lim_{\sigma \to 0} h_1(\sigma) = -1$$

and

$$\lim_{\sigma \to 0} h_2(\sigma) = (e^\varepsilon + 2)\delta,$$

hold, hence the function $h_1(\sigma) + h_2(\sigma) \geq (e^\varepsilon + 2)\delta - 1 \geq 0$ is nonnegative everywhere under this condition.

### C.7. Proof of Lemma 4.4

We first state the unabridged lemma here.

**Lemma C.1** (Lemma 4.4 restated). *For any $\sigma > 0$ let*

$$x_1 := \frac{\Delta - \sqrt{\Delta^2 - 4\sigma^2}}{2}, \ x_2 := \frac{\Delta + \sqrt{\Delta^2 - 4\sigma^2}}{2}, \ g(x) := -e^\varepsilon + \frac{\Delta - x}{x}\exp\left(\frac{2x\Delta - \Delta^2}{2\sigma^2}\right).$$

*The following statements hold for $\min_{x \in [0, \Delta]} f_q(x; \sigma)$ and $\max_{x \in [0, \Delta]} f_q(x; \sigma)$:*

(i) *If $\Delta^2 \leq 4\sigma^2$, then $x = \Delta$ solves $\min_{x \in [0, \Delta]} f_q(x; \sigma)$. Moreover, $f_q(x; \sigma)$ is unimodal on the interval $(0, \Delta/2)$ with a unique maximum. The maximizer of this unimodal region also solves $\max_{x \in [0, \Delta]} f_q(x; \sigma)$.*

(ii) *If $\Delta^2 > 4\sigma^2$ and $g(x_2) \leq 0$, then $x = \Delta$ solves $\min_{x \in [0, \Delta]} f_q(x; \sigma)$. Moreover, $f_q(x; \sigma)$ is unimodal on the interval $(0, x_1)$ with a unique maximum. The maximizer of this unimodal region also solves $\max_{x \in [0, \Delta]} f_q(x; \sigma)$.*

(iii) *If $\Delta^2 > 4\sigma^2$ and $g(x_2) > 0$, then $f_q(x; \sigma)$ is unimodal on the interval $(\Delta/2, x_2)$ with a unique minimum. Either $x = \Delta$ or the minimizer of this unimodal region solves $\min_{x \in [0, \Delta]} f_q(x; \sigma)$. Moreover, $f_q(x; \sigma)$ is unimodal on the interval $(0, x_1)$ with a unique maximum. The maximizer of this unimodal region also solves $\max_{x \in [0, \Delta]} f_q(x; \sigma)$.*

Since this lemma explores the minimizer and maximizer of $f_q(x; \sigma)$ over a nonnegative interval $x \in [0, \Delta]$, we can rewrite it without an absolute value as

$$f_q(x; \sigma) = \frac{e^\varepsilon}{c}\exp\left(-\frac{x^2}{2\sigma^2}\right) + \frac{1}{c}\exp\left(-\frac{(x - \Delta)^2}{2\sigma^2}\right), \tag{29}$$

and its derivative as

$$\frac{\partial}{\partial x} f_q(x; \sigma) = \frac{e^\varepsilon}{c}\left(-\frac{x}{\sigma^2}\right)\exp\left(-\frac{x^2}{2\sigma^2}\right) + \frac{1}{c}\left(-\frac{x - \Delta}{\sigma^2}\right)\exp\left(-\frac{(x - \Delta)^2}{2\sigma^2}\right)$$

$$= \frac{1}{c\sigma^2}\exp\left(-\frac{x^2}{2\sigma^2}\right)\left[-e^\varepsilon x + (\Delta - x)\exp\left(\frac{2x\Delta - \Delta^2}{2\sigma^2}\right)\right], \tag{30}$$

where the first equality follows from the chain rule and the second equality rearranges terms. We now state and prove an intermediary lemma that will be used in the proof of Lemma 4.4.

**Lemma C.2.** *For any $x \in [0, \Delta/2)$, we have $f_q(x; \sigma) > f_q(\Delta - x; \sigma)$.*

*Proof.* We can use (29) and show

$$f_q(x; \sigma) - f_q(\Delta - x; \sigma) = \frac{e^\varepsilon - 1}{c} \left( \exp\left( -\frac{x^2}{2\sigma^2} \right) - \exp\left( -\frac{(x - \Delta)^2}{2\sigma^2} \right) \right).$$

For $x \in [0, \Delta/2)$, we have $(x - \Delta)^2 > x^2$, which implies $\exp(-\frac{x^2}{2\sigma^2}) > \exp(-\frac{(x-\Delta)^2}{2\sigma^2})$ and as a result concludes that the expression above is positive. $\square$

We can now prove the unabridged version of Lemma 4.4.

**Proof of Lemma 4.4.** Since we fix $\sigma > 0$, we will use the shorthand $f_q(x)$ and $f_q'(x)$ for $f_q(x; \sigma)$ and $\partial f_q(x; \sigma)/\partial x$, respectively. We will study the sign of the derivative (30) over $[0, \Delta]$ in order to conclude the proof. To this end, we first plug in $x \in \{0, \Delta/2, \Delta\}$ in (30) and note:

$$f_q'(0) > 0, \ f_q'(\Delta/2) < 0, \ f_q'(\Delta) < 0. \tag{31}$$

From the continuity of $f_q'$, there exists some small enough constant $\xi > 0$ such that $f_q'(x) > 0$ for all $x \in [0, \xi]$. For $x \in [\xi, \Delta]$, we can thus multiply the expression (30) by $x/x$ and obtain:

$$f_q'(x) = \underbrace{\frac{x}{c\sigma^2} \exp\left( -\frac{x^2}{2\sigma^2} \right)}_{=:g^+(x)} \underbrace{\left[ -e^\varepsilon + \frac{\Delta - x}{x} \exp\left( \frac{2x\Delta - \Delta^2}{2\sigma^2} \right) \right]}_{=:g(x)}.$$

Here, we have $g^+(x) > 0$ for all $x \in [0, \xi]$. Thus, the sign of $f_q'(x)$ coincides the sign of $g(x)$ in the region $x \in [\xi, \Delta]$. Then, (31) implies that:

$$g(\xi) > 0, \ g(\Delta/2) < 0, \ g(\Delta) < 0. \tag{32}$$

In order to explore the sign of $g(x)$, we further compute its derivative

$$\frac{\mathrm{d}}{\mathrm{d}x} g(x) =: g'(x) = \exp\left( \frac{2x\Delta - \Delta^2}{2\sigma^2} \right) \left( \frac{\Delta - x}{x} \cdot \frac{\Delta}{\sigma^2} - \frac{\Delta}{x^2} \right)$$

$$= \underbrace{\frac{\Delta}{\sigma^2 x^2} \exp\left( \frac{2x\Delta - \Delta^2}{2\sigma^2} \right)}_{=:g'^1(x)} \underbrace{(x(\Delta - x) - \sigma^2)}_{=:g'^2(x)},$$

where the first equality follows from the chain rule and the second equality rearranges terms. Since $g'^1(x) > 0$ for all $x \in [\xi, \Delta]$, we investigate the sign of $g'^2(x)$ by investigating three cases.

**Case 1 ($\Delta^2 \leq 4\sigma^2$):** Since $x(\Delta - x) \leq \Delta^2/4$ for all $x \in \mathbb{R}$, this case implies that $g'^2(x) \leq 0$ for all $x \in [\xi, \Delta]$, and allows us to conclude that $g$ is monotonically decreasing for $x \in [\xi, \Delta]$. Such monotonicity implies in (32) that there exists some $x_0 \in (\xi, \Delta/2)$ satisfying $g(x) > 0$ for all $x \in (\xi, x_0)$ and $g(x) < 0$ for all $x \in (x_0, \Delta]$. Thus, $f_q'$ is positive on $[0, x_0)$ and negative on $(x_0, \Delta]$, hence $x_0$ is the maximizer of $f_q$ and the minimizer is at the extreme point. From Lemma C.2 we can conclude that $\Delta$ is the minimizer. This discussion also shows that $f_q(x)$ is unimodal on $(0, \Delta/2)$. We thus proved item *(i)* of this lemma.

**Case 2 ($\Delta^2 > 4\sigma^2$):** Since $g'(x) = 0$ if and only if $g'^2(x) = 0$, and since $g'^2(x)$ is a quadratic function, we can solve $g'(x) = 0$ and obtain two roots

$$x_1 = \frac{\Delta - \sqrt{\Delta^2 - 4\sigma^2}}{2}, \quad x_2 = \frac{\Delta + \sqrt{\Delta^2 - 4\sigma^2}}{2}. \tag{33}$$

These roots, along with the fact that $g'(\xi) < 0$ and $g'(\Delta/2) > 0$ imply that $g(x)$ is decreasing in $x \in [\xi, x_1]$ and increasing in $x \in [x_1, x_2]$. This monotonicity implies

$$g(x_1) < g(\Delta/2) < 0, \tag{34}$$

since $\Delta/2 \in (x_1, x_2)$ and $g(\Delta/2) < 0$.

Thus far we discussed that $g(\xi) > 0$ (*cf.* equation 32), $g(x_1) < 0$ (*cf.* equation 34) and $g'(x) < 0$ in $(\xi, x_1)$ (*cf.* equation 33). This concludes that there exists some $y_{\max} \in (\xi, x_1)$ such that $g(x) > 0$ for $x \in (\xi, y_{\max})$ and $g(x) < 0$ for $x \in (y_{\max}, x_1)$, and $f'_q(x)$ exhibits a similar pattern as its sign coincides with the sign of $g(x)$. This shows that $f_q(x)$ is unimodal in the region $(0, x_1)$ and has a local maximizer $y_{\max}$. To investigate the behavior of $f_q(x)$ in the region $[x_1, \Delta]$, we further focus on two cases:

- Suppose that we have $g(x_2) \leq 0$. Given that $g(x)$ is increasing in $[x_1, x_2]$, this implies that $g(x) \leq 0$ for all $x \in [x_1, x_2]$. Furthermore, since we have no roots in $(x_2, \Delta]$, the fact that $g(\Delta) < 0$ implies that $g(x) \leq 0$ for all $x \in (x_2, \Delta]$. Hence, $f'_q(x) \leq 0$ holds for all $x \in [x_1, \Delta]$, which implies that the function $f_q(x)$ is decreasing in $[x_1, \Delta]$. We can conclude that $f_q(x)$ achieves minimum at the extreme point $\Delta$ since $f_q(\Delta) < f_q(0)$ (*cf.* Lemma C.2), and a maximum at $y_{\max} \in (0, x_1)$. We thus proved item *(ii)* of this lemma.

- Suppose now that $g(x_2) > 0$. We investigate the behavior of the function in four regions: $\mathcal{R}_1 = [0, x_1]$, $\mathcal{R}_2 = [x_1, \Delta/2]$, $\mathcal{R}_3 = [\Delta/2, x_2]$, $\mathcal{R}_4 = [x_2, \Delta]$. Note that Lemma C.2 implies that the maximizer must lie in $[0, \Delta/2) = \mathcal{R}_1 \cup \mathcal{R}_2$ and the minimizer in $[\Delta/2, \Delta] = \mathcal{R}_3 \cup \mathcal{R}_4$. We already discussed that $f_q(x)$ is unimodal on $\mathcal{R}_1$ with a maximizer $y_{\max} \in (\xi, x_1)$ for an arbitrarily small $\xi > 0$. We can discard $\mathcal{R}_2$ since we also discussed that $g(x) < 0$ for $x \in \mathcal{R}_2$ (*i.e.*, the function $f_q(x)$ is decreasing in $\mathcal{R}_2$). Since $g(\Delta/2) < 0$, $g(x_2) > 0$, and $g'(x) > 0$, $x \in \mathcal{R}_3$ simultaneously hold, there exists a local minimum $y_{\min} \in (\Delta/2, x_2)$ in $\mathcal{R}_3$. Analogously, for $\mathcal{R}_4$, we have $g(x_2) > 0$ and $g(\Delta) < 0$, and since there is no root of $g'(x) = 0$ in $(x_2, \Delta)$, the function $g(x)$ is decreasing in $\mathcal{R}_4$. This allows us to conclude that there exists a local maximizer of $f_q(x)$ in $\mathcal{R}_4$, but we can ignore this as we only seek minimizers in this region. In this case, the minimizer of $\mathcal{R}_4$ is one of the extreme points ($x_2$ or $\Delta$), but since we already found a local minimum in $\mathcal{R}_3$, the only candidate minimizer for $f_q(x)$ is $\Delta$. We thus proved item *(iii)* of this lemma.

Since we proved each of the three items in the statement of this lemma, we conclude the proof. $\qquad\square$

### C.8. Proof of Lemma 4.5

Let $f_{q,\max}(\sigma) = \max_{x \in [0,\Delta]} f_q(x; \sigma)$ and $f_{q,\min}(\sigma) = \min_{x \in [0,\Delta]} f_q(x; \sigma)$. We consider the three cases studied in Lemma 4.4 and show that $f_{q,\max}(\sigma)/f_{q,\min}(\sigma)$ is decreasing in each of these cases.

For the *first case*, suppose $\Delta^2 \leq 4\sigma^2$. In this case, Lemma 4.4 implies:

$$f_{q,\min}(\sigma) = f_q(\Delta; \sigma) = \frac{1}{c} e^\varepsilon \exp\left(-\frac{\Delta^2}{2\sigma^2}\right). \tag{35}$$

Its derivative satisfies:

$$\frac{\mathrm{d}}{\mathrm{d}\sigma} f_{q,\min}(\sigma) = \frac{1}{c} e^\varepsilon \frac{\Delta^2}{\sigma^3} \exp\left(-\frac{\Delta^2}{2\sigma^2}\right). \tag{36}$$

Moreover, for $x_{\max} \in (0, \Delta/2)$ being the solution that satisfies $f_{q,\max}(\sigma) = f_q(x_{\max}; \sigma)$, we have

$$\frac{\partial}{\partial x}\Big|_{x=x_{\max}} f_q(x; \sigma) = \frac{e^\varepsilon}{c}\left(-\frac{x_{\max}}{\sigma^2}\right) \exp\left(-\frac{x_{\max}^2}{2\sigma^2}\right) + \frac{1}{c}\left(-\frac{x_{\max} - \Delta}{\sigma^2}\right) \exp\left(-\frac{(x_{\max} - \Delta)^2}{2\sigma^2}\right) = 0$$

$$\implies -e^\varepsilon x_{\max} \exp\left(-\frac{x_{\max}^2}{2\sigma^2}\right) + (\Delta - x_{\max}) \exp\left(-\frac{(x_{\max} - \Delta)^2}{2\sigma^2}\right) = 0$$

$$\implies e^\varepsilon \frac{x_{\max}}{\Delta - x_{\max}} \exp\left(-\frac{x_{\max}^2}{2\sigma^2}\right) = \exp\left(-\frac{(x_{\max} - \Delta)^2}{2\sigma^2}\right),$$

where the first equality follows from the definition of the partial derivative of $f_q(x; \sigma)$ in $x$ as derived in Section C.7 (which is equal to 0; *cf.* Lemma 4.4), the implication that follows multiplies the expression by the positive quantity $c\sigma^2$, and the final implication divides the expression by the positive quantity $\Delta - x_{\max}$. The final equality can be substituted in the

definition of $f_q(x_{\max}; \sigma)$ to conclude:

$$f_q(x_{\max}; \sigma) = \frac{e^\varepsilon}{c} \exp\left(-\frac{x_{\max}^2}{2\sigma^2}\right) + \frac{1}{c} \exp\left(-\frac{(x_{\max} - \Delta)^2}{2\sigma^2}\right),$$

$$= \frac{1}{c} e^\varepsilon \frac{\Delta}{\Delta - x_{\max}} \exp\left(-\frac{x_{\max}^2}{2\sigma^2}\right). \tag{37}$$

Note that $x_{\max}$ is a function of $\sigma$ since it is the solution that maximizes $f_q(x; \sigma)$ on $(0, \Delta/2)$. However, it can be treated as constant when computing the derivative of $f_q(x_{\max}; \sigma)$ in $\sigma$, since $x_{\max}$ is a local maximum. To better reflect this, use a generic $x(\sigma)$ instead of $x_{\max}$, we can use the chain rule and show:

$$\frac{d}{d\sigma} f_q(x(\sigma); \sigma) = \underbrace{\frac{\partial}{\partial x} f_q(x; \sigma)}_{=0 \text{ when } x(\sigma) = x_{\max}} \cdot \frac{d}{d\sigma} x(\sigma) + \frac{\partial}{\partial \sigma'} f_q(x(\sigma); \sigma'),$$

and evaluating this derivative at $x(\sigma) = x_{\max}$ gives us

$$\frac{d}{d\sigma}\bigg|_{x(\sigma) = x_{\max}} f_q(x(\sigma); \sigma) = \frac{\partial}{\partial \sigma} f_q(x_{\max}; \sigma),$$

that is, $x_{\max}$ can be treated as a constant. We thus have:

$$\frac{d}{d\sigma} f_{q,\max}(\sigma) = \frac{\partial}{\partial \sigma} f_q(x_{\max}; \sigma) = \frac{1}{c} e^\varepsilon \frac{\Delta}{\Delta - x_{\max}} \frac{x_{\max}^2}{\sigma^3} \exp\left(-\frac{x_{\max}^2}{2\sigma^2}\right). \tag{38}$$

We will conclude the desired result for this case if we can show

$$\left(\frac{f_{q,\max}(\sigma)}{f_{q,\min}(\sigma)}\right)' \le 0. \tag{39}$$

We have

$$\left(\frac{f_{q,\max}(\sigma)}{f_{q,\min}(\sigma)}\right)' \propto f'_{q,\max}(\sigma) f_{q,\min}(\sigma) - f'_{q,\min}(\sigma) f_{q,\max}(\sigma)$$

$$= \left[\frac{1}{c} e^\varepsilon \frac{\Delta}{\Delta - x_{\max}} \frac{x_{\max}^2}{\sigma^3} \exp\left(-\frac{x_{\max}^2}{2\sigma^2}\right) \cdot \frac{1}{c} e^\varepsilon \exp\left(-\frac{\Delta^2}{2\sigma^2}\right)\right] -$$

$$\left[\frac{1}{c} e^\varepsilon \frac{\Delta^2}{\sigma^3} \exp\left(-\frac{\Delta^2}{2\sigma^2}\right) \cdot \frac{1}{c} e^\varepsilon \frac{\Delta}{\Delta - x_{\max}} \exp\left(-\frac{x_{\max}^2}{2\sigma^2}\right)\right]$$

$$\propto \frac{\Delta}{\Delta - x_{\max}} \exp\left(-\frac{\Delta^2}{2\sigma^2}\right) \exp\left(-\frac{x_{\max}^2}{2\sigma^2}\right) \left[\frac{x_{\max}^2}{\sigma^3} - \frac{\Delta^2}{\sigma^3}\right]$$

$$\propto \frac{x_{\max}^2}{\sigma^3} - \frac{\Delta^2}{\sigma^3} \le 0.$$

Here, the first step uses the quotient rule and ignores the positive quantity $f_{q,\min}(\sigma)^{-2}$, the second step uses equations (35), (36), (37) and (38), the third step ignores the positive term $(e^\varepsilon/c)^2$, the fourth step ignores the exponential terms (as they are positive) as well as $\Delta/(\Delta - x_{\max})$ (since $x_{\max} \in (0, \Delta/2)$), and the final inequality follows from $x_{\max} \in (0, \Delta/2)$.

For the *second case*, suppose $\Delta^2 > 4\sigma^2$ and $g(x_2) \le 0$ where $x_2$ and $g$ are defined as in Lemma 4.4. Similarly to the first case the minimizer of $f_{q(x;\sigma)}$ over $[0, \Delta]$ (for fixed $\sigma$) is $x = \Delta$. The maximizer is the local maximizer in the unimodal region $(0, x_1)$, which is a smaller region than the one in the first case ($x_1 \le \Delta/2$), hence the above analysis applies analogously here.

For the *third case*, suppose $\Delta^2 > 4\sigma^2$ and $g(x_2) > 0$ where $x_2$ and $g$ are defined as in Lemma 4.4. Let $x_{\max}$ be the solution that satisfies $f_{q,\max}(\sigma) = f_q(x_{\max}; \sigma)$, and $x_{\min}$ be the solution that satisfies $f_{q,\min}(\sigma) = f_q(x_{\min}; \sigma)$. From Lemma 4.4 it follows that $x_{\max}$ is the local maximum of $f_q(x; \sigma)$ over the unimodal region $x \in (0, x_1)$ for $x_1$ as defined in Lemma 4.4. On the other hand, $x_{\min}$ is either equal to $\Delta$, or it is the local minimum of $f_q(x; \sigma)$ over the unimodal region $x \in (\Delta/2, x_2)$ for $x_2$ as defined in Lemma 4.4. If $x_{\min} = \Delta$, the proof follows analogously as in the first two cases. Hence, for the rest of the proof, we assume without loss of generality that:

*(i)* $x_{\min}$ is the local minimum of $f_q(x;\sigma)$ over the unimodal region $x \in (\Delta/2, x_2)$ for $x_2 = \Delta/2 + \sqrt{\Delta^2 - 4\sigma^2}/2$;

*(ii)* $x_{\max}$ is the local maximum of $f_q(x;\sigma)$ over the unimodal region $x \in (0, x_1)$ for $x_1 = \Delta/2 - \sqrt{\Delta^2 - 4\sigma^2}/2$.

Since $f_{q,\max}(\sigma)$ and $f_{q,\min}(\sigma)$ are positive functions, a sufficient condition for $f_{q,\max}(\sigma)/f_{q,\min}(\sigma)$ being decreasing in $\sigma$ is $\log(f_{q,\max}(\sigma)/f_{q,\min}(\sigma))$ being decreasing in $\sigma$. We have:

$$\frac{\mathrm{d}}{\mathrm{d}\sigma} \log\left(\frac{f_{q,\max}(\sigma)}{f_{q,\min}(\sigma)}\right) = \frac{\mathrm{d}}{\mathrm{d}\sigma}\left[\log(f_{q,\max}(\sigma)) - \log(f_{q,\min}(\sigma))\right]$$

$$= \underbrace{\frac{f'_{q,\max}(\sigma)}{f_{q,\max}(\sigma)}}_{(i)} - \underbrace{\frac{f'_{q,\min}(\sigma)}{f_{q,\min}(\sigma)}}_{(ii)}.$$

We will thus show that $(i) \leq (ii)$ and conclude the desired result. From (38) and (37), we have:

$$\frac{f'_{q,\max}(\sigma)}{f_{q,\max}(\sigma)} = \frac{\frac{1}{c}e^\varepsilon \frac{\Delta}{\Delta - x_{\max}} \frac{x_{\max}^2}{\sigma^3} \exp\left(-\frac{x_{\max}^2}{2\sigma^2}\right)}{\frac{1}{c}e^\varepsilon \frac{\Delta}{\Delta - x_{\max}} \exp\left(-\frac{x_{\max}^2}{2\sigma^2}\right)} = \frac{x_{\max}^2}{\sigma^3}.$$

With analogous calculations, we have:

$$\frac{f'_{q,\min}(\sigma)}{f_{q,\min}(\sigma)} = \frac{x_{\min}^2}{\sigma^3}.$$

Since $x_{\min} \in (\Delta/2, \Delta/2 + \sqrt{\Delta^2 - 4\sigma^2}/2)$ and $x_{\max} \in (0, \Delta/2 - \sqrt{\Delta^2 - 4\sigma^2}/2)$, we always have $x_{\max} < x_{\min}$, hence $(i) \leq (ii)$.

We conclude the proof of monotonicity since we showed that, regardless of the case, the derivative of $f_{q,\max}(\sigma)/f_{q,\min}(\sigma)$ is negative. We can ignore the end-points of the cases since all the functions and their derivatives investigated in this proof are continuous in.

Finally, to see that the function of interest is upper bounded by $e^\varepsilon$ at $\sigma = \sqrt{\Delta^2/(2\varepsilon)}$, note that

$$\max_{x \in [0,\Delta]} f_q(x;\sigma) \leq \frac{1}{\sqrt{2\pi}\sigma} \max\left\{\max_{x \in [0,\Delta]} \exp\left(-\frac{x^2}{2\sigma^2}\right), \max_{x \in [0,\Delta]} \exp\left(-\frac{(|x| - \Delta)^2}{2\sigma^2}\right)\right\}$$

$$= \frac{1}{\sqrt{2\pi}\sigma} \max\left\{\exp\left(-\frac{0}{2\sigma^2}\right), \exp\left(-\frac{0}{2\sigma^2}\right)\right\}$$

$$= \frac{1}{\sqrt{2\pi}\sigma}$$

where the inequality holds since $f_q(\cdot;\sigma)$ is the convex combination of two distributions, hence the maximum of $f_q(\cdot;\sigma)$ is upper bounded by one of the maxima of each of these two distributions, the first equality holds by solving the inner max-problems, and the second equality holds by noting that both inner max problems have the same optimal value. Analogously, we can show

$$\min_{x \in [0,\Delta]} f_q(x;\sigma) \geq \frac{1}{\sqrt{2\pi}\sigma} \min\left\{\min_{x \in [0,\Delta]} \exp\left(-\frac{x^2}{2\sigma^2}\right), \min_{x \in [0,\Delta]} \exp\left(-\frac{(|x| - \Delta)^2}{2\sigma^2}\right)\right\}$$

$$= \frac{1}{\sqrt{2\pi}\sigma} \min\left\{\exp\left(-\frac{\Delta^2}{2\sigma^2}\right), \exp\left(-\frac{\Delta^2}{2\sigma^2}\right)\right\}$$

$$= \frac{1}{\sqrt{2\pi}\sigma} \exp\left(-\frac{\Delta^2}{2\sigma^2}\right).$$

We thus have

$$\frac{\max\limits_{x\in[0,\Delta]} f_q(x;\sigma)}{\min\limits_{x\in[0,\Delta]} f_q(x;\sigma)} \leq \frac{\dfrac{1}{\sqrt{2\pi}\sigma}}{\dfrac{1}{\sqrt{2\pi}\sigma}\exp\left(-\dfrac{\Delta^2}{2\sigma^2}\right)}$$

$$= \exp\left(\frac{\Delta^2}{2\sigma^2}\right)$$

and plugging in $\sigma = \Delta/\sqrt{2\varepsilon}$ sets the above quantity to $e^\varepsilon$.

### C.9. Plots for Section 4

We visualize the monotonicity of the constraint of $\sigma_1$ in (4a), as discussed in Lemma 4.3. We visualize a case where $\Delta = 1$, and plot two cases: the left plot uses $(\varepsilon, \delta) = (1, 0.1)$ and the right plot uses $(\varepsilon, \delta) = (1, 0.25)$. In the first plot, we observe that the function of interest is monotonically increasing on $(0, \sqrt{2(\varepsilon - \log \delta)}\Delta/\varepsilon)$ and is nonnegative beyond this region. The point where the function hits 0 is also visualized as $\sigma_1$. In the second case, we have $e^\varepsilon + 2 \geq \delta^{-1}$, and thus the function is nonnegative (hence $\sigma_1 = 0$).

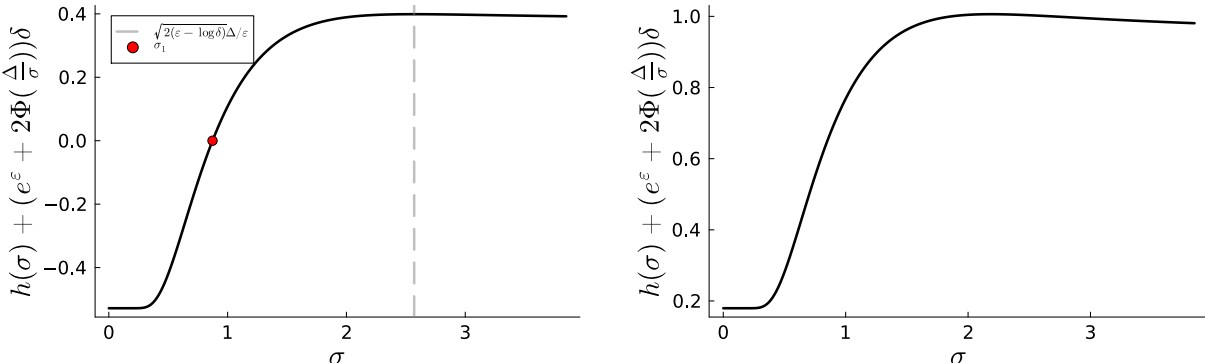

*Figure 5.* The constraint of (4a) as a function of $\sigma$ under $\Delta = 1$, $(\varepsilon, \delta) = (1, 0.1)$ (left) and $(\varepsilon, \delta) = (1, 0.25)$ (right).

Finally, in Figure 6, we visualize Lemmas 4.4 and 4.5 that are used in finding $\sigma_2$ in (5a), for an instance with $\Delta = 1$ and $\varepsilon = 1$. The left figure plots $f_q(x;\sigma)$ on $x \in [0, \Delta]$ for fixed $\sigma = 0.2$ and marks the maximizer and minimizer of this function over this range. We can observe what Lemma 4.4 proves: the maximizer $x_{\max}$ is the local maximum of the function on the unimodal region $(0, x_1)$, and the minimizer $x_{\min}$ is the local minimum of the function on the unimodal region $(\Delta/2, x_2)$. The term $f_q(x_{\max};\sigma)/f_q(x_{\min};\sigma) - e^\varepsilon$, as a function of $\sigma$, is monotonically decreasing as proved in Lemma 4.5, which is displayed in the right figure.

### C.10. Proof of Theorem 4.6

For the correctness of the algorithm, note that the $\sigma_1$ computed coincides with the $\sigma_1$ in Theorem 4.2, which is found via a bisection search method due to the monotonicity presented in Lemma 4.3. This lemma also gives an upper bound for $\sigma$ as beyond this upper bound the function is nonnegative, and Algorithm 3 accordingly restricts the search to a bounded region. Note that the if condition in this algorithm also ensures that we apply the bisection search method only in the case if the function is not nonnegative everywhere, as presented by Lemma 4.3. The $\sigma_2$ computed in this algorithm, similarly, coincides with $\sigma_2$ of Theorem 4.2. This value can be found via a bisection search restricted to $(0, \sqrt{\Delta^2/(2\varepsilon)})$ due to Lemma 4.5. Each evaluation of the bisection search relies on Lemma 4.4 where the if-else if-else condition shows the cases presented in this lemma (the cases are elaborated in the unabridged version of the lemma). Finally, Algorithm 3 returns $\sigma$ as the maximum of $\sigma_1$ and $\sigma_2$, and therefore from Theorem 4.2 we conclude the feasibility of the returned value.

For the runtime complexity, first note that we treat the tolerances in the bisection and golden-section search methods as constants. Note that one needs to be careful in implementation since numerically the tolerances should be reflected in the

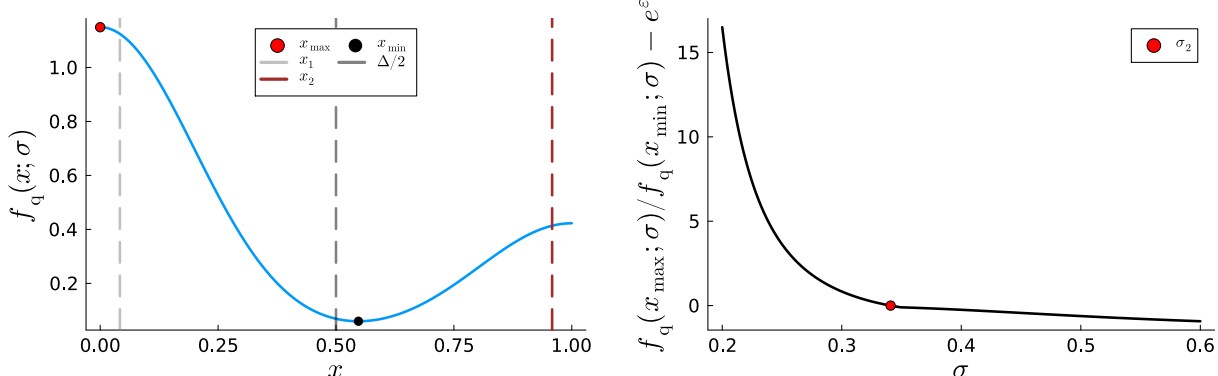

*Figure 6.* (Left) The density $f_q(x;\sigma)$ on $x \in [0,\Delta]$ for $\Delta = 1$, $\varepsilon = 1$, and $\sigma = 0.2$. The notation $x_1, x_2, x_{\max}$ and $x_{\min}$ comes from Algorithm 4. (Right) The term $f_q(x_{\max};\sigma)/f_q(x_{\min};\sigma) - e^\varepsilon$ displayed as a function of $\sigma$ where $\Delta = 1$ and $\varepsilon = 1$. The point that sets this expression to 0 is the $\sigma_2$ of (5a).

presentation of $\delta$ to ensure feasibility. We also set $\Delta = 1$ without loss of generality in the runtime analysis. To see this, let $f_q^\Delta(\cdot;\sigma)$ denote the quasi-Gaussian mixture density with sensitivity $\Delta$. Under the change of variables $y = x/\Delta$ and $\lambda = \sigma/\Delta$, we have

$$f_q^\Delta(\Delta y; \Delta\lambda) = \frac{1}{\Delta}f_q^1(y;\lambda).$$

Thus, the conditions defining $\sigma_1$ and $\sigma_2$ depend on $\sigma$ and $\Delta$ only through $\lambda = \sigma/\Delta$. The scale for the original sensitivity-$\Delta$ problem is then recovered as $\sigma = \Delta\lambda$.

Now, note that Algorithm 3 has two bisection search methods implemented. The first one has a search radius of $\sqrt{2(\varepsilon - \log \delta)}/\varepsilon$ after the normalization $\Delta = 1$ and the evaluation of $\psi_1$ is constant assuming computation of $\Phi(\cdot)$ is constant. The complexity of the first bisection search is thus:

$$\mathcal{O}\left(\log\left(1 + \frac{1}{\varepsilon}\right) + \log\left(1 + \log\frac{1}{\delta}\right)\right).$$

Here the "1+" terms are used only to write a bound that remains nonnegative in the moderate and low-privacy regimes considered in the paper. In particular, when $\varepsilon$ is large, the dependence on the search radius should be treated as constant rather than through the negative quantity $\log(1/\varepsilon)$; similarly, $\log(1 + \log(1/\delta))$ avoids a regime-specific discussion of when $\log\log(1/\delta)$ is nonnegative.

On the other hand, the second bisection search has a radius of $\sqrt{1/(2\varepsilon)}$ after the normalization $\Delta = 1$, where each iteration relies on Algorithm 4 which, in the worst case, calls a golden-section search method whose search region is a subset of $(0, 1)$. The computation of $f_q(x;\sigma)$ during the golden-section search evaluations is constant, hence the golden section has constant complexity under our fixed-tolerance convention. The complexity of the bisection search is overall:

$$\mathcal{O}\left(\log\left(1 + \frac{1}{\varepsilon}\right)\right).$$

The sum of the complexities of both bisection search methods concludes the complexity presented in the statement of this theorem.

### C.11. Proof of Proposition 4.7

Throughout this proof, we fix an arbitrary query sensitivity $\Delta > 0$. The proof of Proposition 4.7 relies on the following auxiliary lemma.

**Lemma C.3.** *For any $\delta \in (0, 1/2)$ and $\varepsilon > 0$ satisfying $\varepsilon \geq \log(\delta^{-1} - 2)$, setting $\sigma = \Delta/\sqrt{2\varepsilon}$ for the quasi-Gaussian mixture distribution gives a feasible $(\varepsilon, \delta)$-DP mechanism. Consequently, the $l_2$-loss of the quasi-Gaussian mixture*

*mechanism satisfies*

$$L_2^{\mathrm{QG}}(\varepsilon, \delta) \ \leq \ \frac{\Delta^2}{2\varepsilon} + \left(4 + \frac{1}{\varepsilon}\right) \Delta^2 e^{-\varepsilon}.$$

*In particular, if $\varepsilon \geq \max\{1, \log(\delta^{-1} - 2)\}$, then we have*

$$L_2^{\mathrm{QG}}(\varepsilon, \delta) \ \leq \ \frac{\Delta^2}{2\varepsilon} + 5\Delta^2 e^{-\varepsilon}.$$

*Proof.* We will first argue that the quasi-Gaussian mixture mechanism with $\sigma = \Delta/\sqrt{2\varepsilon}$ is feasible for $(\varepsilon, \delta)$-DP and subsequently study the $l_2$-loss of this mechanism.

Recall that, by Theorem 4.2, the quasi-Gaussian mixture mechanism is $(\varepsilon, \delta)$-DP whenever $\sigma \geq \max\{\sigma_1, \sigma_2\}$. Since we have $e^\varepsilon + 2 \geq \delta^{-1}$ from the statement of this lemma, Lemma 4.3 implies that we can set $\sigma_1 = 0$ without loss of generality and therefore we have $\sigma \geq \sigma_1$. To conclude feasibility, we thus need to show $\sigma \geq \sigma_2$ holds. This immediately follows from Lemma 4.5 which shows that we always have $\sigma_2 \leq \Delta/\sqrt{2\varepsilon} = \sigma$. The quasi-Gaussian mixture mechanism with $\sigma = \Delta/\sqrt{2\varepsilon}$ therefore satisfies $(\varepsilon, \delta)$-DP, which implies

$$L_2^{\mathrm{QG}}(\varepsilon, \delta) \ \leq \ \mathbb{E}[\tilde{X}^2],$$

where $\tilde{X}$ denotes a quasi-Gaussian random variable with $\sigma = \Delta/\sqrt{2\varepsilon}$. In the remainder of this proof, we bound $\mathbb{E}[\tilde{X}^2]$.

By Definition 4.1, the quasi-Gaussian density can be written as

$$f_{\mathrm{q}}(x; \sigma) \ = \ \alpha_\sigma g_\sigma(x) + (1 - \alpha_\sigma) h_\sigma(x),$$

where $g_\sigma$ is the density of $\mathcal{N}(0, \sigma^2)$, the parameter $\alpha_\sigma$ is defined as

$$\alpha_\sigma := \frac{e^\varepsilon}{e^\varepsilon + 2\Phi(\Delta/\sigma)} \in [0, 1],$$

and the density $h_\sigma$ is defined as

$$h_\sigma(x) := \frac{1}{\sqrt{2\pi}\sigma \, 2\Phi(\Delta/\sigma)} \exp\left(-\frac{(|x| - \Delta)^2}{2\sigma^2}\right).$$

Thus, $\tilde{X}$ is the mixture of a zero-mean Gaussian with standard deviation $\sigma$ as well as another random variable $\tilde{Y}$ with density $h_\sigma(\cdot)$, with weights $\alpha_\sigma$ and $1 - \alpha_\sigma$, respectively. We thus exploit the mixture structure and derive

$$\mathbb{E}[\tilde{X}^2] \ = \ \alpha_\sigma \sigma^2 + (1 - \alpha_\sigma) \mathbb{E}[\tilde{Y}^2]. \tag{40}$$

We next rewrite the term $\mathbb{E}[\tilde{Y}^2]$ in (40) as follows

$$\begin{aligned}
\mathbb{E}[\tilde{Y}^2] \ &= \ \frac{1}{\sqrt{2\pi}\sigma \, \Phi(\Delta/\sigma)} \int_0^\infty x^2 \exp\left(-\frac{(x - \Delta)^2}{2\sigma^2}\right) \mathrm{d}x \\
&= \ \frac{1}{\Phi(\Delta/\sigma)} \underbrace{\int_{-\Delta}^\infty (y + \Delta)^2 \frac{1}{\sqrt{2\pi}\sigma} \exp\left(-\frac{y^2}{2\sigma^2}\right) \mathrm{d}y}_{:=I_\sigma} \\
&= \ \frac{I_\sigma}{\Phi(\Delta/\sigma)}
\end{aligned}$$

where the first equality exploits the symmetry of the absolute value in the density $h_\sigma(x)$, the second equality applies a change of variable $y = x - \Delta$, and the final equality uses the definition of $I_\sigma$.

Plugging this back in (40) yields

$$\mathbb{E}[\tilde{X}^2] \ = \ \alpha_\sigma \sigma^2 + (1 - \alpha_\sigma) \frac{I_\sigma}{\Phi(\Delta/\sigma)} \ = \ \alpha_\sigma \sigma^2 + \frac{2I_\sigma}{e^\varepsilon + 2\Phi(\Delta/\sigma)} \ \leq \ \alpha_\sigma \sigma^2 + 2e^{-\varepsilon} I_\sigma, \tag{41}$$

where the second equality follows from the definition of $\alpha_\sigma$ and the inequality holds by lower bounding the denominator. Finally, by expanding the integration region in the definition of $I_\sigma$, we obtain

$$I_\sigma \leq \int_{-\infty}^{\infty} (y+\Delta)^2 \frac{1}{\sqrt{2\pi}\sigma} \exp\left(-\frac{y^2}{2\sigma^2}\right) dy = \sigma^2 + \Delta^2 \leq \sigma^2 + 2\Delta^2,$$

and plugging this bound in (41) concludes

$$\mathbb{E}[\tilde{X}^2] \leq \alpha_\sigma \sigma^2 + 2e^{-\varepsilon}(\sigma^2 + 2\Delta^2) \leq \sigma^2 + 2e^{-\varepsilon}(\sigma^2 + 2\Delta^2),$$

where the final inequality holds since $\alpha_\sigma \in [0,1]$. Plugging our selection of $\sigma = \Delta/\sqrt{2\varepsilon}$ yields the desired result:

$$\mathbb{E}[\tilde{X}^2] \leq \frac{\Delta^2}{2\varepsilon} + 2e^{-\varepsilon}\left(\frac{\Delta^2}{2\varepsilon} + 2\Delta^2\right) = \frac{\Delta^2}{2\varepsilon} + \left(4 + \frac{1}{\varepsilon}\right)\Delta^2 e^{-\varepsilon}.$$

This proves the first displayed bound. If $\varepsilon \geq 1$, then $4 + 1/\varepsilon \leq 5$, which gives the second displayed bound. $\qquad\square$

We are now ready to prove Proposition 4.7.
**Proof of Proposition 4.7.** Auxiliary Lemma B.3 yields

$$L_2^{\mathrm{AG}}(\varepsilon, \delta) = \frac{\Delta^2}{2\varepsilon} + c_\delta \frac{\Delta^2}{\varepsilon^{3/2}} + \mathcal{O}(\varepsilon^{-2}),$$

where for simplicity of presentation we set

$$c_\delta := \frac{|\Phi^{-1}(\delta)|}{\sqrt{2}} > 0.$$

The remainder term $\mathcal{O}(\varepsilon^{-2})$ is asymptotically smaller than the positive correction term $\varepsilon^{-3/2}$. In particular, since $\varepsilon^{-2} = o(\varepsilon^{-3/2})$, there exists $\varepsilon_1(\delta) > 0$ such that, for all $\varepsilon \geq \varepsilon_1(\delta)$,

$$\mathcal{O}(\varepsilon^{-2}) \geq -\frac{c_\delta}{2}\frac{\Delta^2}{\varepsilon^{3/2}}.$$

Therefore, for all $\varepsilon \geq \varepsilon_1(\delta)$, we obtain

$$L_2^{\mathrm{AG}}(\varepsilon, \delta) \geq \frac{\Delta^2}{2\varepsilon} + \frac{c_\delta}{2}\frac{\Delta^2}{\varepsilon^{3/2}}.$$

Moreover, Auxiliary Lemma C.3 shows that, for all $\varepsilon > 0$ satisfying $\varepsilon \geq \max\{1, \log(\delta^{-1} - 2)\}$, we have

$$L_2^{\mathrm{QG}}(\varepsilon, \delta) \leq \frac{\Delta^2}{2\varepsilon} + 5\Delta^2 e^{-\varepsilon}.$$

Since $e^{-\varepsilon} = o(\varepsilon^{-3/2})$, there exists $\varepsilon_2(\delta) > 0$ such that, for all $\varepsilon \geq \varepsilon_2(\delta)$ we have

$$5e^{-\varepsilon} < \frac{c_\delta}{2}\varepsilon^{-3/2}.$$

Now if we define

$$\varepsilon_0 := \max\left\{\varepsilon_1(\delta),\ \varepsilon_2(\delta),\ 1,\ \log(\delta^{-1} - 2)\right\},$$

then the above derivations conclude that for all $\varepsilon \geq \varepsilon_0$, we obtain

$$L_2^{\mathrm{QG}}(\varepsilon, \delta) \leq \frac{\Delta^2}{2\varepsilon} + 5\Delta^2 e^{-\varepsilon} < \frac{\Delta^2}{2\varepsilon} + \frac{c_\delta}{2}\frac{\Delta^2}{\varepsilon^{3/2}} \leq L_2^{\mathrm{AG}}(\varepsilon, \delta).$$

This proves the desired strict inequality.

*Table 3.* $K \in [10]$ values that attain the best $l_1$-loss for the multi-Gaussian mixture. The star sign indicates instances where quasi-Gaussian outperforms multi-Gaussian with $K = 1$.

| $\delta \downarrow \mid \varepsilon \rightarrow$ | 0.1 | 0.25 | 0.5 | 0.75 | 1 | 2 | 3 | 4 | 5 | 10 |
|---|---|---|---|---|---|---|---|---|---|---|
| $5 \cdot 10^{-7}$ | **1** | 19 | 20 | 18⋆ | 14⋆ | 13⋆ | 14⋆ | 9⋆ | 14⋆ | 9⋆ |
| $10^{-6}$ | 9 | 20 | 20 | 18⋆ | 14⋆ | 12⋆ | 18⋆ | 14⋆ | 13⋆ | 9⋆ |
| $5 \cdot 10^{-6}$ | 4 | 20 | 20 | 17 | 16 | 17 | 18 | 18⋆ | 14⋆ | 9⋆ |
| $10^{-5}$ | 2 | 20 | 20 | 16 | 16 | 18 | 18 | 14⋆ | 14⋆ | 9⋆ |
| $5 \cdot 10^{-5}$ | **1** | 20 | 18 | 16 | 16 | 17 | 18 | 18 | 14⋆ | 4 |
| $10^{-4}$ | 4 | 20 | 16 | 17 | 18 | 8 | 19 | 19 | 14⋆ | 9 |
| $5 \cdot 10^{-4}$ | **1** | 20 | 13 | 9 | 18 | 8 | 19 | 9 | 9⋆ | 9 |
| $10^{-3}$ | **1** | 20 | 12 | 8 | 18 | 8 | 19 | 9 | 9⋆ | 9 |
| $5 \cdot 10^{-3}$ | 20 | 13 | 8 | 6 | 13 | 8 | 19 | 9⋆ | 9⋆ | 9 |
| 0.01 | 18 | 11 | 7 | 5 | 4 | 8 | 9 | 9⋆ | 14 | 9 |
| 0.02 | 13 | 8 | 6 | 4 | 4 | 8 | 9⋆ | 9 | 14 | 9 |
| 0.05 | 7⋆ | 5 | 4 | 3 | 3 | 8 | 9 | 9 | 14 | 9 |
| 0.1 | 4⋆ | 3 | 2 | 2 | 2 | 8⋆ | 9 | 9 | 14 | 9 |
| 0.15 | 2⋆ | 2 | 2 | **1** | 2 | 9 | 9 | 9 | 14 | 9 |
| 0.25 | **1** | 1⋆ | **1** | **1** | **1** | 9 | 9 | 9 | 14 | 9 |

*Table 4.* Improvement (% of $l_2$-loss) of multimodal Gaussians over the best unimodal Gaussian.

| $\delta \downarrow \mid \varepsilon \rightarrow$ | 0.1 | 0.25 | 0.5 | 0.75 | 1 | 2 | 3 | 4 | 5 | 10 |
|---|---|---|---|---|---|---|---|---|---|---|
| $5 \cdot 10^{-7}$ | NA | 4.05% | 30.27% | 83.54% | 83.05% | 86.57% | 93.38% | 94.54% | 96.97% | 96.93% |
| $10^{-6}$ | NA | 5.49% | 33.94% | 82.59% | 82.11% | 85.93% | 93.05% | 94.30% | 94.77% | 98.55% |
| $5 \cdot 10^{-6}$ | NA | 6.41% | 44.22% | 86.25% | 87.22% | 91.03% | 94.52% | 96.79% | 98.19% | 99.93% |
| $10^{-5}$ | NA | 7.22% | 57.79% | 85.08% | 86.16% | 90.68% | 94.15% | 96.59% | 98.16% | 99.94% |
| $5 \cdot 10^{-5}$ | NA | 11.04% | 79.64% | 81.48% | 82.89% | 88.38% | 93.08% | 95.98% | 97.93% | 99.75% |
| $10^{-4}$ | 0.16% | 13.26% | 77.24% | 79.24% | 81.32% | 87.07% | 92.43% | 95.65% | 97.74% | 99.77% |
| $5 \cdot 10^{-4}$ | NA | 25.69% | 69.28% | 72.53% | 75.24% | 83.51% | 90.57% | 94.84% | 97.30% | 99.73% |
| $10^{-3}$ | NA | 57.84% | 64.23% | 68.58% | 71.41% | 81.38% | 89.48% | 94.29% | 97.04% | 99.71% |
| $5 \cdot 10^{-3}$ | 1.13% | 42.08% | 50.19% | 55.34% | 58.95% | 73.88% | 85.80% | 92.50% | 96.18% | 99.64% |
| 0.01 | 26.14% | 30.59% | 40.01% | 48.12% | 53.14% | 68.78% | 83.43% | 91.37% | 95.68% | 99.61% |
| 0.02 | 19.77% | 26.77% | 26.20% | 40.59% | 39.50% | 61.66% | 80.21% | 89.88% | 95.00% | 99.56% |
| 0.05 | 23.32% | 21.65% | 14.70% | 25.82% | 22.16% | 46.75% | 73.80% | 87.01% | 93.72% | 99.48% |
| 0.1 | 25.14% | 19.12% | 3.26% | 22.87% | 15.91% | 31.45% | 65.96% | 83.62% | 92.26% | 99.39% |
| 0.15 | 14.19% | 12.75% | 9.57% | 1.17% | 1.57% | 35.73% | 59.17% | 80.78% | 91.05% | 99.33% |
| 0.25 | 6.27% | 12.26% | 9.46% | 20.98% | 13.65% | 15.03% | 46.33% | 75.55% | 88.88% | 99.21% |

## D. Omitted details in numerical experiments

### D.1. Best $K$ values for Table 1

Table 3 reports the values of $K \in \{1, \ldots, 20\}$ that yield the smallest $l_1$-loss for the multi-Gaussian mechanism across the 150 combinations of $(\varepsilon, \delta)$ considered. The results show that all values within the grid are being chosen, demonstrating the importance of tuning $K$. Although both the multi-Gaussian mechanism with $K = 1$ and the quasi-Gaussian mechanism produce distributions with three modes, there is no clear hierarchy between them. In fact, entries marked with a red star indicate cases where the quasi-Gaussian mechanism outperforms the multi-Gaussian mechanism with $K = 1$.

### D.2. $l_2$-Counterpart of Table 1

Table 4 is the $l_2$-counterpart of Table 1. We note that the improvements for the $l_2$-loss are qualitatively better compared to the improvements for the $l_1$-loss. We observe that in $143/150$ of the cases, multimodality provides strict improvements (expected $l_2$-loss) over unimodality. Across all instances, the mean improvement is $61.86\%$ (sd $35.35$) and the median improvement is $79.44\%$.

*Table 5.* Improvement (% of $l_1$-loss) of the quasi-Gaussian mixture mechanism (green) over the analytic Gaussian mechanism (red).

| $\delta \downarrow \mid \varepsilon \rightarrow$ | 0.1 | 0.25 | 0.5 | 0.75 | 1 | 2 | 3 | 4 | 5 | 10 |
|---|---|---|---|---|---|---|---|---|---|---|
| $5 \cdot 10^{-7}$ | -0.51% | -0.77% | -1.76% | -2.30% | -2.47% | -1.18% | 1.70% | 4.76% | 7.56% | 20.58% |
| $10^{-6}$ | -0.53% | -1.12% | -1.84% | -2.37% | -2.54% | -1.17% | 1.86% | 5.07% | 8.04% | 22.13% |
| $5 \cdot 10^{-6}$ | -0.58% | -1.26% | -2.07% | -2.54% | -2.71% | -1.14% | 2.30% | 5.98% | 9.41% | 27.11% |
| $10^{-5}$ | -0.56% | -1.31% | -2.15% | -2.63% | -2.79% | -1.12% | 2.54% | 6.46% | 10.15% | 30.34% |
| $5 \cdot 10^{-5}$ | -0.67% | -1.44% | -2.35% | -2.86% | -3.02% | -1.03% | 3.28% | 7.93% | 12.39% | 60.60% |
| $10^{-4}$ | -0.70% | -1.51% | -2.45% | -2.98% | -3.13% | -0.96% | 3.70% | 8.78% | 13.69% | 59.40% |
| $5 \cdot 10^{-4}$ | -0.79% | -1.70% | -2.74% | -3.30% | -3.44% | -0.69% | 5.11% | 11.56% | 18.10% | 56.14% |
| $10^{-3}$ | -0.84% | -1.79% | -2.89% | -3.47% | -3.59% | -0.49% | 6.01% | 13.35% | 21.08% | 54.48% |
| $5 \cdot 10^{-3}$ | -0.91% | -2.04% | -3.30% | -3.92% | -3.99% | 0.41% | 9.54% | 20.79% | 37.45% | 49.75% |
| 0.01 | -0.88% | -2.13% | -3.49% | -4.14% | -4.16% | 1.20% | 12.41% | 27.97% | 56.58% | 47.20% |
| 0.02 | -0.72% | -2.15% | -3.67% | -4.35% | -4.29% | 2.56% | 17.43% | 53.82% | 53.28% | 44.20% |
| 0.05 | 0.01% | -1.89% | -3.76% | -4.49% | -4.20% | 6.73% | 45.29% | 47.67% | 47.63% | 39.27% |
| 0.1 | 1.46% | -1.08% | -3.42% | -4.11% | -3.32% | 21.24% | 37.64% | 41.25% | 41.83% | 34.41% |
| 0.15 | 3.00% | 0.01% | -2.69% | -3.08% | -1.23% | 23.61% | 31.70% | 36.35% | 37.46% | 30.86% |
| 0.25 | -2.86% | 3.21% | 1.90% | 6.11% | 3.63% | 10.04% | 21.68% | 28.22% | 30.29% | 25.22% |

## D.3. Optimality gaps closed

In the numerical experiments, we compared our mechanisms against the analytic Gaussian mechanism in terms of expected losses of the noise. We also investigated the optimality gaps of Gaussian mechanisms compared to the best possible loss any additive noise mechanism can attain under $(\varepsilon, \delta)$-DP. To this end, we use the work of Selvi et al. (2025), who propose a sequential numerical optimization approach to construct a distribution, where in each iteration the upper bounds (the losses attained by the distribution) and associated lower bounds provably converge. Since in practice this convergence can take a significant amount of time, we run the algorithm until the upper and lower bounds are almost equal with a tolerance of $10^{-3}$, and then use the lower bound as a tight estimation of the ideal loss one can attain under $(\varepsilon, \delta)$-DP. If we denote by $o$ this estimation, then the analytic Gaussian mechanism has an optimality gap of $100\% \cdot (a - o)/o$. Under this analysis, we find that the mean and median gaps closed are $67.67\%$ (sd $34.78$) and $85.47\%$, respectively. Moreover, for $\varepsilon = 5$, we close more than $90\%$ of the optimality gap for any value of $\delta > 0$, reaching to $99.72\%$ for $\varepsilon = 10$.

## D.4. Quasi-Gaussian experiments

Recall that the hyperparameter-free quasi-Gaussian mechanism (3) simply takes the convex combination of a zero-mean Gaussian density with a quasi-Gaussian density. Table 5 demonstrates that it can offer significant improvements in the expected $l_1$-loss in the targeted low-privacy regimes. The entries of the table represent $100\% \cdot (a - q)/\max\{a, q\}$ where $q$ and $a$ are the expected losses attained by the quasi-Gaussian and the analytic Gaussian mechanisms, respectively. We observe that in $75/150$ cases (exactly half) the quasi-Gaussian mechanism improves over the analytic Gaussian mechanism. Across all instances, the mean improvement is $17.62\%$ (sd $17.62$). The $l_2$-counterpart of this table is available in Table 6 with qualitatively better results, where in $76/150$ cases the quasi-Gaussian mechanism improves over the analytic Gaussian mechanism. Across all instances, the mean improvement is $14.96\%$ (sd $25.93$).

We note that there is an intuitive explanation for why the improvement offered by the quasi-Gaussian mixture mechanism is less substantial in higher-privacy regimes than the multi-Gaussian mixture mechanism. When $\varepsilon$ is small, DP forces near-indistinguishability, so feasible mechanisms must concentrate around zero. The quasi-Gaussian mixture mechanism has a rigid template, combining a zero-mean Gaussian with side mass around $\pm\Delta$, and therefore cannot adapt to this regime: feasibility requires increasing $\sigma$, which eliminates much of the gain. In contrast, the multi-Gaussian mixture mechanism can vary modality $K$, allowing it to perform strongly across a broader range of privacy regimes. The quasi-Gaussian mixture mechanism should therefore be viewed as a lightweight surrogate that is particularly designed for low-privacy regimes.

## D.5. Runtimes of experiments

Over the $(\varepsilon, \delta)$ grid of privacy parameters that we consider in this work (*e.g.*, Tables 1 and 5), the average runtime of the analytic Gaussian mechanism is $0.16$ seconds with a maximum of $0.28$ seconds across all privacy parameters (which is the fastest method, as expected). The average/maximum runtimes of the quasi-Gaussian mechanism are $0.87/1.27$ seconds, which, as designed, is closer to the analytic Gaussian mechanism. The multi-Gaussian mixture mechanism, however,

*Table 6.* Improvement (% of $l_2$-loss) of the quasi-Gaussian mixture mechanism (green) over the analytic Gaussian mechanism (red).

| $\delta\downarrow\mid\varepsilon\rightarrow$ | 0.1 | 0.25 | 0.5 | 0.75 | 1 | 2 | 3 | 4 | 5 | 10 |
|---|---|---|---|---|---|---|---|---|---|---|
| $5\cdot10^{-7}$ | -0.88% | -1.23% | -2.98% | -3.94% | -4.22% | -1.86% | 3.53% | 9.25% | 14.47% | 36.92% |
| $10^{-6}$ | -0.91% | -1.92% | -3.13% | -4.05% | -4.33% | -1.84% | 3.82% | 9.84% | 15.34% | 39.35% |
| $5\cdot10^{-6}$ | -0.99% | -2.15% | -3.54% | -4.33% | -4.61% | -1.76% | 4.65% | 11.50% | 17.79% | 46.86% |
| $10^{-5}$ | -0.95% | -2.23% | -3.66% | -4.47% | -4.75% | -1.71% | 5.10% | 12.38% | 19.10% | 51.46% |
| $5\cdot10^{-5}$ | -1.13% | -2.44% | -3.98% | -4.84% | -5.11% | -1.51% | 6.46% | 15.02% | 23.01% | 84.45% |
| $10^{-4}$ | -1.18% | -2.55% | -4.15% | -5.03% | -5.29% | -1.37% | 7.23% | 16.50% | 25.21% | 83.49% |
| $5\cdot10^{-4}$ | -1.31% | -2.83% | -4.59% | -5.53% | -5.76% | -0.86% | 9.75% | 21.29% | 32.47% | 80.73% |
| $10^{-3}$ | -1.36% | -2.97% | -4.80% | -5.78% | -5.98% | -0.48% | 11.31% | 24.28% | 37.14% | 79.24% |
| $5\cdot10^{-3}$ | -1.36% | -3.24% | -5.35% | -6.41% | -6.53% | 1.14% | 17.26% | 35.96% | 59.67% | 74.71% |
| 0.01 | -1.18% | -3.26% | -5.55% | -6.67% | -6.73% | 2.50% | 21.84% | 46.18% | 79.05% | 72.08% |
| 0.02 | -0.66% | -3.08% | -5.65% | -6.85% | -6.82% | 4.77% | 29.44% | 74.39% | 75.75% | 68.82% |
| 0.05 | 1.30% | -1.99% | -5.29% | -6.66% | -6.38% | 11.36% | 62.07% | 67.11% | 69.54% | 63.06% |
| 0.1 | 4.98% | 0.52% | -3.85% | -5.38% | -4.46% | 31.45% | 50.72% | 58.54% | 62.41% | 56.91% |
| 0.15 | 8.86% | 3.54% | -1.59% | -2.87% | -0.43% | 30.00% | 40.89% | 51.34% | 56.55% | 52.12% |
| 0.25 | -0.52% | 12.26% | 9.46% | 14.04% | 5.36% | 2.92% | 22.28% | 38.11% | 46.01% | 43.98% |

*Table 7.* Improvement (% of $l_2$-loss) of multimodal Gaussians (green) over the best approximate DP benchmark (red).

| $\delta\downarrow\mid\varepsilon\rightarrow$ | 0.1 | 0.25 | 0.5 | 0.75 | 1 | 2 | 3 | 4 | 5 | 10 |
|---|---|---|---|---|---|---|---|---|---|---|
| $5\cdot10^{-7}$ | -86.10% | -86.99% | -83.54% | -33.75% | -38.09% | -29.51% | 24.86% | 34.37% | 61.69% | 53.06% |
| $10^{-6}$ | -84.88% | -85.75% | -81.35% | -33.00% | -37.36% | -28.55% | 25.44% | 35.05% | 37.21% | 78.83% |
| $5\cdot10^{-6}$ | -81.15% | -82.46% | -73.49% | 0.94% | 3.51% | 23.35% | 49.01% | 68.02% | 80.84% | 99.14% |
| $10^{-5}$ | -79.04% | -80.53% | -61.74% | 1.14% | 3.59% | 25.89% | 49.13% | 68.07% | 81.73% | 99.20% |
| $5\cdot10^{-5}$ | -72.30% | -73.91% | -0.13% | 1.33% | 3.35% | 23.32% | 49.31% | 68.01% | 82.39% | 97.21% |
| $10^{-4}$ | -68.29% | -69.77% | -0.09% | 0.54% | 4.66% | 22.04% | 49.01% | 68.01% | 82.10% | 97.59% |
| $5\cdot10^{-4}$ | -55.82% | -51.70% | -0.80% | 0.36% | 3.25% | 21.62% | 48.90% | 69.04% | 82.36% | 97.59% |
| 0.001 | -48.93% | -1.54% | -2.06% | 0.31% | 1.87% | 21.20% | 48.78% | 69.01% | 82.35% | 97.59% |
| 0.005 | -31.35% | -0.20% | -1.20% | -1.16% | -1.16% | 18.72% | 48.07% | 68.81% | 82.30% | 97.59% |
| 0.01 | -1.42% | -6.08% | -5.48% | -1.30% | 0.69% | 16.33% | 47.47% | 68.64% | 82.40% | 97.59% |
| 0.02 | -7.39% | -2.64% | -12.75% | -0.63% | -9.37% | 12.33% | 46.31% | 68.33% | 82.31% | 97.59% |
| 0.05 | -7.89% | -4.38% | -14.41% | -6.30% | -15.18% | 2.56% | 43.49% | 67.53% | 82.10% | 97.59% |
| 0.1 | -12.09% | -9.07% | -20.68% | -1.17% | -11.38% | -4.78% | 39.65% | 66.45% | 81.81% | 97.59% |
| 0.15 | -26.17% | -17.80% | -14.17% | -19.72% | -19.48% | 11.71% | 36.26% | 65.51% | 81.56% | 97.59% |
| 0.25 | -34.19% | -20.47% | -13.68% | 4.59% | -1.24% | -0.58% | 30.12% | 63.83% | 81.11% | 97.59% |

requires one to tune $K\in\mathbb{N}$ and $\eta>0$. In our experiments, we fixed $\eta=0.01$ and chose the best $K$ over the grid $K\in[20]$. The multi-Gaussian mixture mechanism with $K=10$, for instance, has average/maximum runtimes of $6.64/32.15$ seconds, which is the slowest mechanism.

### D.6. Data entries for Figure 2

We note that, across all benchmark mechanisms, while the order of the methods depend on the exact combination of $(\varepsilon,\delta)$, the *winning* approach is always either the truncated Laplace mechanism or the Tulap mechanism. With this in mind, Table 2 compared the performance of our best multimodal mechanism (computed similarly to Table 1) with the best out of the truncated Laplace mechanism and the Tulap mechanisms. The $l_2$-counterpart of the same experiment is shared in Table 7.

### D.7. Price of privacy in ML (unabridged)

Thus far, we have compared privacy mechanisms based on their expected losses. We now evaluate the performance of these mechanisms in a machine learning task implemented under $(\varepsilon,\delta)$-DP. Specifically, we compare the in- and out-of-sample performances of privacy-preserving linear classifiers. Each classifier is trained under the same $(\varepsilon,\delta)$-DP guarantee and differs only in the mechanism used to sample the additive noise: A-G (analytic Gaussian), Q-G (quasi-Gaussian mixture), or M-G (multi-Gaussian mixture). This experiment allows us to examine whether privacy mechanisms with smaller noise levels reduce the price of privacy in a machine learning setting, following a similar experiment conducted for the numerical optimization-based algorithm (Selvi et al., 2025, §5.3).

More specifically, we work with a dataset $\{(\boldsymbol{x}^i,y^i)\}_{i=1}^N$ where each data-point $i$ has feature vector $\boldsymbol{x}^i\in\mathbb{R}^d$ and binary

*Table 8.* Mean in- and out-of-sample errors of privacy-preserving classifiers under different mechanisms. Bold: best DP mechanism; Italics: second-best DP mechanism in the row.

| Dataset description | | | In-sample errors | | | | Out-of-sample errors | | | |
|---|---|---|---|---|---|---|---|---|---|---|
| Dataset | $N$ | $d$ | A–G | Q–G | M–G | PCD | A–G | Q–G | M–G | PCD |
| adult | 45,222 | 57 | 22.53% | **22.50%** | *22.53%* | 22.52% | 22.18% | **22.16%** | *22.18%* | 22.18% |
| annealing | 898 | 42 | 16.45% | *16.34%* | **16.19%** | 16.24% | 18.32% | *18.18%* | **18.07%** | 18.13% |
| breast-cancer | 683 | 26 | 5.40% | *5.35%* | **5.23%** | 4.20% | 7.28% | *7.25%* | **7.07%** | 5.59% |
| colon-cancer | 62 | 2,000 | *12.82%* | 13.68% | **6.09%** | 0.00% | *30.67%* | 30.92% | **29.63%** | 29.00% |
| dermatology | 366 | 98 | 25.42% | **24.84%** | *25.02%* | 13.27% | 26.74% | *26.46%* | **26.16%** | 15.66% |
| ecoli | 336 | 8 | 15.45% | **14.99%** | *15.14%* | 6.75% | 14.19% | *13.77%* | **13.46%** | 4.41% |
| spect | 160 | 23 | *21.85%* | 21.96% | **18.68%** | 16.55% | *34.43%* | 34.59% | **31.97%** | 30.63% |

label $y^i \in \{-1, 1\}$. We train an $l_1$-regularized logistic classifier $\boldsymbol{h} \in \mathbb{R}^d$ which minimizes the empirical logistic loss $\frac{1}{N} \sum_{i=1}^{N} \log(1 + \exp(-y^i \cdot \boldsymbol{h}^\top \boldsymbol{x}^i)) + \lambda \cdot \|\boldsymbol{h}\|_1$, where $\lambda > 0$ is the regularization parameter. We then predict the label of a new instance with features $\boldsymbol{\chi}$ to be the class that maximizes $[1 + \exp(-y \cdot \boldsymbol{h}^\top \boldsymbol{\chi})]^{-1}$. We train the logistic classifier via the proximal coordinate descent method (Friedman et al., 2010; Parikh & Boyd, 2014; Richtárik & Takáč, 2014) in $T = 100$ iterations with $P = \lceil d/4 \rceil$ proximal updates per iteration, using a regularization parameter $\lambda = 10^{-8}$. The DP counterpart of this algorithm is due to the work of (Mangold et al., 2022) who inject noise into each proximal update step (which is a scalar update and thus our mechanisms are applicable), and the post processing property (Dwork & Roth, 2014, §2.1) of differential privacy guarantees that the resulting classifier is also differentially private. The proximal operator has sensitivity $\Delta = 2$ assuming the feature vectors are pre-processed so that $\|\boldsymbol{x}^i\|_\infty \leq 1$, $i \in [N]$. We target the global guarantee of $(\varepsilon = 5, \delta = 1/N^2)$-DP and compute the required per-iteration noise either using Theorem 3.1 of (Mangold et al., 2022) or via numerical composition (Gopi et al., 2021), which we implement in Python for our mechanisms since the quasi-Gaussian mixture mechanism does not satisfy the zCDP property verbatim.

Table 8 reports the mean in- and out-of-sample errors across 500 random training-set/test-set $(80/20\%)$ splits of various datasets. We use 6 of the most popular UCI classification datasets (Dua & Graff, 2017); additionally, since the proximal coordinate descent method is commonly used for datasets where $d \gg N$, we also include the colon-cancer dataset of LIBSVM (Chang & Lin, 2011). All datasets are included in our implementation supplements. Note that we *(i)* convert labels with more than two classes into a binary label via binning (if the label is ordinal) or distinguishing the majority class from all others (if the label is nominal); *(ii)* apply dummy encoding for the nominal features. Our earlier experiments suggest that $K = 10$ is a good rule-of-thumb for the multi-Gaussian mixture mechanism. We observe in Table 8 that either the quasi-Gaussian mixture mechanism or the multi-Gaussian mixture mechanism achieves the lowest error across datasets. We also note that the analytic Gaussian mechanism never improves upon the multi-Gaussian mixture mechanism. However, it outperforms the quasi-Gaussian mixture mechanism in some instances, conforming with our earlier findings.

