# OpenReview forum: "Mind the Gap: Mixtures of Gaussians in Approximate Differential Privacy"
_ICML.cc/2026/Conference — ICML 2026 regular_

### Official Review · Reviewer_gMFk · 2026-03-04

**Soundness:** 3
**Presentation:** 2
**Significance:** 3
**Originality:** 3
**Overall Recommendation:** 4
**Confidence:** 3

**Summary:**

Differential privacy mechanisms typically involve a trade-off between privacy and accuracy. While the Gaussian mechanism does not always achieve the optimal privacy-utility trade-off, it has desirable properties such as unbounded support and tight compositions. Motivated by prior work that formulates the privacy-utility trade-off as an optimization problem leading to multi-modal noise distributions, this paper proposes a mixture-of-Gaussians additive noise mechanism. The proposed approach aims to improve empirical performance, particularly in moderate and low privacy regimes (e.g., when $\epsilon \geq 1$). The authors also introduce a hyperparameter-free quasi-Gaussian mixture variant and provide algorithms to determine parameters for both mechanisms such that the desired privacy constraints are satisfied. Experimental results on scalar queries and machine learning tasks demonstrate improvements over baseline methods.

**Compliance With Llm Reviewing Policy:**

Affirmed.

**Final Justification:**

The author's rebuttal provides additional discussion on the theoretical justification and implementation details, which addresses my concerns. I decide to raise my score from 3 (Weak Reject) to 4 (Weak Accept).

**Key Questions For Authors:**

How does the running time of the proposed parameter search procedure compare with baseline methods for scalar outputs? For example, a cited prior work [1] reports that their optimization problem can be solved in approximately two seconds. A direct comparison of computational cost would help clarify the practical efficiency of the proposed approach.

[1] Selvi, Aras, Huikang Liu, and Wolfram Wiesemann. "Differential privacy via distributionally robust optimization." Operations Research 74, no. 1 (2026): 356-376.

**Limitations:**

Yes

**Strengths And Weaknesses:**

Strengths:
- The paper provides a detailed analysis of the procedures used to determine the parameters of the proposed mechanisms under privacy constraints.
- Empirical evaluations on scalar mechanisms are comprehensive and include comparisons with several baselines.

Weaknesses:
- The primary evidence supporting the proposed mechanisms comes from empirical experiments. The paper does not provide theoretical guarantees demonstrating that the proposed mixture mechanisms achieve strictly better privacy-utility trade-offs in specific privacy regimes.
- The paper motivates the work by noting that Gaussian mechanisms are often “preferred in large-scale practice, especially in moderate-privacy regimes where composition, tail behavior, and simplicity outweigh asymptotic optimality.” However, it is not entirely clear why the proposed approach should remain restricted to Gaussian-like mechanisms rather than considering more general noise distributions that may offer stronger theoretical optimality or better performance.
- Although the paper suggests that Gaussian-like mechanisms are preferred due to simplicity, the proposed mechanisms do not appear significantly simpler or more computationally efficient than existing approaches.

---

> ### Author Rebuttal · Authors · 2026-03-30
>
> We sincerely thank you for your rigorous assessment. Your feedback led us to establish formal theoretical guarantees that were missing from the initial submission, and that in our view significantly strengthen the paper. We have worked diligently to address all of your concerns as thoroughly as possible.
>
> ## Weakness 1: No theoretical guarantees for strict improvement
>
> This is an excellent point. Prompted by your feedback, we were able to prove the following result formally.
>
> **Theorem (informal).** For any fixed $\delta\in(0,1/2)$ and $\Delta>0$, there exists $\varepsilon_0(\delta)$ such that for all $\varepsilon\geq\varepsilon_0(\delta)$, both the quasi-Gaussian and $K=1$ multi-Gaussian mechanisms attain strictly smaller $l_2$-loss than the optimal analytic Gaussian mechanism.
>
> *Proof sketch.* We set $\sigma^*=\Delta/\sqrt{2\varepsilon}$ and show that both mechanisms satisfy $(\varepsilon,\delta)$-DP whenever $\delta\geq p_\varepsilon:=1/(e^\varepsilon+2)$. At this noise scale, our mechanisms achieve $l_2$-loss $\Delta^2/(2\varepsilon) + O(e^{-\varepsilon})$, sharing the same dominant term as the analytic Gaussian, whose $l_2$-loss is $\Delta^2/(2\varepsilon) + O(\varepsilon^{-3/2})$. A large-$\varepsilon$ expansion of the Balle-Wang's analytical calibration equation reveals the analytic Gaussian incurs a much larger $O(\varepsilon^{-3/2})$ correction. Since $e^{-\varepsilon}=o(\varepsilon^{-3/2})$, the mixture mechanisms strictly dominate for all sufficiently large $\varepsilon$.
>
> Importantly, this is not merely asymptotic. We characterize exact non-asymptotic regimes via the analytic Gaussian calibration function $\Gamma_\varepsilon(u)$, which gives the minimum $\delta$ required for a Gaussian with $l_2$-loss $u\Delta^2$ to satisfy $(\varepsilon,\delta)$-DP. Since $\Gamma_\varepsilon$ is strictly decreasing in $u$, any target $\delta<\Gamma_\varepsilon(u)$ rules out every Gaussian with loss at most $u\Delta^2$. The non-asymptotic regime is strong precisely in the low-privacy settings that motivate our paper. At $\varepsilon=5$, our feasibility threshold is $p_5\approx 6.65\times 10^{-3}$, while $\Gamma_5(u_{\mathrm{MG}})\approx 0.308$ and $\Gamma_5(u_{\mathrm{QG}})\approx 0.308$, so every $\delta\in[6.65\times 10^{-3}, 0.308)$ yields strict improvement. At $\varepsilon=10$, this expands to $[4.54\times 10^{-5}, 0.413)$. This matches the appendix numerics: Table 3 shows 88-98% $l_2$-improvement at $\varepsilon=5$, and Appendix D.3 reports $>90$% of the optimality gap closed for every $\delta>0$ at $\varepsilon=5$, reaching 99.72% at $\varepsilon=10$.
>
> We are grateful for this push and we believe the new theorem is a major addition.
>
> ## Weakness 2: Why restrict to Gaussian-like mechanisms?
>
> Our multi-Gaussian mechanism preserves the same zCDP parameter $\rho=\Delta^2/(2\sigma^2)$ as the standard Gaussian (Corollary 3.6), so composition remains lossless under zCDP. Non-Gaussian alternatives (truncated Laplace, Tulap) lack Gaussian tails and do not compose as tightly. Our goal is precisely to close the utility gap while retaining these properties.
>
> ## Weakness 3: Not simpler or more efficient than existing approaches
>
> We agree and now clarify in the paper the following: the simplicity of our mechanisms lies in their analytical characterization: closed-form PDFs, CDFs, exact moments, and straightforward sampling (Algorithm 5 for the quasi-Gaussian, for instance, requires only standard Gaussian and uniform samples). This contrasts with, for instance, Selvi et al. (2025), whose numerical solutions lack closed-form representations and have bounded supports. Parameter tuning is a one-time offline cost; the practical advantage surfaces at deployment, auditing, and accounting.
>
> ## Question 1: Runtime comparison with Selvi et al.
>
> We have optimized our implementation (grid reduction, memoization of the privacy shortfall function, within-instance multithreading), yielding a $9.48\times$ speedup across the evaluated instances. On the experimental grid from Selvi et al. (which uses computationally easier/larger $\delta$ values), our method averages 0.16s/instance vs. their reported 2.2s.
>
> However, we emphasize that runtime is not our primary contribution. The central point is distilling structural insights from generic numerical optimization into explicit Gaussian-mixture families with provable guarantees. While parameter calibration is a one-time offline step, deployment requires reliability: our families provide analytical tractability for downstream tasks. The multi-Gaussian mechanism preserves exactly the same zCDP composition as the analytic Gaussian, so practitioners retain familiar composition theory while gaining substantial one-shot improvements (median 62% in $l_1$-loss, up to 99% of the optimality gap closed). Our mechanisms also retain Gaussian tails and unbounded support, which means they avoid the brittleness of bounded-support alternatives and integrate seamlessly into existing DP toolkits.

---

> > ### Author Rebuttal · Reviewer_gMFk · 2026-04-03
> >
> > Thanks for your response. It would be great to inlcude the theorem and implementation details in the revised version. I will adjust my score accordingly.

---

> > > ### Author Response · Authors · 2026-04-04
> > >
> > > Dear Reviewer gMFk,
> > >
> > > We are thrilled to hear that your concerns have been adequately addressed. Thank you also for increasing your score! We would like to kindly note that the score change is not yet reflected on OpenReview. May we ask whether you had a chance to update your score?
> > >
> > > That said, we will include the new theorem and implementation details in the revised version as you suggested. We initially wanted to include the whole derivation with an anonymous link. However, ICML policy states that we can only show figures with an anonymous link. To this end, here we instead summarize our new theory and implementation improvements in more formal detail.
> > > --- ---
> > > *Theorem 1.* Fix $\delta\in(0,1/2)$ and $\Delta>0$. Let
> > > $l_2^{\mathrm{AG}}(\varepsilon,\delta,\Delta)$,
> > > $l_2^{\mathrm{QG}}(\varepsilon,\delta,\Delta)$, and
> > > $l_2^{\mathrm{MG}}(\varepsilon,\delta,\Delta)$
> > > denote the $l_2$-losses of the analytic Gaussian mechanism,
> > > the quasi-Gaussian mechanism, and the $K=1$ multi-Gaussian
> > > mechanism, respectively. Then there exists
> > > $\varepsilon_0=\varepsilon_0(\delta)$ such that for all
> > > $\varepsilon\ge \varepsilon_0$,
> > > $$
> > > l_2^{\mathrm{QG}}(\varepsilon,\delta,\Delta)
> > > <
> > > l_2^{\mathrm{AG}}(\varepsilon,\delta,\Delta),
> > > \qquad
> > > l_2^{\mathrm{MG}}(\varepsilon,\delta,\Delta)
> > > <
> > > l_2^{\mathrm{AG}}(\varepsilon,\delta,\Delta).
> > > $$
> > >
> > > The proof relies on three auxiliary lemmas.
> > >
> > > *Lemma 1.* Whenever $\varepsilon\ge \log(\delta^{-1}-2)$, the choice
> > > $\sigma=\Delta/\sqrt{2\varepsilon}$ is feasible for the
> > > quasi-Gaussian mechanism, and hence
> > > $$
> > > l_2^{\mathrm{QG}}(\varepsilon,\delta,\Delta)
> > > \le
> > > \frac{\Delta^2}{2\varepsilon}
> > > +
> > > \Bigl(4+\frac{1}{\varepsilon}\Bigr)\Delta^2 e^{-\varepsilon}.
> > > $$
> > > In particular, if
> > > $\varepsilon\ge \max\\{1,\log(\delta^{-1}-2)\\}$, then
> > > $$
> > > l_2^{\mathrm{QG}}(\varepsilon,\delta,\Delta)
> > > \le
> > > \frac{\Delta^2}{2\varepsilon}
> > > +
> > > 5\Delta^2 e^{-\varepsilon}.
> > > $$
> > >
> > > *Lemma 2.* Whenever $\varepsilon\ge \log(\delta^{-1}-2)$, the choice
> > > $\sigma=\Delta/\sqrt{2\varepsilon}$ is feasible for the
> > > multi-Gaussian mechanism with $K=1$, and hence
> > > $$
> > > l_2^{\mathrm{MG}}(\varepsilon,\delta,\Delta)
> > > \le
> > > \frac{\Delta^2}{2\varepsilon}
> > > +
> > > 2\Delta^2 e^{-\varepsilon}.
> > > $$
> > >
> > > *Lemma 3.* Fix $\delta\in(0,1/2)$ and $\Delta>0$. Let $z_\delta:=|\Phi^{-1}(\delta)|$, and let $\sigma_{\mathrm{AG}}(\varepsilon,\delta,\Delta)$ denote the smallest feasible standard deviation of the analytic Gaussian mechanism. Then, as $\varepsilon\to\infty$,
> > >
> > > $$ \sigma_{\mathrm{AG}}(\varepsilon,\delta,\Delta) = \frac{\Delta}{\sqrt{2\varepsilon}} + \frac{\Delta z_\delta}{2\varepsilon} + O(\varepsilon^{-3/2}). $$
> > >
> > > Thus,
> > >
> > > $$ l_2^{\mathrm{AG}}(\varepsilon,\delta,\Delta) = \sigma_{\mathrm{AG}}^2(\varepsilon,\delta,\Delta) = \frac{\Delta^2}{2\varepsilon} + \frac{z_\delta\Delta^2}{\sqrt{2}\,\varepsilon^{3/2}} + O(\varepsilon^{-2}). $$
> > > --- ---
> > > *Implementation details*
> > > 1. *$\varphi$/step-cap reductions:* In the low- and moderate-$\varepsilon$ solvers, we kept the same privacy-checking routine but reduced the maximum number of $\varphi$ grid points used to scan shifts in $[0,\Delta]$. This removes a large amount of over-resolution in the original implementation, so each $\sigma$ evaluation uses far fewer quadrature calls.
> > > 2. *shortfall($\sigma$) memoization:* In the outer Brent/bisection search over $\sigma$, we cache previously computed values of the privacy shortfall function instead of recomputing them whenever the root finder revisits the same $\sigma$. This is an exact implementation improvement: it does not change the algorithmic criterion, only avoids redundant work.
> > > 3. *within-instance multithreading:* For the expensive low-privacy regimes, we parallelized the per-$\varphi$ privacy checks across CPU threads within a single instance, rather than only parallelizing across benchmark instances. This leaves the mathematical computation unchanged, but reduces wall-clock time by evaluating many independent shift checks simultaneously.
> > >
> > > Total speedup (across all 150 instances from the paper): 9.48$\times$
> > >
> > > Average per-instance speedup: 25.15$\times$

---

### Official Review · Reviewer_PAue · 2026-03-05

**Soundness:** 3
**Presentation:** 3
**Significance:** 3
**Originality:** 4
**Overall Recommendation:** 5
**Confidence:** 3

**Summary:**

The paper considers the problem of answering a scalar query with bounded sensitivity under approximate DP while minimizing the loss. The Gaussian mechanism is not optimal in this setting, but a method that numerically optimizes the loss is inefficient and has undesirable properties like bounded support. The authors propose two variants of the Gaussian mechanism that improve on the utility of the Gaussian mechanism but also are easier to implement than the numerically optimal method and retain nice properties of the Gaussian. The first is a mixture of Gaussians with means of [-K, -K+1, ..., K] with weights proportional to $e^-{|i|\epsilon}$ for the Gaussian centered at i. While the worst case is not necessarily that the sensitivity of a query is equal to its bound, the authors show that by sweeping a grid of possible sensitivities they can achieve an almost-tight privacy analysis, whose error can be absorbed into the delta term. The second is a 'quasi-Gaussian' mechanism which adds a mixture of a Gaussian centered at 0, and a quasi-Gaussian with pdf proportional to $\exp(-(|x|-\Delta)^2/2\sigma^2)$. For the privacy analysis, they show sufficient conditions for privacy, and show these conditions have nice properties that allow one to do bisection searches over divergences between distributions to tune sigma. Empirically, the authors demonstrate the first  variant improves on the Gaussian mechanism, sometimes reducing its l1/l2-losses by over 99% in the high-epsilon setting, whereas the quasi-Gaussian mechanism is more efficient and improves by up to 84% in the high-epsilon setting, but sometimes is worse than the base Gaussian mechanism.

**Compliance With Llm Reviewing Policy:**

Affirmed.

**Final Justification:**

As mentioned in the reply to the rebuttal, my concerns have been addressed and I have increased my score accordingly. Namely, the authors have explained that the Selvi et al. paper also has a runtime increasing in 1/delta, which in my view was the main limitation of the authors' approach, and that the authors are trying much harder delta values compared to the Selvi et al. paper. This confirms to me that the paper is a significant empirical improvement over past work, whereas it was unclear at the time of the initial review.

**Key Questions For Authors:**

Clarifications on the following points in Weaknesses would help with evaluating the paper:
* Based on the runtimes of your experiments, do you have an estimate for at what point does your Algorithm 2 become more inefficient than the Selvi et al. method? (Or, does the Selvi et al. method also slow down as delta gets smaller, and your method is always more efficient?)
* Validate that the error in numerical integration in Algorithm 2 can be absorbed into the delta term.

**Limitations:**

The authors are up-front about the limited generalization to higher dimensions and the runtimes of their algorithms.

**Strengths And Weaknesses:**

Strengths:
* Very strong empirical results; able to close 85% of the gap b/t the Gaussian mechanism and a lower bound on average with a much more efficient method than the optimal one.
* Authors' methods have some nice qualitative properties the numerically optimized method does not since they are built on Gaussians. e.g likely more amenable to composition, easy to sample from in practice.
* The proofs for the privacy analyses of the methods are pretty technically interesting contributions, involves some nice manipulation of divergence functions and methods for calculating them.

Weaknesses:
* The runtime for Algorithm 2 is linear in $1/\delta$. At this point, Monte Carlo methods for estimating delta with formal privacy guarantees (e.g. https://arxiv.org/abs/2304.07927) start to become viable (this paper has some chance of an empty output, but this can be corrected for). So while Theorem 3.4 is a nice technical result, it doesn't offer much in terms of asymptotic efficiency gains over black-box methods.
* The 1/delta dependence means for sufficiently small delta (e.g. the US Census used $10^{-10}$, I believe the paper only sweeps down to $5 \cdot 10^{-7}$) the runtime of Algorithm 2 would actually exceed the numerically optimized method, unless the numerically optimized method also has a runtime dependence on delta; it would be good to clarify this, but this means there is a limit to the regime in which it is favorable over the numerically optimized method.
* Several minor technical suggestions:
  * Lemma 3.3 has a pretty lengthy proof, but can be proved straightforwardly using the post-processing property of DP. For the first part, we can post-process your mechanism with Gaussian variance $\sigma^2$ by adding noise with variance $\sigma'^2 - \sigma^2$ to arrive at the same mechanism with Gaussian variance $\sigma'^2$. For the second part, we can post-process the Gaussian mechanism by adding a mean sampled from the distribution over means to arrive at your mechanism.
  * In practice, the numerical integration in Algorithm 2 will require some error due to discretization. For mechanisms with a monotonic privacy loss (e.g. the Gaussian mechanism) this is not a problem since we can do the numerical integration in a way that over-estimates delta. The paper should explain how to do the integration 'safely', i.e. ensuring we always over-estimate the integral (I think the QuadGK library handles this by allowing precision of the integral, but it would be good to make it explicit; the paper should at least say e.g. we are adding the integral's  precision to the final result to ensure it is an over-estimate).
  * Presumably for the quasi-Gaussian, we can assume e.g. $\Delta = 1$ wlog because answering a sensitivity-$\Delta$ query with the quasi-Gaussian mechanism w/ noise $\sigma$ is the same as answering a sensitivity $1$ query with the quasi-Gaussian mechanism with noise $\sigma / \Delta$ and then multiplying the mechanism output by $\Delta$ afterwards. Why does Theorem 4.6 have a dependence on $\Delta$ then?

---

> ### Author Rebuttal · Authors · 2026-03-30
>
> We are grateful for your thorough and positive review, and especially for going through our proofs rigorously. We have addressed all suggestions and believe they have improved the paper.
>
> ## Weakness 1: Algorithm 2 runtime is $O(1/\delta)$; Monte Carlo methods are competitive
>
> Thank you for this challenge. We agree that Theorem 3.4, viewed in isolation, should not be presented as a major asymptotic improvement over modern Monte Carlo verifiers. The stronger calibration result is Theorem 4.6: the hyperparameter-free quasi-Gaussian family admits tuning with only logarithmic dependence on $\delta$, and Section 4 was designed precisely as the faster-to-optimize alternative. We will revise the paper to foreground this distinction and present the multi-Gaussian tuner (Algorithm 1) as a constructive tool for the richer family, not as a competing asymptotic result.
>
> We also emphasize a key operational difference than EVR-style Monte Carlo verifiers: our algorithms compute a $\sigma$ that directly certifies the mechanism offline, producing an explicit parametric family that can then be reused across arbitrarily many deployments with no runtime verification step or rejection events. This one-time calibration model aligns naturally with how practitioners deploy DP mechanisms in production.
>
> Implementation-level optimizations during the rebuttal (grid reduction, memoization, multithreading) have also yielded a $9.48\times$ empirical speedup for Algorithm 1, but we view this as complementary to the asymptotic point. The main contribution is the mechanism family itself: explicit Gaussian mixtures closing much of the utility gap while preserving zCDP composition, Gaussian tails, unbounded support, and closed-form moments.
>
> ## Weakness 2 + Question 1: $1/\delta$ dependence; crossover with Selvi et al.
>
> We agree that the naive runtime worsens as $\delta$ shrinks. However, the Selvi et al. method also slows down: decreasing $\delta$ increases the support radius of the optimized distribution, introducing more variables and exponentially more constraints in their cutting-plane LP. On the experimental grid used by Selvi et al. (which features computationally easier/larger $\delta$ values), our optimized implementation already averages 0.16s/instance vs. their reported 2.2s, so the crossover would occur at substantially smaller $\delta$ than their experiments cover.
>
> Our implementation mitigates the worst-case $1/\delta$ growth through capped grids, $\varepsilon$-dependent truncation, pruning of negligible mixture components, and regime-specific integration strategies. We do not claim to remove the $1/\delta$ theoretical bound, but these accelerations substantially reduce observed growth in practice. More importantly, for practitioners needing fast calibration at very small $\delta$, Algorithm 3 (quasi-Gaussian) provides an alternative with only logarithmic $\delta$-dependence (Theorem 4.6). We will clarify this separation of roles more explicitly.
>
> ## Minor 1: Lemma 3.3 proof via post-processing
>
> Thank you. We agree partially. For the second part (showing $\sigma=\sigma_g$ satisfies (2)), the post-processing argument works and simplifies the presentation: sample the mixture mean independently of the data and add it as post-processing to the Gaussian mechanism. Since DP is closed under post-processing, shifting, and mixtures, this immediately certifies $(\varepsilon,(1-\eta)\delta)$-DP. We will adopt this.
>
> For the first part (monotonicity in $\sigma$), there is a subtle distinction. Post-processing proves monotonicity of the true DP guarantee, but Lemma 3.3 is stated at the level of certificate (2) which is used by Algorithm 1 via $\psi(\sigma;\eta)$. Bisection in Theorem 3.4 needs monotonicity of (2) itself. Our convolution/Jensen proof establishes this stronger statement. We welcome suggestions if you see a way to obtain certificate-level monotonicity from post-processing alone.
>
> ## Minor 2 + Question 2: Safe numerical integration
>
> Sharp point! Our code already handles this via `atol = max(delta * 1e-3, 1e-12)`, tying QuadGK's absolute tolerance to $\delta$. Any failed quadrature is treated conservatively (candidate rejected, forcing larger $\sigma$). We will add a remark making this explicit.
>
> ## Minor 3: Theorem 4.6 $\Delta$-dependence
>
> Excellent observation! The $\Delta$-dependence can indeed be removed. With a substitution $y=x/\Delta$ and $\lambda=\sigma/\Delta$, the density shape depends only on $\lambda$, not on $\Delta$ separately. Specifically: $h_1$ and $h_2$ in Theorem 4.2 depend on $\sigma$ and $\Delta$ only through $\sigma/\Delta$, and the max/min ratio over $x\in[0,\Delta]$ becomes a ratio over $y\in[0,1]$ depending only on $\lambda$. Setting $\Delta=1$ wlog, the complexity of Theorem 4.6 simplifies to $O(\log(1/\varepsilon) + \log(-\log(\delta)/\varepsilon))$, giving us a cleaner bound. We will incorporate this in the revision. Thank you for pointing us to this improvement!

---

> > ### Author Rebuttal · Reviewer_PAue · 2026-03-31
> >
> > Thanks to the authors for their response. I agree with the point the authors made on Lemma 3.3 being applied to the certificate, thank you for clarifying that. I also appreciate the authors clarifying that the Selvi et al. paper also suffers from delta being too small. This strengthens my view of the paper. I am happy to adjust my score accordingly.

---

> > > ### Author Response · Authors · 2026-04-01
> > >
> > > Dear Reviewer PAue,
> > >
> > > We are pleased to hear that you are satisfied with our rebuttal. We appreciate your thorough review again. Thank you for increasing your score!
> > >
> > > Best regards,
> > > Authors of Paper 26836

---

### Official Review · Reviewer_RPmg · 2026-03-13

**Soundness:** 3
**Presentation:** 4
**Significance:** 3
**Originality:** 4
**Overall Recommendation:** 5
**Confidence:** 4

**Summary:**

The paper proposes two new additive-noise mechanisms for scalar queries under $(\varepsilon,\delta)$-DP: a multi-Gaussian mixture and a quasi-Gaussian mixture. The aim is to reduce the variance of the noise and increase accuracy for a given privacy level. Motivated by the observation that the best DP mechanism distributions appear multimodal in moderate/low-privacy regimes, it replaces a single Gaussian with a multimodal mixture whose components are shifted by the sensitivity scale and weighted geometrically. The paper gives sufficient conditions for $(\varepsilon,\delta)$-DP guarantees, efficient algorithms (with complexity analysis) to tune the scale parameter $\sigma$ to minimize expected noise, and proof that the multi-Gaussian mechanism retains the same zero-concentrated DP (zCDP) guarantees as the Gaussian mechanism. Empirically, the proposed mechanisms reduce expected noise substantially compared to the analytic Gaussian mechanism across a wide $(\varepsilon,\delta)$ grid, and the best mixture mechanism reportedly beats a set of strong non-Gaussian baselines.

**Compliance With Llm Reviewing Policy:**

Affirmed.

**Final Justification:**

Overall, I find this paper to be technically sound and original, addressing a fundamental problem in approximate differential privacy. The proposed Gaussian mixture mechanisms are creative and demonstrate meaningful empirical and theoretical improvements over the analytic Gaussian, particularly in relevant privacy regimes.

My initial concerns focused on the lack of intuition behind the construction, limited theoretical characterization of improvement regimes, and some practical issues such as parameter tuning and computational cost. The authors’ rebuttal addressed these concerns well, providing clearer intuition (e.g., the “discrete first, then smooth” view), stronger theoretical guarantees, and additional clarification on practical aspects.

While some limitations remain, such as computational overhead and the restricted setting, the overall contribution is significant. The clarity can also be improved by incorporating the explanations provided in the rebuttal into the main paper. The main remaining limitation is that the method is restricted to the scalar setting, with extension to vector-valued queries deferred to future work.

I believe the paper can be accepted, provided that the rebuttal clarifications are fully incorporated into the final version.

**Key Questions For Authors:**

1. It is not clear to me how to determine some hyperparameters of the proposed distribution. For example, how can we choose the number of mixtures K and why? Additionally, why do we have to locate the mixtures by distance of Δ? Especially in many applications, we only have access to an upper bound for Δ. In such cases, the form of the distribution might vary from the proposed near-optimal one.

2. While you empirically demonstrate that the proposed methods outperform the analytic Gaussian mechanism, is there any theorem or compelling intuition that confirms the claim of improvement in particular regimes?

3. To present a composition theorem, you employ zCDP to determine how the privacy parameters change under composition. Can you provide any explicit composition theorem in terms of ε and δ, as in the Gaussian mechanism composition theorem?

**Limitations:**

yes.

**Strengths And Weaknesses:**

Strengths:

1. Well-motivated problem: The paper targets DP systems that operate at moderate or even low privacy, where classical Gaussian calibration can be quite suboptimal.

2. Theoretical support: This is supported by theoretical analysis of sufficient conditions and composition statements.

3. Novel construction: The multimodal Gaussian mixture is a fresh and creative idea.

4. Comprehensive evaluation: The evaluation is not limited to the analytic Gaussian baseline. The paper compares against non-Gaussian baselines and reports large gains over unimodal Gaussian mechanisms in most settings.

 Weaknesses:

1. Tightness of sufficient conditions: There is no discussion about how tight the sufficient conditions are.

2. Computational cost of multi-Gaussian: The multi-Gaussian mechanism requires tuning K and η, and Algorithm 2 involves O(Δ²K²/(ηδ)) integral evaluations. For the experiments, the authors fix η=0.01 and search K∈[20], leading to runtimes up to 32 seconds (vs. 0.16s for analytic Gaussian). This may limit practical adoption where fast mechanism calibration is needed. The practicality of the multi-Gaussian mechanism is somewhat mixed.

3. Quasi-Gaussian's mixed performance: Table 4 shows the quasi-Gaussian mechanism improves over analytic Gaussian in only half the cases (75/150). For ε<1, it often performs worse (negative improvements). The authors' claim that it targets "low-privacy regimes" is accurate, but the paper does not provide intuition for why it fails in high-privacy regimes.

4. Lack of formal characterization: While the authors show empirically that the proposed method outperforms the analytic Gaussian mechanism, there is no formal theorem explicitly characterizing the regime in which this claim is valid.

5. Limited discussion of mechanism interpretation: The multi-Gaussian mixture weights are fixed as e^(-|k|ε). Why this specific weighting? Could optimizing weights (as noted in future work) yield further improvements? The current choice seems natural but is not justified beyond "this is what we tried."

---

> ### Author Rebuttal · Authors · 2026-03-30
>
> We sincerely thank you for the accurate summary and constructive feedback. Your suggestions led to significant improvements, including a new formal theorem. We address all points below.
>
> ## Weakness 4 + Question 2: Lack of formal characterization of improvement
>
> Thank you! Prompted by this and similar feedback, we now prove the following.
>
> **Theorem (informal).** For any $\delta\in(0,1/2)$ and $\Delta>0$, there exists $\varepsilon_0(\delta)$ such that for all $\varepsilon\geq\varepsilon_0(\delta)$, both the quasi-Gaussian and $K=1$ multi-Gaussian mechanisms attain strictly smaller $l_2$-loss than the optimal analytic Gaussian.
>
> *Proof sketch.* Set $\sigma^*=\Delta/\sqrt{2\varepsilon}$. Both mechanisms satisfy $(\varepsilon,\delta)$-DP whenever $\delta\geq 1/(e^\varepsilon+2)$. At this scale, they achieve $l_2$-loss $\Delta^2/(2\varepsilon) + O(e^{-\varepsilon})$, sharing the same dominant term as the analytic Gaussian, whose $l_2$-loss is $\Delta^2/(2\varepsilon) + O(\varepsilon^{-3/2})$. Since $e^{-\varepsilon}=o(\varepsilon^{-3/2})$, strict dominance follows for large $\varepsilon$.
>
> This is not merely asymptotic. We also give concrete non-asymptotic certificates. At $\varepsilon=5$, our construction is feasible for $\delta\geq p_5\approx 6.65\times 10^{-3}$, and the $l_2$-loss it achieves is strictly below the minimum attainable by any Gaussian mechanism whenever $\delta < 0.308$. Hence every $\delta\in[6.65\times 10^{-3}, 0.308)$ yields a provable strict improvement. At $\varepsilon=10$, this interval expands to $[4.54\times 10^{-5}, 0.413)$. This matches Table 3 (88-98% $l_2$-improvement at $\varepsilon=5$) and Appendix D.3 ($>90$% gap closed at $\varepsilon=5$, 99.72% at $\varepsilon=10$).
>
> ## W5 + Question 1: Design intuition
>
> The key idea is "discrete first, then smooth". Decompose $K=1$ multi-Gaussian noise as $\tilde{X}=\tilde{Z}+\tilde{G}$, where $\tilde{Z}\in\\{-a,0,a\\}$ is discrete and $\tilde{G}\sim\mathcal{N}(0,\sigma^2)$ independent. We choose $\tilde{Z}$ to satisfy DP by itself for a discrete query $q$ taking values in $\\{-\Delta,0 , \Delta\\}$. For neighbors $q(D)=0$, $q(D')=\Delta$: if $a\neq\Delta$, either $\delta=1$ or $\delta\geq 1-p_\varepsilon$, both worse than $\delta=p_\varepsilon$ at $a=\Delta$. Given $a=\Delta$, the constraint on $\\{-\Delta,0\\}$ forces $q_\varepsilon/p_\varepsilon\leq e^\varepsilon$; equality is tightest. This yields our design: means at multiples of $\Delta$, weights as $e^{-|k|\varepsilon}$.
>
> On $K$: $K=0$ recovers the analytic Gaussian, so $K\in\\{0\\}\cup[20]$ guarantees weak improvement. Our new theorem shows $K=1$ strictly improves in the characterized regimes.
>
> On $\Delta$: sensitivity is assumed known, as in all benchmarks; an upper bound yields a conservative mechanism.
>
> ## Weakness 1: Tightness of sufficient conditions
>
> We now discuss this explicitly. For the quasi-Gaussian (Thm 4.2), tuning is already tight: we find exact $\sigma_1,\sigma_2$, so the returned $\sigma$ is the tightest our framework permits. For the multi-Gaussian (Thm 3.2), the certificate depends on $\eta$, but (i) our default $\eta=0.01$ already closes up to 99% of the optimality gap, and (ii) our new theorem above provides a closed-form $\sigma^*$ guaranteeing strict improvement without numerical tuning, that is, any $\eta$-tuning can only strengthen this.
>
> ## Weakness 2: Computational cost of multi-Gaussian
>
> Based on your comment, we have further optimized our implementation via grid reduction, memoization, and multithreading, yielding a $9.48\times$ speedup. Moreover, we emphasize the separation of roles: Algorithm 1 is a constructive tuner for the richer multi-Gaussian family, while Algorithm 3 provides a hyperparameter-free quasi-Gaussian alternative with only logarithmic dependence on $\delta$ (Thm 4.6). Calibration is a one-time offline cost; at deployment, our mechanisms require only standard Gaussian and uniform samples (Algorithm 5), with no additional overhead vs. the analytic Gaussian.
>
> ## Weakness 3: Quasi-Gaussian's mixed performance for $\varepsilon<1$
>
> When $\varepsilon$ is small, DP forces near-indistinguishability, so feasible mechanisms must concentrate around zero. The quasi-Gaussian's rigid template (zero-mean Gaussian + side mass at $\pm\Delta$) cannot adapt: feasibility requires increasing $\sigma$, eliminating the gain. The multi-Gaussian can vary $K$, performing strongly everywhere. The quasi-Gaussian is a lightweight surrogate; the multi-Gaussian is the main recommendation. We now state this more clearly.
>
> ## Question 3: Explicit $(\varepsilon,\delta)$-composition theorem
>
> Yes! Our multi-Gaussian has zCDP parameter $\rho=\Delta^2/(2\sigma^2)$ (Corollary 3.6), composing additively: $\rho_{\mathrm{tot}}=\sum_t \Delta_t^2/(2\sigma_t^2)$. The zCDP-to-$(\varepsilon,\delta)$ conversion gives $(\rho_{\mathrm{tot}} + 2\sqrt{\rho_{\mathrm{tot}}\log(1/\delta_{\mathrm{tot}})},\delta_{\mathrm{tot}})$-DP. We will state this as a theorem thanks to your suggestion.

---

> > ### Author Rebuttal · Reviewer_RPmg · 2026-04-05
> >
> > Thank you for your responses. I will increase my score. I recommend providing these points and intuitions (especially the discrete input intuition) either in the main text or in the appendix to clarify the arguments.

---

> > > ### Author Response · Authors · 2026-04-06
> > >
> > > Dear Reviewer RPmg,
> > >
> > > We are pleased to hear that you are satisfied with our rebuttal. We appreciate your constructive review. Thank you for increasing your score! We will incorporate these points into the revised version of the paper.
> > >
> > > Best regards,
> > > Authors of Paper 26836

---

### Official Review · Reviewer_22EE · 2026-03-24

**Soundness:** 3
**Presentation:** 3
**Significance:** 3
**Originality:** 3
**Overall Recommendation:** 5
**Confidence:** 3

**Summary:**

This papers the fundamental problem of one dimensional additive noise mechanisms for approximate differential privacy. This problem has been extensively studied in the past with a number of perturbation distributions being proposed. The authors point out that while the analytic Gaussian is optimal in the high privacy regime (approximately eps<1), it is not optimal in the low privacy regime. Distributions like the truncated Laplace have been proposed in the high privacy regime but these mechanisms lack the strong privacy composition properties of Gaussian noise. The authors propose a specific mixture of Gaussians mechanism that addresses both of these concerns. It outperforms the analytic Gaussian significantly in the all privacy regimes, although improvements are only significant in the high privacy regime. It outperforms the best benchmark distribution (e.g. truncated Laplace) in the high privacy regime for a single iteration. It performs the best overall over many iterations since it inherits the same strong composition properties as the standard Gaussian noise addition.

**Compliance With Llm Reviewing Policy:**

Affirmed.

**Key Questions For Authors:**

- I was surprised that the proposed mechanism was not unimodal and not something like the staircase mechanism. This seems inherent (e.g. not just an artifact of wanting to use Gaussians) from the numerically optimized distribution in Figure 1. Do the authors have intuition for this?
- Large epsilons are more common in local differential privacy than central differential privacy. Are there use cases for Gaussian noise addition in the local model? (My understanding is that it is typically outperformed by other mechanisms like RAPPOR in the high privacy regime) Could this mechanism be used in the local model?

**Limitations:**

- The problem setting is important but limited.

**Strengths And Weaknesses:**

Strengths:
- While the setting is limited (additive noise addition in the high privacy setting for one dimensional statistics), this is such a foundational and well studied problem in differential privacy that any improvement seems notable. The form of the proposed distribution is surprising (the fact that it is not unimodal is to my knowledge unique barring the numerically optimized distribution from Selvi et al.). The improvements are practically significant even for epsilon as small as 0.75.
- Analytic Gaussian noise addition is very common in implementations of DP-SGD. While this work is about the one dimensional setting, it seems like it could inspire improvements to the high dimensional version of the problem. This would be practically significant.
- The authors focus on Gaussian mixture models because standard Gaussians have strong privacy composition theorems that means they generally outperform other noise addition mechanisms for iterative algorithms. The authors prove that their proposed mechanism also satisfies these strong privacy compositions.

Weaknesses:
- The paper is clear and easy to follow. However, it provides little intuition for how the proposed distribution is designed and why it performs well for computing a single statistic. For example, I understand why Gaussian mixture models might be desirable for composition. But, why this specific Gaussian mixture model? Because it models the staircase mechanism or the numerically optimal distribution? Where do the coefficients in Defn 3.1 come from? Why space the means apart by Delta?

Minor comments:
- Most of the large epsilons mentioned were in settings that are not comparable to the work here. For example, they were in the local model of DP (e.g. Apple use cases), or the high privacy budget was split among many statistics each of which had a small privacy budget (e.g. the US Census). Are there use cases, perhaps from implementations of DPSGD, that use a large central epsilon for a single statistic?
- It would be helpful to emphasize in the text to the reader that Defn 3.1 is a specific Gaussian mixture model, not just the general defn of a Gaussian mixture model.
- Why is Defn 4.1 called a quasi-Gaussian mixture?
- I would recommend moving at least some of the results from Fig 2 to earlier in the paper. It would have helped motivate the rest of the paper to know the scale of the improvement up front.
- Fig 1 and Fig 2 seem to convey the same type of data. I thought Fig 1 was considerably easier to parse.

---

> ### Author Rebuttal · Authors · 2026-03-30
>
> We sincerely thank you for the thorough review. We are glad that you find our improvements significant and the proposed distribution surprising. We address all of your comments below.
>
> ## Weakness 1: Little intuition for the design; why this specific GMM?
>
> Thank you! We plan to add a detailed design motivation to the paper. The key idea is a "discrete first, then smooth" construction.
>
> Consider $K=1$: the multi-Gaussian noise decomposes as $\tilde{X} = \tilde{Z} + \tilde{G}$, where $\tilde{Z}$ is discrete on $\\{-a, 0, a\\}$ with $\mathbb{P}[\tilde{Z}=0]=q_\varepsilon$, $\mathbb{P}[\tilde{Z}=\pm a]=p_\varepsilon$ for weights $p_{\varepsilon}, q_{\varepsilon}$, and $\tilde{G}\sim\mathcal{N}(0,\sigma^2)$ is independent. We choose the discrete part $\tilde{Z}$ to already provide a DP guarantee by itself for a discrete query $q$ taking values in $\\{-\Delta, 0, \Delta \\}$. For neighboring queries $q(D)=0$, $q(D')=\Delta$, the supports of $q(D)+\tilde{Z}$ and $q(D')+\tilde{Z}$ are $\\{-a,0,a\\}$ and $\\{\Delta-a,\Delta,\Delta+a\\}$. If $a\notin\\{\Delta/2,\Delta\\}$, DP forces $\delta=1$. If $a=\Delta/2$, the overlap forces $\delta\geq 1-p_\varepsilon$, which exceeds the $\delta=p_\varepsilon$ achieved at $a=\Delta$. So $a=\Delta$ is optimal. Given $a=\Delta$, the event $\\{-\Delta\\}$ forces $\delta\geq p_\varepsilon$, and the DP constraint on $\\{-\Delta,0\\}$ gives $q_\varepsilon/p_\varepsilon\leq e^\varepsilon$; equality is tightest. This yields exactly our coefficients: means spaced by $\Delta$, weights scaling as $e^{-|k|\varepsilon}$.
>
> The Gaussian smoothing $\tilde{G}$ then turns this favorable discrete structure into a continuous mechanism. This also explains why multimodality helps: a single broad Gaussian centered at zero smooths out the favorable peak structure, allocating mass to suboptimal intermediate regions. Our mixture instead places a separate Gaussian bump at each preferred location, paying only a local smoothing cost.
>
> Beyond this intuition, during the rebuttals we also proved formally that for any $\delta\in(0,1/2)$, there exists $\varepsilon_0(\delta)$ such that for $\varepsilon\geq\varepsilon_0(\delta)$ the $K=1$ multi-Gaussian and quasi-Gaussian mechanisms attain strictly smaller $l_2$-loss than the optimal analytic Gaussian. The proof sets $\sigma^*=\Delta/\sqrt{2\varepsilon}$ and shows our mechanisms incur only an $O(e^{-\varepsilon})$ correction to the leading $\Delta^2/(2\varepsilon)$ loss, while the analytic Gaussian incurs $O(\varepsilon^{-3/2})$.
>
> ## Minor (1) + Q2: Large-$\varepsilon$ use cases; local DP applicability
>
> Thank you for the sharp observation that several cited examples (Apple, Census) are local DP or split budgets. We plan to cite central-DP examples thanks to your advice: the Opacus DP-SGD tutorial trains a classifier with $\varepsilon=50$, and Google's VaultGemma LLM uses sequence-level $\varepsilon=2$. Both use Gaussian noise for central DP.
>
> Regarding local DP: our mechanism applies if a user holds a bounded numeric value and adds noise locally before sending to a server. We plan to discuss this extension in the conclusions and thank you for pointing us in this direction.
>
> ## Minor (2): Defn 3.1 is a specific GMM
>
> Agreed! We will emphasize early on that Defn 3.1 is a structured Gaussian mixture with a specific weighting rule ($e^{-|k|\varepsilon}$) and fixed mean spacing ($\Delta$), not a general GMM. Thank you!
>
> ## Minor (3): Why "quasi-Gaussian"?
>
> The second component in Defn 4.1 has density proportional to $\exp(-(|x|-\Delta)^2/(2\sigma^2))$, which replaces $x$ with $|x|$ in the standard Gaussian exponent. It resembles a Gaussian but is not one, hence we think of it as a "quasi-Gaussian". We would be happy to revise this name should you have any recommendation.
>
> ## Minor (4)+(5): Figure placement and readability
>
> Thank you! We will move Figure 2 earlier in the paper to motivate the work up front. Should our paper be accepted, in the camera-ready (with the extra page), we will also present the data behind Figure 2 in tabular form similar to Table 1, which you found easier to parse.
>
> ## Question 1: Intuition for multimodality
>
> The non-unimodality is not an artifact of using Gaussians, but it reflects a fundamental structure. Selvi et al. (2025) showed that adding monotonicity constraints (mass weakly decreasing away from origin) to their numerical optimizer already produces a lower bound on achievable loss that is worse than the upper bound from an unconstrained (non-monotonic) optimized distribution. This means optimal mechanisms must place mass at locations away from zero. As explained in our response to Weakness 1, the "discrete first, then smooth" view shows exactly where this mass should sit: at multiples of $\Delta$, with geometrically decaying weights. A unimodal Gaussian smooths too aggressively, washing out this structure; a Gaussian mixture preserves it, paying only local smoothing costs.

---

> > ### Author Rebuttal · Reviewer_22EE · 2026-04-06
> >
> > Thank you to the authors for their response. Since my score was already an accept, I will keep my score.

---

> > > ### Author Response · Authors · 2026-04-07
> > >
> > > Dear Reviewer 22EE,
> > >
> > > We appreciate your rebuttal acknowledgement and the positive evaluation of our work!
> > >
> > > Best regrads,
> > > Authors of paper 26836

---

### Decision · Program_Chairs · 2026-04-30

**Decision:**

Accept (regular)

**Comment:**

The paper considers the problem of answering a scalar query with bounded sensitivity under approximate DP while minimizing the loss. The authors propose two variants of the Gaussian mechanism: a mixture of Gaussians approach and a quasi-Gaussian approach. All reviewers found the paper to be tackling a basic problem in DP, providing novel answers and with a very promising experimental section. Since the submission (plus the authors' rebuttal) has convinced all reviewers to advocate for acceptance, we have no choice but to do the same: also recommend acceptance to ICML. Terrific work!